# An Adaptive Algorithm for Bilevel Optimization on Riemannian Manifolds

**Xu Shi**[* 1]      **Rufeng Xiao**[* 1]      **Rujun Jiang**[‡ 1, 2]

[1]School of Data Science, Fudan University
[2]Shanghai Key Laboratory for Contemporary Applied Mathematics, Fudan University

{xshi22, rfxiao24}@m.fudan.edu.cn
rjjiang@fudan.edu.cn

## Abstract

Existing methods for solving Riemannian bilevel optimization (RBO) problems require prior knowledge of the problem's first- and second-order information and curvature parameter of the Riemannian manifold to determine step sizes, which poses practical limitations when these parameters are unknown or computationally infeasible to obtain. In this paper, we introduce the Adaptive Riemannian Hypergradient Descent (AdaRHD) algorithm for solving RBO problems. To our knowledge, AdaRHD is the first method to incorporate a fully adaptive step size strategy that eliminates the need for problem-specific parameters in RBO. We prove that AdaRHD achieves an $\mathcal{O}(1/\epsilon)$ iteration complexity for finding an $\epsilon$-stationary point, thus matching the complexity of existing non-adaptive methods. Furthermore, we demonstrate that substituting exponential mappings with retraction mappings maintains the same complexity bound. Experiments demonstrate that AdaRHD achieves comparable performance to existing non-adaptive approaches while exhibiting greater robustness.

## 1 Introduction

Bilevel optimization has garnered significant attention owing to its diverse applications in fields such as reinforcement learning [49, 35], meta-learning [6, 67, 41], hyperparameter optimization [26, 73, 95], adversarial learning [8, 87, 88], and signal processing [50, 24]. This paper focuses on "Riemannian bilevel optimization (RBO)" [33, 23, 54], a framework that arises in applications such as Riemannian meta-learning [80], neural architecture search [77], image segmentation [28], Riemannian min-max optimization [44, 39, 102, 30, 89, 38], and low-rank adaption [98]. The general formulation of RBO is as follows:

$$
\begin{aligned}
\min_{x \in \mathcal{M}_x} \quad & F(x) := f(x, y^*(x)), \\
\text{s.t.} \quad & y^*(x) = \arg\min_{y \in \mathcal{M}_y} g(x, y),
\end{aligned}
\tag{1}
$$

where $\mathcal{M}_x, \mathcal{M}_y$ are $d^x$- and $d^y$-dimensional complete Riemannian manifolds [54], the functions $f, g : \mathcal{M}_x \times \mathcal{M}_y \to \mathbb{R}$ are smooth, the lower-level function $g(x, y)$ is geodesic strongly convex w.r.t. $y$, and the upper-level function $f$ is not required to be convex.

To address Problem (1), we extend the recent advances in Euclidean adaptive bilevel optimization [93] to propose an adaptive Riemannian hypergradient method. Although certain aspects of Euclidean

---

[*]Equal contributions.

[‡]Corresponding author.

39th Conference on Neural Information Processing Systems (NeurIPS 2025).

analysis extend naturally to Riemannian settings, the manifold's curvature parameters introduce many analytical challenges. We propose key technical difficulties in designing adaptive methods for solving RBO problems as follows:

(i) RBO necessitates that variables are constrained on Riemannian manifolds, making it intrinsically more complex than its Euclidean counterpart. Furthermore, critical parameters such as geodesic strong convexity, Lipschitz continuity, and curvature parameters over Riemannian manifolds are more computationally demanding to estimate than their Euclidean counterparts. Consequently, the geometric structure of the manifold creates distinct analytical challenges in the convergence analysis of adaptive algorithms.

(ii) Compared to single-level Riemannian (adaptive) optimization, RBO inherently involves interdependent variable updates, resulting in complex step size selection challenges arising from coupled (hyper)gradient dynamics.

(iii) Unlike non-adaptive RBO methods, adaptive step-size strategies can cause divergent behavior during initial iterations since the initial step size may be too large, which must be rigorously addressed to enable robust theoretical convergence analysis.

In this work, we present a comprehensive convergence analysis that demonstrates our approach to resolving these challenges.

## 1.1 Related works

Table 1 summarizes key studies on (Riemannian) bilevel optimization that are relevant to our methods. The table compares their applicable scenarios, whether they are adaptive, and their computational complexity of first- and second-order information. Note that constants, such as condition numbers, are omitted from the complexity analysis for simplicity. For a comprehensive review of the related works, please refer to Appendix A.

Table 1: Comparisons of first-order and second-order complexities for reaching an $\epsilon$-stationary point. Here, "Euc" and "Rie" represent the feasible region of the algorithms are Euclidean space and Riemannian manifolds, respectively. The notations "Det", "F-S", and "Sto" represent the applicable problems as deterministic, function-sum, and stochastic, respectively. Additionally, $G_f$ and $G_g$ are the gradient complexities of $f$ and $g$, respectively. $JV_g$ and $HV_g$ are the complexities of computing the second-order cross derivative and Hessian-vector product of $g$. The notation $\tilde{\mathcal{O}}$ denotes the omission of logarithmic terms in contrast to the standard $\mathcal{O}$ notation. Furthermore, the notation "NA" represents that the corresponding complexity is not applicable.

| Methods | Space | Adaptive | Type | $G_f$ | $G_g$ | $JV_g$ | $HV_g$ |
|---|---|---|---|---|---|---|---|
| D-TFBO [93] | Euc | ✓ | Det | $\mathcal{O}(1/\epsilon)$ | $\mathcal{O}(1/\epsilon^2)$ | $\mathcal{O}(1/\epsilon)$ | $\mathcal{O}(1/\epsilon^2)$ |
| S-TFBO [93] | | | | $\tilde{\mathcal{O}}(1/\epsilon)$ | $\tilde{\mathcal{O}}(1/\epsilon)$ | $\tilde{\mathcal{O}}(1/\epsilon)$ | $\tilde{\mathcal{O}}(1/\epsilon)$ |
| RHGD-HINV [33] | Rie | ✗ | Det | $\mathcal{O}(1/\epsilon)$ | $\tilde{\mathcal{O}}(1/\epsilon)$ | $\mathcal{O}(1/\epsilon)$ | NA |
| RHGD-CG [33] | | | | $\mathcal{O}(1/\epsilon)$ | $\tilde{\mathcal{O}}(1/\epsilon)$ | $\mathcal{O}(1/\epsilon)$ | $\tilde{\mathcal{O}}(1/\epsilon)$ |
| RHGD-NS [33] | | | | $\mathcal{O}(1/\epsilon)$ | $\tilde{\mathcal{O}}(1/\epsilon)$ | $\mathcal{O}(1/\epsilon)$ | $\tilde{\mathcal{O}}(1/\epsilon)$ |
| RHGD-AD [33] | | | | $\mathcal{O}(1/\epsilon)$ | $\tilde{\mathcal{O}}(1/\epsilon)$ | $\mathcal{O}(1/\epsilon)$ | $\tilde{\mathcal{O}}(1/\epsilon)$ |
| RSHGD-HINV [33] | | | F-S | $\mathcal{O}(1/\epsilon^2)$ | $\tilde{\mathcal{O}}(1/\epsilon^2)$ | $\mathcal{O}(1/\epsilon^2)$ | NA |
| RieBO [54] | Rie | ✗ | Det | $\mathcal{O}(1/\epsilon)$ | $\tilde{\mathcal{O}}(1/\epsilon)$ | $\mathcal{O}(1/\epsilon)$ | $\tilde{\mathcal{O}}(1/\epsilon)$ |
| RieSBO [54] | | | Sto | $\mathcal{O}(1/\epsilon^2)$ | $\tilde{\mathcal{O}}(1/\epsilon^2)$ | $\mathcal{O}(1/\epsilon^2)$ | $\tilde{\mathcal{O}}(1/\epsilon^2)$ |
| RF²SA [23] | Rie | ✗ | Det | $\tilde{\mathcal{O}}(1/\epsilon^{3/2})$ | $\tilde{\mathcal{O}}(1/\epsilon^{3/2})$ | NA | NA |
| RF²SA [23] | | | Sto | $\tilde{\mathcal{O}}(1/\epsilon^{7/2})$ | $\tilde{\mathcal{O}}(1/\epsilon^{7/2})$ | NA | NA |
| **AdaRHD-GD (Ours)** | Rie | ✓ | Det | $\mathcal{O}(1/\epsilon)$ | $\mathcal{O}(1/\epsilon^2)$ | $\mathcal{O}(1/\epsilon)$ | $\mathcal{O}(1/\epsilon^2)$ |
| **AdaRHD-CG (Ours)** | | | | $\mathcal{O}(1/\epsilon)$ | $\mathcal{O}(1/\epsilon^2)$ | $\mathcal{O}(1/\epsilon)$ | $\tilde{\mathcal{O}}(1/\epsilon)$ |

## 1.2 Contributions

This paper introduces the Adaptive Riemannian Hypergradient Descent (AdaRHD) algorithm, the first method to incorporate a fully adaptive step size strategy, eliminating the need for parameter-specific prior knowledge for RBO. The contributions of this work are summarized as follows:

(i) We develop an adaptive algorithm for solving RBO problems, eliminating reliance on prior knowledge of strong convexity, Lipschitzness, or curvature parameters. Our method achieves a convergence rate matching those of non-adaptive parameter-dependent approaches. Furthermore, by replacing exponential mappings with computationally efficient retraction mappings, our algorithm maintains comparable convergence guarantees while reducing computational costs.

(ii) We establish upper bounds for the total iterations of resolutions of the lower-level problem in Problem (1) and the corresponding linear system required for computing the Riemannian hypergradient. The derived bounds match the iteration complexity of the standard AdaGrad-Norm method for solving the strongly convex problems [92].

(iii) We evaluate our algorithm against existing methods on various problems over Riemannian manifolds, including the Riemannian hyper-representation and robust optimization problems. The results demonstrate that our method performs comparably with non-adaptive methods while exhibiting significantly greater robustness.

## 2 Preliminaries

In this section, we review standard definitions and preliminary results on Riemannian optimization. All results presented here are available in the literature [53, 10, 33, 54], we restate them for conciseness.

We first establish essential properties and notations for Riemannian manifolds. The definition of a Riemannian manifold has been conducted in the literature [53, 10]. The Riemannian metric on a Riemannian manifold $\mathcal{M}$ at $x \in \mathcal{M}$ is denoted by $\langle \cdot, \cdot \rangle_x : T_x\mathcal{M} \times T_x\mathcal{M} \to \mathbb{R}$, with the induced norm on the tangent space $T_x\mathcal{M}$ defined as $\|u\|_x = \sqrt{\langle u, u \rangle_x}$ for any $u \in T_x\mathcal{M}$. We then recall the definitions of exponential mapping and parallel transport. For an tangent vector $u \in T_x\mathcal{M}$ and a geodesic $c : [0, 1] \to \mathcal{M}$ satisfying $c(0) = x$, $c(1) = y$, and $c'(0) = u$, the exponential mapping $\mathrm{Exp}_x : T_x\mathcal{M} \to \mathcal{M}$ is defined as $\mathrm{Exp}_x(u) = y$. The inverse of the exponential mapping, $\mathrm{Exp}_x^{-1} : \mathcal{M} \to T_x\mathcal{M}$, is called logarithm mapping [10, Definition 10.20], and the Riemannian distance between two points $x, y \in \mathcal{M}$ is given by $d(x, y) = \|\mathrm{Exp}_x^{-1}(y)\|_x = \|\mathrm{Exp}_y^{-1}(x)\|_y$. Parallel transport $\mathcal{P}_{x_1}^{x_2} : T_{x_1}\mathcal{M} \to T_{x_2}\mathcal{M}$ is a linear operator that preserves the inner product structure, satisfying $\langle u, v \rangle_{x_1} = \langle \mathcal{P}_{x_1}^{x_2}u, \mathcal{P}_{x_1}^{x_2}v \rangle_{x_2}$ for all $u, v \in T_{x_1}\mathcal{M}$.

We now define key properties of functions on Riemannian manifolds. The Riemannian gradient $\mathcal{G}f(x) \in T_x\mathcal{M}$ of a differentiable function $f : \mathcal{M} \to \mathbb{R}$ is the unique vector satisfying

$$\langle \mathcal{G}f(x), u \rangle_x = \mathrm{D}f(x)[u], \quad \forall u \in T_x\mathcal{M},$$

where $\mathrm{D}f(x)[u]$ denotes the directional derivative of $f$ at $x$ along $u$ [84, 10, 54]. For twice-differentiable $f$, the Riemannian Hessian $\mathcal{H}f(x)$ is defined as the covariant derivative of $\mathcal{G}f(x)$. A function $f : \mathcal{M} \to \mathbb{R}$ is "geodesically (strongly) convex" if $f(c(t))$ is (strongly) convex w.r.t. $t \in [0, 1]$ for all geodesics $c : [0, 1] \to \Omega$, where $\Omega \subseteq \mathcal{M}$ is a geodesically convex set [54, Definition 4]. Particularly, if $f$ is twice-differentiable, $\mu$-geodesic strong convexity of $f$ is equivalent to $\mathcal{H}f(x) \succeq \mu\mathrm{Id}$, where $\mathrm{Id}$ is the identity operator. Further, $f$ is "geodesically $L_f$-Lipschitz smooth" if

$$\|\mathcal{G}f(x) - \mathcal{P}_y^x\mathcal{G}f(y)\|_x \leq L_f d(x, y), \quad \forall x, y \in \mathcal{M},$$

and for twice-differentiable $f$, this property is equivalent to $\mathcal{H}f(x) \preceq L_f\mathrm{Id}$.

Finally, for a bi-function $f : \mathcal{M}_x \times \mathcal{M}_y \to \mathbb{R}$, the Riemannian gradients of $f$ w.r.t. $x$ and $y$ are denoted by $\mathcal{G}_x f(x, y)$ and $\mathcal{G}_y f(x, y)$, respectively, while the Riemannian Hessians are $\mathcal{H}_x f(x, y)$ and $\mathcal{H}_y f(x, y)$. Moreover, the Riemannian cross-derivatives $\mathcal{G}_{xy}^2 f(x, y) : T_y\mathcal{M}_y \to T_x\mathcal{M}_x$ and $\mathcal{G}_{yx}^2 f(x, y) : T_x\mathcal{M}_x \to T_y\mathcal{M}_y$, discussed in [33, 54], are linear operators. Furthermore, the operator norm of a linear operator $\mathcal{G} : T_y\mathcal{M}_y \to T_x\mathcal{M}_x$ is defined as

$$\|\mathcal{G}\|_x = \sup_{u \in T_y\mathcal{M}_y, \|u\|_y = 1} \|\mathcal{G}[u]\|_x.$$

Building on the concepts of Lipschitz continuity (cf. Definition B.1) and geodesic strong convexity for functions defined on Riemannian manifolds, we present the following proposition.

**Proposition 2.1** ([10, 33, 54]). *For a function $f : \mathcal{M} \to \mathbb{R}$, if its Riemannian gradient $\mathcal{G}f$ is $L$-Lipschitz continuous, then for all $x, y \in \mathcal{U} \subseteq \mathcal{M}$, it holds that*

$$f(y) \leq f(x) + \left\langle \mathcal{G}f(x), \mathrm{Exp}_x^{-1}(y) \right\rangle_x + \frac{L}{2}d^2(x, y).$$

*If $f$ is $\mu$-geodesic strongly convex, then for all $x, y \in \mathcal{U}$, it holds that*

$$f(y) \geq f(x) + \left\langle \mathcal{G}f(x), \mathrm{Exp}_x^{-1}(y) \right\rangle_x + \frac{\mu}{2}d^2(x, y). \tag{2}$$

Additionally, for the Lipschitzness of the functions and operators, and an important result of the trigonometric distance bound over the Riemannian manifolds, we present them in Appendix B.

# 3 Adaptive Riemannian hypergradient descent algorithms

Standard Riemannian bilevel optimization (RBO) approaches [23, 33, 54] determine step sizes for updating the variables using problem-specific parameters such as strong convexity, Lipschitzness, and curvature constants. However, these parameters are frequently impractical to estimate or compute, posing challenges for step size determination. This limitation underscores the need for adaptive RBO algorithms that operate without prior parameter knowledge.

## 3.1 Approximate Riemannian hypergradient

Prior to presenting our primary algorithm, we first introduce the Riemannian hypergradient, as the core concept of our methodology relies on this construct. Following general bilevel optimization frameworks, we define the Riemannian hypergradient of $F(x)$ in Problem (1) as follows:

**Proposition 3.1.** *[33, 54] The Riemannian hypergradient of $F(x)$ are given by*

$$\mathcal{G}F(x) = \mathcal{G}_x f(x, y^*(x)) - \mathcal{G}_{xy}^2 g(x, y^*(x)) \mathcal{H}_y^{-1} g(x, y^*(x)) [\mathcal{G}_y f(x, y^*(x))] \in T_x \mathcal{M}. \tag{3}$$

Proposition 3.1 fundamentally hinges on the implicit function theorem for Riemannian manifolds [30], which necessitates the invertibility of the Hessian of the lower-level objective function $g$ w.r.t. $y$. In this paper, this requirement is satisfied as $g$ is geodesically strongly convex w.r.t. $y$. Consequently, $y^*(x)$ is unique and differentiable.

However, the Riemannian hypergradient defined in (3) is computationally challenging to evaluate. First, the exact solution $y^*(x)$ of the lower-level objective is not explicitly available, necessitating the use of an approximate solution $\hat{y}$, we employ the adaptive Riemannian gradient descent method to compute this approximation. Second, calculating the Hessian-inverse-vector product $\mathcal{H}_y^{-1} g(x, y^*(x))[\mathcal{G}_y f(x, y^*(x))]$ in (3) incurs prohibitive computational costs. To address this, given the approximation $\hat{y}$ of $y^*(x)$, we can approximate the Hessian-inverse-vector product $\mathcal{H}_y^{-1} g(x, \hat{y})[\mathcal{G}_y f(x, \hat{y})]$ by solving the linear system $\mathcal{H}_y g[v](x, \hat{y}) = \mathcal{G}_y f(x, \hat{y})$, which is originated from the following quadratic problem:

$$\min_{v \in T_{\hat{y}}\mathcal{M}_y} R(x, \hat{y}, v) = \frac{1}{2} \left\langle v, \mathcal{H}_y g(x, \hat{y})[v] \right\rangle_{\hat{y}} - \left\langle v, \mathcal{G}_y f(x, \hat{y}) \right\rangle_{\hat{y}}. \tag{4}$$

Denote $\hat{v}$ as the approximate solution of Problem (4). In this paper, we use the adaptive gradient descent method (cf. Steps 11-15 of Algorithm 1) or the tangent space conjugate gradient method [81, 10] (cf. Step 16 of Algorithm 1) to obtain such a $\hat{v}$. Specifically, the tangent space conjugate gradient algorithm is presented in Algorithm 2 of Appendix C.

Then, given the estimations $\hat{y}$ and $\hat{v}$, the approximate Riemannian hypergradient [23, 33, 54] is defined as follows,

$$\widehat{\mathcal{G}}F(x, \hat{y}, \hat{v}) = \mathcal{G}_x f(x, \hat{y}) - \mathcal{G}_{xy}^2 g(x, \hat{y})\hat{v} \in T_x \mathcal{M}_x. \tag{5}$$

## 3.2 Adaptive Riemannian hypergradient descent algorithm: AdaRHD

Motivated by the recent adaptive methods [93] for solving general bilevel optimization, we employ the "inverse of cumulative (Riemannian) gradient norm" strategy to design the adaptive step sizes [92, 90, 93], i.e., adapting the step sizes based on accumulated Riemannian (hyper)gradient norms.

Given the total iterations $T$, set the accuracies for solving the lower-level problem of Problem (1) and linear system as $\epsilon_y = 1/T$ and $\epsilon_v = 1/T$. Our Adaptive Riemannian Hypergradient Descent (AdaRHD) algorithm for solving Problem (1) is presented in Algorithm 1. For simplicity, when employing the gradient descent (Steps 11-15) and conjugate gradient (Step 16) to solve the linear system, respectively, we denote AdaRHD as AdaRHD-GD and AdaRHD-CG, respectively.

---

**Algorithm 1 Ada**ptive **R**iemannian **H**ypergradient **D**escent (AdaRHD)

---
1: Initial points $x_0 \in \mathcal{M}_x, y_0 \in \mathcal{M}_y$, and $v_0 \in T_{y_0}\mathcal{M}$, initial step sizes $a_0 > 0, b_0 > 0$, and $c_0 > 0$, total iterations $T$, and error tolerances $\epsilon_y = \epsilon_v = \frac{1}{T}$.
2: **for** $t = 0, 1, 2, ..., T-1$ **do**
3:      Set $k = 0$ and $y_t^0 = y_{t-1}^{K_{t-1}}$ if $t > 0$ and $y_0$ otherwise.
4:      **while** $\|\mathcal{G}_y g(x_t, y_t^k)\|_{y_t^k}^2 > \epsilon_y$ **do**
5:          $b_{k+1}^2 = b_k^2 + \|\mathcal{G}_y g(x_t, y_t^k)\|_{y_t^k}^2$,
6:          $y_t^{k+1} = \mathrm{Exp}_{y_t^k}(-\frac{1}{b_{k+1}}\mathcal{G}_y g(x_t, y_t^k))$,
7:          $k = k + 1$.
8:      **end while**
9:      $K_t = k$.
10:     Set $n = 0$ and $v_t^0 = \mathcal{P}_{y_{t-1}^{K_{t-1}}}^{y_t^{K_t}} v_{t-1}^{N_{t-1}}$ if $t > 0$ and $v_0$ otherwise.
11:     **while** $\|\nabla_v R(x_t, y_t^{K_t}, v_t^n)\|_{y_t^{K_t}}^2 > \epsilon_v$ **do**
12:         $c_{n+1}^2 = c_n^2 + \|\nabla_v R(x_t, y_t^{K_t}, v_t^n)\|_{y_t^{K_t}}^2$,
13:         $v_t^{n+1} = v_t^n - \frac{1}{c_{n+1}}\nabla_v R(x_t, y_t^{K_t}, v_t^n)$,          ▷ Gradient descent
14:         $n = n + 1$.
15:     **end while**
16:     **Or** set $v_t^0 = 0$ and invoke $v_t^n = \mathrm{TSCG}(\mathcal{H}_y g(x_t, y_t^{K_t}), \mathcal{G}_y f(x_t, y_t^{K_t}), v_t^0, \epsilon_v)$.   ▷ Conjugate gradient
17:     $N_t = n$.
18:     $\widehat{\mathcal{G}}F(x_t, y_t^{K_t}, v_t^{N_t}) = \mathcal{G}_x f(x_t, y_t^{K_t}) - \mathcal{G}_{xy}^2 g(x_t, y_t^{K_t})[v_t^{N_t}]$,
19:     $a_{t+1}^2 = a_t^2 + \|\widehat{\mathcal{G}}F(x_t, y_t^{K_t}, v_t^{N_t})\|_{x_t}^2$,
20:     $x_{t+1} = \mathrm{Exp}_{x_t}(-\frac{1}{a_{t+1}}\widehat{\mathcal{G}}F(x_t, y_t^{K_t}, v_t^{N_t}))$.
21: **end for**

---

### 3.3 Convergence analysis

This section establishes the errors between the approximate Riemannian hypergradient (5) and the exact Riemannian hypergradient (3), the convergence result of Algorithm 1, and the corresponding computation complexity.

#### 3.3.1 Definitions and assumptions

Similar to the definition of an $\epsilon$-stationary point in general bilevel optimization [28, 43, 42], the definition of an $\epsilon$-stationary point in Riemannian bilevel optimization [23, 33, 54] is given as follows.

**Definition 3.1** ($\epsilon$-stationary point). *A point $x \in \mathcal{M}_x$ is an $\epsilon$-stationary point of Problem* (1) *if it satisfies* $\|\mathcal{G}F(x)\|_x^2 \leq \epsilon$.

We first concern the upper bound of curvature constant $\zeta(\tau, c)$ defined in Lemma B.1.

**Assumption 3.1.** *The manifolds $\mathcal{M}_x$ and $\mathcal{M}_y$ are complete Riemannian manifolds. Moreover, $\mathcal{M}_y$ is a Hadamard manifold whose sectional curvature is lower bounded by $\tau < 0$. Moreover, for all iterates $y_t^k$ in the lower-level problem, the curvature constant $\zeta(\tau, d(y_t^k, y^*(x_t)))$ defined in Lemma B.1 is always upper bounded by a constant $\zeta$ for all $t \geq 0$ and $k \geq 0$.*

Assumption 3.1 is commonly used in the Riemannian optimization, which ensures that the lower-level objective $g$ can be geodesically strongly convex [100, 54] and is essential for the convergence of the lower-level problem [33, 54].

We then adopt the following assumptions concerning the fundamental properties of the upper- and lower-level objectives of Problem (1).

**Assumption 3.2.** *Function $f(x, y)$ is continuously differentiable and $g(x, y)$ is twice continuously differentiable for all $(x, y) \in \mathcal{U} = \mathcal{U}_x \times \mathcal{U}_y \subseteq \mathcal{M}_x \times \mathcal{M}_y$, and $g(x, y)$ is $\mu$-geodesic strongly convex w.r.t. $y \in \mathcal{U}_y$ for any $x \in \mathcal{U}_x$.*

This assumption is prevalent in (Riemannian) bilevel optimization [28, 43, 42, 57, 35, 51, 23, 33, 54]. Under this assumption, the optimal solution $y^*(x)$ of the lower-level problem is unique for all $x \in \mathcal{M}_x$ and differentiable [33, 54], ensuring the existence of the Riemannian hypergradient (3.1).

**Assumption 3.3.** *Function $f(x, y)$ is $l_f$-Lipschitz continuous in $\mathcal{U} \subseteq \mathcal{M}_x \times \mathcal{M}_y$. The Riemannian gradients $\mathcal{G}_x f(x, y)$ and $\mathcal{G}_y f(x, y)$ are $L_f$-Lipschitz continuous in $\mathcal{U}$. The Riemannian gradients $\mathcal{G}_x g(x, y)$ and $\mathcal{G}_y g(x, y)$ are $L_g$-Lipschitz continuous in $\mathcal{U}$. Furthermore, the Riemannian Hessian $\mathcal{H}_y g(x, y)$, cross derivatives $\mathcal{G}_{xy}^2 g(x, y)$, $\mathcal{G}_{yx}^2 g(x, y)$ are $\rho$-Lipschitz in $\mathcal{U}$.*

Assumption 3.3 is a standard condition in the existing literature on (Riemannian) bilevel optimization [28, 43, 42, 33, 54]. These conditions ensure the smoothness of the objective function $F$ in Problem (1) and the Riemannian hypergradient $\mathcal{G}F$ defined in (3).

**Assumption 3.4.** *The minimum of $F$ over $\mathcal{M}_x$, denote as $F^*$, is lower-bounded.*

Assumption 3.4 concerns the existence of the minimum of $F$, which is a common requirement in the literature of adaptive (bilevel) optimization problems [90, 92, 93].

### 3.3.2 Convergence results

Denote $\hat{v}_t^*(x, y) := \arg\min_{v \in T_y \mathcal{M}_y} R(x, y, v)$. We can then bound the estimation error of the proposed schemes of approximated hypergradient as follows.

**Lemma 3.1** (Hypergradient approximation error bound). *Suppose that Assumptions 3.1, 3.2, 3.3, and 3.4 hold. The error for the approximated Riemannian hypergradient generated by Algorithm 1 satisfies,*

$$\|\widehat{\mathcal{G}}F(x_t, y_t^{K_t}, v_t^{N_t}) - \mathcal{G}F(x_t)\|_{x_t} \leq L_1 d\left(y^*(x_t), y_t^{K_t}\right) + L_g \|v_t^{N_t} - \hat{v}_t^*(x_t, y_t^{K_t})\|_{y_t^{K_t}},$$

*where $L_1 := L_f + \frac{l_f \rho}{\mu} + L_g\left(\frac{L_f}{\mu} + \frac{l_f \rho}{\mu^2}\right)$.*

Lemma 3.1 establishes that the error between the approximate and exact Riemannian hypergradients is bounded by the errors arising from the resolutions of the lower-level problem in Problem (1) and the linear system (4). Consequently, we can actually integrate the stopping criterion $\|\widehat{\mathcal{G}}F(x_t, y_t^{K_t}, v_t^{N_t})\|_{x_t}^2 \leq 1/T$ into Algorithm 1 to enable early termination. By Lemma 3.1, this criterion ensures that $\|\mathcal{G}F(x_t)\|_{x_t}^2 \leq \mathcal{O}(1/T)$, which implies that $x_t$ is an $\mathcal{O}(1/T)$-stationary point of Problem (1). Moreover, integrating this criterion does not increase the overall computational complexity of Algorithm 1, since the norm of $\widehat{\mathcal{G}}F(x_t, y_t^{K_t}, v_t^{N_t})$ is already computed at Step 20.

Similar to Proposition 2 in [93], we derive upper bounds on the total number of iterations required to solve both the lower-level problem in Problem (1) and the linear system (4). However, owing to the geometric structures inherent in Riemannian manifolds, these bounds cannot be directly inferred from [93, Proposition 2] and require a meticulous analysis.

**Proposition 3.2.** *Suppose that Assumptions 3.1, 3.2, 3.3, and 3.4 hold. Then, for any $0 \leq t \leq T$, the numbers of iterations $K_t$ and $N_t$ required in Algorithm 1 satisfy:*

$$K_t \leq \frac{\log(C_b^2/b_0^2)}{\log(1 + \epsilon_y/C_b^2)} + \frac{1}{(\mu/\bar{b} - \zeta L_g^2/\bar{b}^2)} \log\left(\frac{\tilde{b}}{\epsilon_y}\right).$$

*AdaRHD-GD:*

$$N_t \leq \frac{\log(C_c^2/c_0^2)}{\log(1 + \epsilon_v/C_c^2)} + \frac{\bar{c}}{\mu} \log\left(\frac{\tilde{c}}{\epsilon_v}\right),$$

*where $C_b, C_c, \bar{b}, \bar{c}, \tilde{b},$ and $\tilde{c}$ are constants defined in Appendix G.*

*AdaRHD-CG:*

$$N_t \leq \frac{1}{2}\log\left(\frac{4L_g^3 l_f^2}{\mu^3 \epsilon_v}\right)/\log\left(\frac{\sqrt{L_g/\mu} + 1}{\sqrt{L_g/\mu} - 1}\right).$$

**Remark 3.1.** *Proposition 3.2 provides upper bounds for the total iterations of the resolutions of the lower-level problem in Problem (1) and the linear system (4). Since $1/\log(1 + \epsilon_y)$ is of the same order as $1/\epsilon_y$, it holds that $K_t = \mathcal{O}(1/\epsilon_y)$, which matches the standard AdaGrad-Norm for solving*

*strongly convex problems [92]. Additionally, we also have $N_t = \mathcal{O}(1/\epsilon_v)$ and $N_t = \mathcal{O}(\log(1/\epsilon_v))$ when employing gradient descent and conjugate gradient methods, respectively. Moreover, similar to AdaGrad-Norm [92], the step size adaptation for the lower-level problem proceeds in two stages: Stage 1 requires at most $\mathcal{O}(1/\log(1 + \epsilon_y))$ iterations, while Stage 2 requires at most $\mathcal{O}(\log(1/\epsilon_y))$ iterations. Furthermore, due to geometric properties of Riemannian manifolds, these bounds cannot be directly derived from [93] and introduce additional technical challenges. Specifically, the constants $C_b, C_c, \bar{b}, \bar{c}, \tilde{b}$, and $\tilde{c}$ are all related to curvature $\zeta$, and increase as $\zeta$ increases.*

Based on Proposition 3.2, we have the following convergence result of Algorithm 1.

**Theorem 3.1.** *Suppose that Assumptions 3.1, 3.2, 3.3, and 3.4 hold. The sequence $\{x_t\}_{t=0}^T$ generated by Algorithm 1 satisfies,*

$$\frac{1}{T} \sum_{t=0}^{T-1} \|\mathcal{G}F(x_t)\|_{x_t}^2 \leq \frac{C}{T} = \mathcal{O}\left(\frac{1}{T}\right),$$

*where $C$ is a constant defined in Appendix G.*

Theorem 3.1 demonstrates that our proposed adaptive algorithm achieves a convergence rate matching that of standard non-adaptive (Riemannian) bilevel algorithms (e.g., [28, 43, 42, 33, 54]), thereby demonstrating its computational efficiency. Moreover, although Algorithm 1 shares a similar algorithmic structure with D-TFBO [93] and some Euclidean analyses extend naturally to Riemannian settings, its convergence analysis still presents significant theoretical challenges. For example, the geometric structure of Riemannian manifolds necessitates the use of the trigonometric distance bound (Lemma B.1) instead of its Euclidean counterpart and requires incorporating the curvature constant into the step size adaptation (cf. Proposition 3.2), which substantially increases the analytical difficulty. We subsequently derive the complexity bound for Algorithm 1.

**Corollary 3.1.** *Suppose that Assumptions 3.1, 3.2, 3.3, and 3.4 hold. Algorithm 1 needs $T = \mathcal{O}(1/\epsilon)$ iterations to achieve an $\epsilon$-accurate stationary point of Problem (1). The gradient complexities of $f$ and $g$ are $G_f = \mathcal{O}(1/\epsilon)$ and $G_g = \mathcal{O}(1/\epsilon^2)$, respectively. The complexities of computing the second-order cross derivative and Hessian-vector product of $g$ are $JV_g = \mathcal{O}(1/\epsilon)$, $HV_g = \mathcal{O}(1/\epsilon^2)$ for AdaRHD-GD, and $HV_g = \tilde{\mathcal{O}}(1/\epsilon)$ for AdaRHD-CG, respectively.*

Corollary 3.1 demonstrates that the gradient complexity ($G_g$) and Hessian-vector product complexity ($HV_g$) of AdaRHD-GD surpass those of non-adaptive Riemannian bilevel optimization methods [33, 54] by a factor of $1/\epsilon$. This gap originates from the additional iterations needed to guarantee the solution of the lower-level problem in Problem (1) and linear system 4, which is necessitated by the lack of prior knowledge regarding the strong convexity, Lipschitzness, and curvature constants.

Designing algorithms to eliminate the $1/\epsilon$-gap is an interesting future direction. Potential strategies to bridge this gap could be drawn from adaptive methods in Riemannian optimization, such as sharpness-aware minimization on Riemannian manifolds (Riemannian SAM) [96], Riemannian natural gradient descent (RNGD) [37], and the framework for Riemannian adaptive optimization [70]. Riemannian extensions of Euclidean approaches like adaptive accelerated gradient descent [61], adaptive proximal gradient method (adaPGM) [52], adaptive Barzilai-Borwein algorithm (AdaBB) [103], and adaptive Nesterov accelerated gradient (AdaNAG) [76] may also be viable alternatives to the inverse of cumulative gradient norms strategy [92] employed in Algorithm 1.

### 3.4 Extend to retraction mapping

Given that retraction mapping is often preferred in practice for its computational efficiency over exponential mapping, this section demonstrates that its incorporation into Algorithm 1 achieves comparable theoretical convergence guarantees. A modified version of Algorithm 1, incorporating retraction mapping, is presented in Algorithm 3 (cf. Appendix D). To formalize this extension, the second condition of Assumption 3.1 must be modified as follows.

**Assumption 3.5.** *All the iterates of the lower-level problem generated by Algorithm 3 lie in a bounded set belonging to $\mathcal{M}_y$ that contains the optimal solution, i.e., there exists a constant $\bar{D} > 0$ such that $d(y_t^k, y^*(x_t)) \leq \bar{D}$ holds for all $t \geq 0$ and $k \geq 0$.*

Under Assumption 3.5, the curvature constant $\zeta(\tau, d(y_t^k, y^*(x_t)))$ in Assumption 3.1 is bounded by its definition, thereby ensuring consistency with Assumption 3.1. Additionally, it is necessary to consider the error between the exponential and retraction mappings.

**Assumption 3.6.** *Given $z_1 \in \mathcal{U} \subseteq \mathcal{M}$ (here $\mathcal{M}$ can be $\mathcal{M}_x$ or $\mathcal{M}_y$) and $u \in T_{z_1}\mathcal{M}$, let $z_2 = \text{Retr}_{z_1}(u)$. There exist constants $c_u \geq 1, c_R \geq 0$ such that $d^2(z_1, z_2) \leq c_u \|u\|_{z_1}^2$ and $\|\text{Exp}_{z_1}^{-1}(z_2) - u\|_{z_1} \leq c_R \|u\|_{z_1}^2$.*

Assumption 3.6 is standard (e.g., [33, Assumption 2] and [46, Assumption 1]) in bounding the error between the exponential mapping and retraction mapping, given that retraction mapping is a first-order approximation to the exponential mapping [33].

Then, similar to Theorem 3.1, we have the following convergence result of Algorithm 3.

**Theorem 3.2.** *Suppose that Assumptions 3.1, 3.2, 3.3, 3.4, 3.5, and 3.6 hold. The sequence $\{x_t\}_{t=0}^T$ generated by Algorithm 3 satisfies,*

$$\frac{1}{T} \sum_{t=0}^{T-1} \|\mathcal{G}F(x_t)\|_{x_t}^2 \leq \frac{C_{\text{retr}}}{T} = \mathcal{O}\left(\frac{1}{T}\right),$$

*where $C_{\text{retr}}$ is a constant defined in Appendix H.*

Algorithm 3 achieves a convergence rate nearly identical to that of Algorithm 1 while being more practical to implement. Furthermore, the complexity analysis of Algorithm 3 closely aligns with that of Algorithm 1 and is, therefore, omitted here.

# 4 Experimental results

In this section, adhering to the experimental framework established by [33], we conduct comprehensive experiments to evaluate our algorithm against the RHGD method. Since the RieBO algorithm proposed by [54] shares the same algorithmic framework as RHGD, we categorize them as a single method for comparison. We designate RHGD-50 and RHGD-20 as configurations with maximum iteration limits for solving the lower-level problem in RHGD set to 50 and 20, respectively. For consistency, we exclusively employ Algorithm 3 in this section due to its computational efficiency in practice. Detailed experimental settings and additional experiments are provided in Appendix I, and our codes are available at `https://github.com/RufengXiao/AdaRHD`.

## 4.1 Simple problem

In the first experiment, following [33], we consider a simple problem, which aims to determine the maximum similarity between two matrices $\mathbf{X} \in \mathbb{R}^{n \times d}$ and $\mathbf{Y} \in \mathbb{R}^{n \times r}$, where $n \geq d \geq r$, formulated as:

$$\max_{\mathbf{W} \in \text{St}(d,r)} \quad \text{trace}(\mathbf{M}^*(\mathbf{W})\mathbf{X}^\top \mathbf{Y}\mathbf{W}^\top)$$
$$\text{s.t.} \quad \mathbf{M}^*(\mathbf{W}) = \underset{\mathbf{M} \in \mathbb{S}_{++}^d}{\arg\min} \ \langle \mathbf{M}, \mathbf{X}^\top \mathbf{X} \rangle + \langle \mathbf{M}^{-1}, \mathbf{W}\mathbf{Y}^\top \mathbf{Y}\mathbf{W}^\top + \lambda \mathbf{I} \rangle,$$

where $\text{St}(d,r) = \{\mathbf{W} \in \mathbb{R}^{d \times r} : \mathbf{W}^\top \mathbf{W} = \mathbf{I}_r\}$ and $\mathbb{S}_{++}^d = \{\mathbf{M} \in \mathbb{R}^{d \times d} : \mathbf{M} \succ 0\}$ represent the Stiefel manifold and the symmetric positive definite (SPD) matrices, respectively, and $\lambda > 0$ is the regular parameter. The matrix $\mathbf{W} \in \text{St}(d,r)$ aligns $\mathbf{X}$ and $\mathbf{Y}$ in a shared dimensional space, while the lower-level problem learns an appropriate geometric metric $\mathbf{M} \in \mathbb{S}_{++}^d$ [97]. Additionally, the geodesic strong convexity of the lower-level problem and the Hessian inverse expression can be found in Appendix H of [33]. In this experiment, we generate random data matrices $\mathbf{X}$ and $\mathbf{Y}$ with two sample sizes: $n = 100$ and $n = 1000$, where $d = 50$ and $r = 20$.

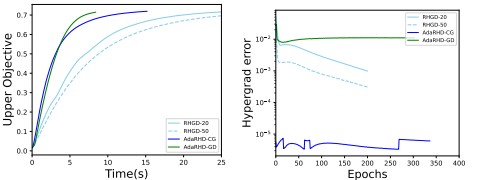
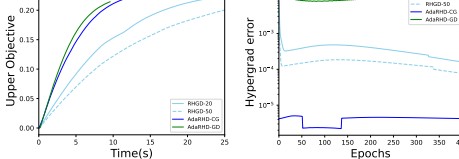

Figure 1: Performances of methods in $n = 100$.  Figure 2: Performances of methods in $n = 1000$.

Figures 1 and 2 show the evolution of the upper-level objective function (Upper Objective) over time and the associated hypergradient estimation errors (Hypergrad error) across outer iterations.

Figure 1 corresponds to $n = 100$, while Figure 2 represents the $n = 1000$ case. These results demonstrate that our algorithm exhibits faster convergence and superior performance compared to RHGD while maintaining scalability with increasing sample dimensionality, confirming its efficiency and robustness. Moreover, although GD achieves quicker initial convergence, CG demonstrates greater robustness, as evidenced by its lower hypergradient estimation errors.

## 4.2 Robustness analysis

In the second experiment, also from [33, Section 4.2], we address the deep hyper-representation problem for classification, which is a subclass of hyper-representation problems [26, 73, 95, 59, 74]. In contrast to the 2-layer SPD network employed for ETH-80 dataset classification in [33], to demonstrate the efficacy of our algorithm, here we utilize a 3-layer SPD network [40] as the upper-level architecture to optimize input embeddings over the larger AFEW dataset [20], comprising seven emotion classes. The training set contains 1,747 matrices with imbalanced class distribution (267, 235, 173, 292, 342, 288, and 150 samples per class, respectively). This optimization problem is formulated as follows:

$$\min_{\mathbf{A}_1, \mathbf{A}_2, \mathbf{A}_3} \quad -\sum_{i \in \mathcal{D}_{\text{val}}} \frac{\mathbf{y}_i^\top \log(\text{SPDnet}(\mathbf{D}_i; \mathbf{A}_1, \mathbf{A}_2, \mathbf{A}_3)\beta^*(\mathbf{A}_1, \mathbf{A}_2, \mathbf{A}_3))}{|\mathcal{D}_{\text{val}}|},$$

$$\text{s.t.} \quad \beta^*(\mathbf{A}_1, \mathbf{A}_2, \mathbf{A}_3) = \operatorname*{arg\,min}_{\beta \in \mathbb{R}^{r(r+1)/2}} -\sum_{i \in \mathcal{D}_{\text{tr}}} \frac{\mathbf{y}_i^\top \log(\text{SPDnet}(\mathbf{D}_i; \mathbf{A}_1, \mathbf{A}_2, \mathbf{A}_3)\beta))}{|\mathcal{D}_{\text{tr}}|} + \frac{\lambda}{2} \|\beta\|^2,$$

$$\mathbf{A}_1 \in \text{St}(d, d_1), \ \mathbf{A}_2 \in \text{St}(d_1, d_2), \ \mathbf{A}_3 \in \text{St}(d_2, r),$$

where $\text{SPDnet}(\cdot; \mathbf{A}_1, \mathbf{A}_2, \mathbf{A}_3)$ denotes the 3-layer SPD network with layer parameters $\mathbf{A}_1$, $\mathbf{A}_2$, and $\mathbf{A}_3$ [40], the term $\mathbf{y}_i$ represents the one-hot encoded label, and $\mathcal{D} = \{\mathbf{D}_i\}_{i=1}^n$ denote a set of SPD matrices where $\mathbf{D}_i \in \mathbb{S}_{++}^d$. Each matrix $\mathbf{D}_i$ has dimensions of $400 \times 400$, and we set $d_1 = 100$, $d_2 = 20$, and $r = 5$.

In this study, we perform a series of experiments with varying initial step sizes to evaluate the robustness and advantages of our proposed AdaRHD algorithm. To address computational constraints, we utilize a 5% subset of the AFEW dataset [20] rather than the full dataset, ensuring tractable training durations. For AdaRHD, we initialize hyperparameters $a_0$, $b_0$, and $c_0$ to equal values and test performance across the range $\{0.2, 1, 2, 10, 20\}$. For RHGD, we fix $\eta_x$ and $\eta_y$ to equal values and evaluate the set $\{5, 1, 0.5, 0.1, 0.05\}$. To mitigate sampling bias and validate robustness, each algorithm is executed five times with distinct random seeds, each iteration employing a unique 5% data subset. This methodology enables systematic assessment of optimization sensitivity to step sizes and establishes the generalizability of our algorithm under practical constraints.

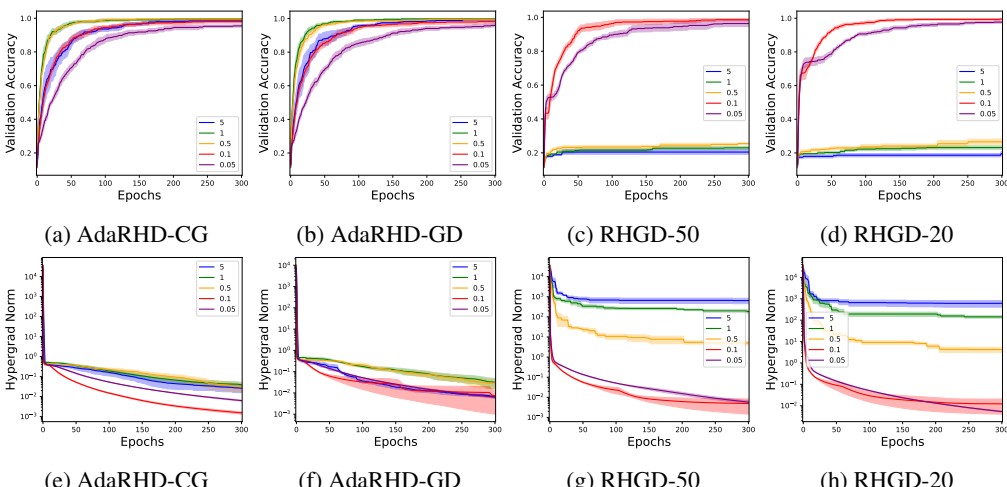

Figure 3: Epoch vs. validation accuracy and ergodic performance $\min_{i \in [0,t]} \|\widehat{\mathcal{G}}F(x_i, y_i^{K_i}, v_i^{N_i})\|_{x_i}^2$ under different initial step sizes for each algorithm. In the figures for "AdaRHD-X", the labels indicate the values of $1/a_0 = 1/b_0 = 1/c_0$; in the figures for "RHGD-X", the labels represent the values of $\eta_x = \eta_y$.

The results are presented in Figure 3 and Table 2. It can be observed that the RHGD-50 and RHGD-20 fail when the initial step sizes are set to 5, 1, or 0.5, whereas the corresponding configurations of

Table 2: Time to reach a specific validation accuracy under different initial step sizes for each algorithm. The values outside the parentheses indicate the mean over five random trials, while the values inside the parentheses represent the standard deviation. "Step Size (A/R)" denotes the initial step sizes used in each method: $1/a_0 = 1/b_0 = 1/c_0$ in the "AdaRHD-X" algorithm; $\eta_x = \eta_y$ in the "RHGD-X" algorithm. "X%" indicates the time required to reach the corresponding validation accuracy.

| Step Size | AdaRHD-CG | | | AdaRHD-GD | | | RHGD-50 | | | RHGD-20 | | |
|---|---|---|---|---|---|---|---|---|---|---|---|---|
| | 50% | 70% | 85% | 50% | 70% | 85% | 50% | 70% | 85% | 50% | 70% | 85% |
| 5.0 | **287.20** | **464.79** | **1217.06** | 432.66 | 690.62 | 1488.26 | / | / | / | / | / | / |
| | **(93.61)** | **(124.76)** | **(635.57)** | (157.76) | (136.52) | (426.67) | / | / | / | / | / | / |
| 1.0 | **131.84** | **191.51** | **245.63** | 200.26 | 317.31 | 421.99 | / | / | / | / | / | / |
| | **(18.76)** | **(34.15)** | **(72.42)** | (31.21) | (68.92) | (88.31) | / | / | / | / | / | / |
| 0.5 | **118.09** | **169.79** | **241.54** | 192.51 | 291.83 | 477.49 | / | / | / | / | / | / |
| | **(19.91)** | **(28.59)** | **(33.09)** | (22.86) | (54.21) | (120.17) | / | / | / | / | / | / |
| 0.1 | 217.32 | 312.58 | **427.84** | 297.90 | 447.60 | 546.18 | 171.97 | 370.69 | 808.48 | **36.01** | **168.47** | 540.46 |
| | (33.96) | **(33.66)** | **(53.40)** | (42.60) | (53.50) | (66.68) | (78.36) | (68.61) | (146.57) | **(12.37)** | (120.86) | (88.54) |
| 0.05 | 251.47 | 415.51 | 633.46 | 358.25 | 504.80 | 648.86 | 137.88 | 715.91 | 1510.09 | **41.36** | **179.11** | 1067.92 |
| | (36.27) | (49.75) | (96.54) | (32.89) | **(15.58)** | **(42.91)** | (136.10) | (104.62) | (363.41) | **(6.88)** | (157.76) | (122.98) |

AdaRHD remain relatively stable. Notably, even when using a step size greater than 1 (i.e., $a_0 = b_0 = c_0 = 0.2$), although the performance is less stable compared to other settings, AdaRHD is still able to converge effectively. Table 2 also shows that AdaRHD-CG consistently achieves 85% validation accuracy in the shortest amount of time. On the other hand, step sizes of $a_0 = b_0 = c_0 = 1$ or 2 yield the best performance among all initializations for the AdaRHD variants, while the corresponding RHGD configurations fail to converge. These results further demonstrate the robustness of the proposed AdaRHD method, which significantly reduces the sensitivity to the choice of initial step size and greatly improves training efficiency. Moreover, although RHGD-20 requires the shortest time to reach 50% and 70% validation accuracies, it demands more time than our method to achieve 85% validation accuracy, demonstrating the efficiency of our approach. Meanwhile, the lower standard deviation exhibited by our method further highlights the robustness of our algorithm.

## 5 Conclusion

This paper proposes an adaptive algorithm for solving Riemannian bilevel optimization (RBO) problems, employing two strategies: gradient descent and conjugate gradient, to compute approximate Riemannian hypergradients. To our knowledge, this is the first fully adaptive RBO algorithm, incorporating a step size mechanism, eliminating prior knowledge requirements of problem parameters. We establish that the method achieves an $\mathcal{O}(1/\epsilon)$ iteration complexity to reach an $\epsilon$-stationary point, matching the complexity of the standard non-adaptive algorithms. Additionally, we show that substituting the exponential mapping with a computationally efficient retraction mapping maintains this complexity guarantee.

Notably, this work focuses exclusively on developing an adaptive double-loop algorithm for deterministic Riemannian bilevel optimization (RBO) problems when the lower-level objective is geodesically strongly convex. Future research directions include: (1) designing single-loop adaptive algorithms [93]; (2) extending the framework to stochastic settings [33, 23, 54] via adaptive step sizes such as inverse cumulative stochastic (hyper)gradient norms [92] or other strategies [22, 47, 86, 58, 64, 96, 70]; (3) addressing geodesically convex (non-strongly convex) lower-level objectives through regularization [2] or Polyak-Łojasiewicz (PŁ) conditions [15]; and (4) investigating alternative adaptive step size strategies [61, 96, 37, 70, 52, 76] to resolve the $1/\epsilon$-gap in gradient ($G_g$) and Hessian-vector product ($HV_g$) complexities compared to non-adaptive Riemannian bilevel methods [33, 54].

## Acknowledgements

We sincerely appreciate the reviewers for their invaluable feedback and insightful suggestions. This work is partly supported by the National Key R&D Program of China under grant 2023YFA1009300, National Natural Science Foundation of China under grants 12171100, and the Major Program of NFSC (72394360,72394364).

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

# Appendix

## A  Related works

**Bilevel optimization:** Bilevel optimization is first introduced by [13] and rooted in the Stackelberg game [14, 8, 87, 88]. When the lower-level objective is strongly convex, various methods have been proposed [28, 17, 43, 42, 19, 57, 35, 51]. Hypergradient-based approaches include approximate implicit differentiation (AID) [21, 65]; iterative differentiation (ITD) [60, 25, 73, 29]; Neumann series (NS) [28]; and conjugate gradient (CG) [43]. Recent studies address bilevel problems with constraints on either the upper- or lower-level objective. For example, [16, 35] focus on scenarios where constraints are imposed solely on the upper-level objective, whereas [83, 91, 94] develop methods for problems with lower-level constraints. For a comprehensive overview, we refer readers to [43, 42, 51] and the references therein.

**Riemannian optimization:** Riemannian optimization has attracted considerable attention due to its broad applications, including low-rank matrix completion [11, 85]; phase retrieval [5, 79]; dictionary learning [18, 78]; dimensionality reduction [34, 82, 63]; and manifold regression [56, 55]. Various methods have been developed, such as Riemannian (stochastic) gradient descent [27, 1, 9, 5, 12]; nonlinear conjugate gradients [71]; and variance-reduced stochastic gradients [99, 101, 46, 72, 104]. For further details, see [10] and references therein.

**Riemannian adaptive optimization:** Several adaptive algorithms have been developed to solve Riemannian optimization problems. Notable examples include the Riemannian adaptive stochastic gradient algorithm (RASA) [45], Riemannian AMSGrad (RAMSGrad) [3], modified RAMSGrad [69], constrained RMSProp (cRMSProp) [68], sharpness-aware minimization on Riemannian manifolds (Riemannian SAM) [96], Riemannian natural gradient descent (RNGD) [37], and a framework for Riemannian adaptive optimization [70]. For a comprehensive overview of their application scopes, we refer readers to [70].

**Riemannian bilevel optimization:** Research on Riemannian bilevel optimization remains limited to a few recent studies. [54] investigated hypergradient computation for such problems on Riemannian manifolds and developed deterministic and stochastic algorithms, namely the algorithm for Riemannian bilevel optimization (RieBO) and the algorithm for Riemannian stochastic bilevel optimization (RieSBO), for solving Riemannian bilevel and stochastic bilevel optimization, respectively. In parallel, [33] proposed a framework termed Riemannian hypergradient descent (RHGD), which provides multiple hypergradient estimation strategies, supported by convergence and complexity analyses. The authors also extended their framework to address Riemannian minimax and compositional optimization problems. By leveraging the value-function reformulation and the Lagrangian method, [23] introduced a fully stochastic first-order approach, the Riemannian first-order fast stochastic approximation (RF$^2$SA), which is applicable to scenarios where both objectives are stochastic or deterministic.

**Adaptive bilevel optimization:** The closest related approach to our method is the D-TFBO proposed in [93], which tackles Euclidean bilevel optimization by employing a similar adaptive step size strategy. While some aspects of Euclidean analysis extend directly to Riemannian settings, geometric curvature introduces distortions that create unique analytical challenges. We derive convergence guarantees similar to those of D-TFBO and enhance our algorithm's efficiency by replacing exponential mapping with retraction mapping, thereby reducing computational overhead. To our knowledge, this work presents the first fully adaptive method with theoretical convergence guarantees for solving Riemannian bilevel optimization problems.

## B  Additional preliminaries for Section 2

The Lipschitzness of the functions and operators in the Riemannian manifolds is defined as follows:

**Definition B.1.** *[33, Definition 1] For any $x, x_1, x_2 \in \mathcal{U}_x, y, y_1, y_2 \in \mathcal{U}_y$, where $\mathcal{U}_x \times \mathcal{U}_y \subseteq \mathcal{M}_x \times \mathcal{M}_y$,*

*(i) a function $f : \mathcal{M}_x \to \mathbb{R}$ is said to have L-Lipschitz Riemannian gradient in $\mathcal{U}_x$ if $\|\mathcal{P}_{x_1}^{x_2}\mathcal{G}f(x_1) - \mathcal{G}f(x_2)\|_{x_2} \leq Ld(x_1, x_2)$.*

*(ii)* *a bi-function $f : \mathcal{M}_x \times \mathcal{M}_y \to \mathbb{R}$ is said to have L-Lipschitz Riemannian gradient in $\mathcal{U}_x \times \mathcal{U}_y$ if $\|\mathcal{P}_{y_1}^{y_2}\mathcal{G}_y f(x, y_1) - \mathcal{G}_y f(x, y_2)\|_{y_2} \leq Ld(y_1, y_2)$, $\|\mathcal{G}_x f(x, y_1) - \mathcal{G}_x f(x, y_2)\|_x \leq Ld(y_1, y_2)$, $\|\mathcal{P}_{x_1}^{x_2}\mathcal{G}_x f(x_1, y) - \mathcal{G}_x f(x_2, y)\|_{x_2} \leq Ld(x_1, x_2)$ and $\|\mathcal{G}_y f(x_1, y) - \mathcal{G}_y f(x_2, y)\|_y \leq Ld(x_1, x_2)$.*

*(iii)* *a linear operator $\mathcal{G}(x, y) : T_y\mathcal{M}_y \to T_x\mathcal{M}_x$ (e.g. $\mathcal{G}_{xy}g(x, y)$), is said to be $\rho$-Lipschitz in $\mathcal{U}_x \times \mathcal{U}_y$ if $\|\mathcal{P}_{x_1}^{x_2}\mathcal{G}(x_1, y) - \mathcal{G}(x_2, y)\|_{x_2} \leq \rho\, d(x_1, x_2)$ and $\|\mathcal{G}(x, y_1) - \mathcal{G}(x, y_2)\mathcal{P}_{y_1}^{y_2}\|_x \leq \rho\, d(y_1, y_2)$.*

*(iv)* *a linear operator $\mathcal{H}(x, y) : T_y\mathcal{M}_y \to T_y\mathcal{M}_y$ (e.g. $\mathcal{H}_y g(x, y)$), is said to be $\rho$-Lipschitz in $\mathcal{U}_x \times \mathcal{U}_y$ if $\|\mathcal{P}_{y_1}^{y_2}\mathcal{H}(x, y_1)\mathcal{P}_{y_2}^{y_1} - \mathcal{H}(x, y_2)\|_{y_2} \leq \rho\, d(y_1, y_2)$ and $\|\mathcal{H}(x_1, y) - \mathcal{H}(x_2, y)\|_y \leq \rho\, d(x_1, x_2)$.*

Due to the curvature of Riemannian manifolds, the notion of distance differs from that in Euclidean space. Accordingly, we have the following trigonometric distance bound on Riemannian manifolds.

**Lemma B.1** (Trigonometric distance bound [100, 99, 32]). *Let $x_a, x_b, x_c \in \mathcal{U} \subseteq \mathcal{M}$ and denote $a = d(x_b, x_c)$, $b = d(x_a, x_c)$ and $c = d(x_a, x_b)$ as the geodesic side lengths. Then, it holds that*

$$a^2 \leq \zeta(\tau, c)b^2 + c^2 - 2\left\langle \mathrm{Exp}_{x_a}^{-1}(x_b), \mathrm{Exp}_{x_a}^{-1}(x_c)\right\rangle_{x_a}$$

*where $\zeta(\tau, c) = \frac{\sqrt{|\tau|}c}{\tanh(\sqrt{|\tau|}c)}$ if $\tau < 0$ and $\zeta(\tau, c) = 1$ if $\tau \geq 0$, and $\tau$ denotes the lower bound of the sectional curvature of $\mathcal{U}$ [66, Section 3.1.3].*

## C   Tangent space conjugate gradient algorithm

In this section, we introduce the tangent-space conjugate gradient method [81, 10], which is used in Section 3.1.

---

**Algorithm 2** Tangent Space Conjugate Gradient $v^n = \mathrm{TSCG}(\mathcal{H}_y g(x, \hat{y}), \mathcal{G}_y f(x, \hat{y}), v^0, \epsilon_v)$

---

1: Let $r^0 = \mathcal{G}_y f(x, \hat{y}) \in T_y\mathcal{M}_y, p^0 = r^0$.
2: **while** $\|\mathcal{H}_y g(x, \hat{y})[v^n] - \mathcal{G}_y f(x, \hat{y})\|_{\hat{y}}^2 > \epsilon_v$ **do**
3:     Compute $\mathcal{H}_y g(x, \hat{y})[p^n]$.
4:     $a^{n+1} = \frac{\|r^n\|_{\hat{y}}^2}{\langle p^n, \mathcal{H}_y g(x, \hat{y})[p^n]\rangle_{\hat{y}}}$,
5:     $v^{n+1} = v^n + a^{n+1}p^n$,
6:     $r^{n+1} = r^n - a^{n+1}\mathcal{H}_y g(x, \hat{y})[p^n]$,
7:     $b^{n+1} = \frac{\|r^{n+1}\|_{\hat{y}}^2}{\|r^n\|_{\hat{y}}^2}$,
8:     $p^{n+1} = r^{n+1} + b^{n+1}p^n$,
9:     $n = n + 1$.
10: **end while**

---

## D   Additional results of Section 3.4

Since the retraction mapping only affects the total number of iterations required for the lower-level problem in Problem (1) to converge, without influencing the error bounds, the hypergradient approximation error bound (Lemma 3.1) remains consistent with that in Algorithm 1. Moreover, analogous to Proposition 3.2, we establish the following upper bounds on the total number of iterations required to solve the lower-level problem in Problem (1) and the linear system (4).

**Proposition D.1.** *Suppose that Assumptions 3.1, 3.2, 3.3, 3.4, 3.5, and 3.6 hold. Then, for any $0 \leq t \leq T$, the iterations $K_t$ and $N_t$ required in Algorithm 3 satisfy:*

$$K_t \leq \frac{\log(C_b^2/b_0^2)}{\log(1 + \epsilon_y/C_b^2)} + \frac{1}{(\mu/\bar{b} - \bar{\zeta}L_g^2/\bar{b}^2)}\log\left(\frac{\tilde{b}}{\epsilon_y}\right).$$

**Algorithm 3** AdaRHD with Retraction (AdaRHD-R)

---

1: Initial points $x_0 \in \mathcal{M}_x, y_0 \in \mathcal{M}_y$, and $v_0 \in T_{y_0}\mathcal{M}$, initial step sizes $a_0 > 0$, $b_0 > 0$, and $c_0 > 0$, total iterations $T$, and error tolerances $\epsilon_y = \epsilon_v = \frac{1}{T}$.
2: **for** $t = 0, 1, 2, ..., T-1$ **do**
3:      Set $k = 0$ and $y_t^0 = y_{t-1}^{K_{t-1}}$ if $t > 0$ and $y_0$ otherwise.
4:      **while** $\|\mathcal{G}_y g(x_t, y_t^k)\|_{y_t^k}^2 > \epsilon_y$ **do**
5:          $b_{k+1}^2 = b_k^2 + \|\mathcal{G}_y g(x_t, y_t^k)\|_{y_t^k}^2$,
6:          $y_t^{k+1} = \mathrm{Retr}_{y_t^k}(-\frac{1}{b_{k+1}}\mathcal{G}_y g(x_t, y_t^k))$,
7:          $k = k + 1$.
8:      **end while**
9:      $K_t = k$.
10:      Set $n = 0$ and $v_t^0 = \mathcal{P}_{y_{t-1}^{K_{t-1}}}^{y_t^{K_t}} v_{t-1}^{N_{t-1}}$ if $t > 0$ and $v_0$ otherwise.
11:      **while** $\|\nabla_v R(x_t, y_t^{K_t}, v_t^n)\|_{y_t^{K_t}}^2 > \epsilon_v$ **do**
12:          $c_{n+1}^2 = c_n^2 + \|\nabla_v R(x_t, y_t^{K_t}, v_t^n)\|_{y_t^{K_t}}^2$,
13:          $v_t^{n+1} = v_t^n - \frac{1}{c_{n+1}}\nabla_v R(x_t, y_t^{K_t}, v_t^n)$,                ▷ Gradient descent
14:          $n = n + 1$.
15:      **end while**
16:      **Or** set $v_t^0 = 0$ and invoke $v_t^n = \mathrm{TSCG}(\mathcal{H}_y g(x_t, y_t^{K_t}), \mathcal{G}_y f(x_t, y_t^{K_t}), v_t^0, \epsilon_v)$.    ▷ Conjugate gradient
17:      $N_t = n$.
18:      $\widehat{\mathcal{G}}F(x_t, y_t^{K_t}, v_t^{N_t}) = \mathcal{G}_x f(x_t, y_t^{K_t}) - \mathcal{G}_{xy}^2 g(x_t, y_t^{K_t})[v_t^{N_t}]$,
19:      $a_{t+1}^2 = a_t^2 + \|\widehat{\mathcal{G}}F(x_t, y_t^{K_t}, v_t^{N_t})\|_{x_t}^2$,
20:      $x_{t+1} = \mathrm{Retr}_{x_t}(-\frac{1}{a_{t+1}}\widehat{\mathcal{G}}F(x_t, y_t^{K_t}, v_t^{N_t}))$.
21: **end for**

---

*AdaRHD-GD:*

$$N_t \leq \frac{\log(C_c^2/c_0^2)}{\log(1 + \epsilon_v/C_c^2)} + \frac{\bar{c}}{\mu}\log\left(\frac{\tilde{c}}{\epsilon_v}\right),$$

where $C_b, C_c, \bar{b}, \bar{c}, \tilde{b}, \tilde{c}$, and $\bar{\zeta}$ are constants defined in Appendix H.

*AdaRHD-CG:*

$$N_t \leq \frac{1}{2}\log\left(\frac{4L_g^3 l_f^2}{\mu^3 \epsilon_v}\right) / \log\left(\frac{\sqrt{L_g/\mu} + 1}{\sqrt{L_g/\mu} - 1}\right).$$

## E    Extension to Riemannian min-max Problems

Riemannian min-max problems [44, 39, 31, 102, 89, 30, 62, 38] have recently attracted increasing attention. The general form of such problems is

$$\min_{x \in \mathcal{M}_x} \max_{y \in \mathcal{M}_y} f(x, y), \tag{6}$$

which can be regarded as a special case of the Riemannian bilevel optimization problem (1) with $g(x, y) = -f(x, y)$.

If $g$ is geodesically strongly convex w.r.t. $y$ over $\mathcal{M}_y$, the Riemannian hypergradient (3) reduces to $\mathcal{G}F(x) = \mathcal{G}_x f(x, y^*(x))$, and the approximate Riemannian hypergradient becomes $\widehat{\mathcal{G}}F(x, \hat{y}) = \mathcal{G}_x f(x, \hat{y})$. The pseudocode of the adaptive method for solving Problem (6) is summarized in Algorithm 4.

Before establishing the convergence result of Algorithm 4, we present the required assumptions.

**Assumption E.1.** *The following conditions hold:*

**Algorithm 4** AdaRHD for Riemannian Min-Max Optimization

---

1: Initial points $x_0 \in \mathcal{M}_x$ and $y_0 \in \mathcal{M}_y$, initial step sizes $a_0 > 0$, and $b_0 > 0$, total iterations $T$, and error tolerance $\epsilon_y = \frac{1}{T}$.
2: **for** $t = 0, 1, 2, \ldots, T - 1$ **do**
3:      Set $k = 0$ and initialize $y_t^0 = y_{t-1}^{K_{t-1}}$ if $t > 0$, otherwise $y_t^0 = y_0$.
4:      **while** $\|\mathcal{G}_y g(x_t, y_t^k)\|^2 > \epsilon_y$ **do**
5:          Update $b_{k+1}^2 = b_k^2 + \|\mathcal{G}_y g(x_t, y_t^k)\|_{y_t^k}^2$.
6:          $y_t^{k+1} = \mathrm{Exp}_{y_t^k}\left(-\frac{1}{b_{k+1}}\mathcal{G}_y g(x_t, y_t^k)\right)$,
7:          **or** $y_t^{k+1} = \mathrm{Retr}_{y_t^k}\left(-\frac{1}{b_{k+1}}\mathcal{G}_y g(x_t, y_t^k)\right)$.          ▷ retraction step
8:          $k \leftarrow k + 1$.
9:      **end while**
10:     $K_t \leftarrow k$.
11:     Compute $\widehat{\mathcal{G}}F(x_t, y_t^{K_t}) = \mathcal{G}_x f(x_t, y_t^{K_t})$.
12:     Update $a_{t+1}^2 = a_t^2 + \|\widehat{\mathcal{G}}F(x_t, y_t^{K_t})\|_{x_t}^2$.
13:     $x_{t+1} = \mathrm{Exp}_{x_t}\left(-\frac{1}{a_{t+1}}\widehat{\mathcal{G}}F(x_t, y_t^{K_t})\right)$,
14:     **or** $x_{t+1} = \mathrm{Retr}_{x_t}\left(-\frac{1}{a_{t+1}}\widehat{\mathcal{G}}F(x_t, y_t^{K_t})\right)$.          ▷ retraction step
15: **end for**

---

*(i) Assumptions 3.1 and 3.4 are satisfied (Assumptions 3.5 and 3.6 are required when retraction is used);*

*(ii) $f(x, y)$ is continuously differentiable, and $-f(x, y)$ is geodesically strongly convex w.r.t. $y$;*

*(iii) $f(x, y)$ is $l_f$-Lipschitz continuous, and its Riemannian gradients $\mathcal{G}_x f(x, y)$ and $\mathcal{G}_y f(x, y)$ are $L_f$-Lipschitz continuous.*

We now state the convergence guarantee of Algorithm 4.

**Theorem E.1.** *Suppose that Assumption E.1 holds. Then, the sequence $\{x_t\}_{t=0}^T$ generated by Algorithm 4 satisfies*

$$\frac{1}{T}\sum_{t=0}^{T-1} \|\mathcal{G}F(x_t)\|_{x_t}^2 \leq \frac{C_{mm}}{T} = \mathcal{O}\left(\frac{1}{T}\right),$$

*where $C_{mm}$ is a constant depending on the curvature, Lipschitz, and strong convexity parameters.*

*Moreover, the overall gradient complexity of Algorithm 4 is $\mathcal{O}(1/\epsilon^2)$.*

# F    Preliminary results for proofs

**Lemma F.1.** *[90, Lemma 3.2] For any non-negative $\alpha_1, ..., \alpha_T$, and $\alpha_1 \geq 1$, we have*

$$\sum_{l=1}^T \frac{\alpha_l}{\sum_{i=1}^l \alpha_i} \leq \log\left(\sum_{l=1}^T \alpha_l\right) + 1.$$

**Lemma F.2.** *Suppose that Assumptions 3.1, 3.2, and 3.3 hold. Then, the following statements hold,*

*(1) For $x_1, x_2 \in \mathcal{M}_x$, it holds that*

$$d(y^*(x_1), y^*(x_2)) \leq L_y d(x_1, x_2),$$

*where $L_y := \frac{L_g}{\mu}$ and $y^*(x)$ is the optimal solution of the lower-level problem in Problem (1);*

*(2) The Riemannian hypergradient $\mathcal{G}F(x)$ satsifies $\|\mathcal{G}F(x)\|_x \leq l_f(1 + L_g/\mu)$, and for $x_1, x_2 \in \mathcal{M}_x$, it holds that*

$$\|\mathcal{G}F(x_1) - \mathcal{P}_{x_2}^{x_1}\mathcal{G}F(x_2)\|_{x_1} \leq L_F d(x_1, x_2),$$

*where $L_F := \left(L_f + \frac{\rho l_f}{\mu}\right)\left(1 + \frac{L_g}{\mu}\right)^2$;*

*Proof.* The proofs of Lemma F.2 can be derived from [28, Lemma 2.2]; here we provide them for completeness.

(1): For $x_1, x_2 \in \mathcal{M}_x$, there exist two geodesics $c_1 : [0, 1] \to \mathcal{M}_y$ and $c_2 : [0, 1] \to \mathcal{M}_x$ such that $c_1(t) = y^*(c_2(t))$ for $t \in [0, 1]$, and $c_2(0) = x_1, c_2(1) = x_2$. Then, we have

$$d(y^*(x_1), y^*(x_2)) = \int_0^1 \|c_1'(t)\|_{c_1(t)} dt = \int_0^1 \|\mathrm{D}y^*(c_2(t))[c_2'(t)]\|_{c_2(t)} dt \leq \frac{L_g}{\mu} \int_0^1 \|c_2'(t)\|_{c_2(t)} dt = \frac{L_g}{\mu} d(x_1, x_2),$$

where the first inequality follows from $\mathrm{D}y^*(x) := -\mathcal{H}_y^{-1} g(x, y^*(x)) \mathcal{G}_{yx}^2 g(x, y^*(x))$.

(2): By the definition of $\mathcal{G}F(x)$, from Assumptions 3.2 and 3.3, it holds that

$$\|\mathcal{G}F(x)\|_x = \|\mathcal{G}_x f(x, y^*(x)) - \mathcal{G}_{xy}^2 g(x, y^*(x))[\mathcal{H}_y^{-1} g(x, y^*(x))[\mathcal{G}_y f(x, y^*(x))]]\|_x$$

$$= \|\mathcal{G}_x f(x, y^*(x))\|_x + \|\mathcal{G}_{xy}^2 g(x, y^*(x))\|_{y^*(x)} \|\mathcal{H}_y^{-1} g(x, y^*(x))\|_{y^*(x)} \|\mathcal{G}_y f(x, y^*(x))\|_{y^*(x)} \leq l_f + \frac{L_g}{\mu} l_f.$$

Furthermore, we have

$$\|\mathcal{P}_{x_1}^{x_2} \mathcal{G}F(x_1) - \mathcal{G}F(x_2)\|_{x_2}$$
$$\leq \left\| \mathcal{P}_{x_1}^{x_2} \mathcal{G}_x f(x_1, y^*(x_1)) - \mathcal{G}_x f(x_2, y^*(x_2)) \right\|_{x_2}$$
$$+ \left\| \mathcal{P}_{x_1}^{x_2} \mathcal{G}_{xy}^2 g(x_1, y^*(x_1)) - \mathcal{G}_{xy}^2 g(x_2, y^*(x_2)) \mathcal{P}_{y^*(x_1)}^{y^*(x_2)} \right\|_{x_2} \left\| \mathcal{H}_y^{-1} g(x_1, y^*(x_1)) \right\|_{y^*(x_1)} \left\| \mathcal{G}_y f(x_1, y^*(x_1)) \right\|_{y^*(x_1)}$$
$$+ \|\mathcal{G}_{xy}^2 g(x_2, y^*(x_2))\|_{x_2} \left\| \mathcal{P}_{y^*(x_1)}^{y^*(x_2)} \mathcal{H}_y^{-1} g(x_1, y^*(x_1)) \mathcal{P}_{y^*(x_2)}^{y^*(x_1)} - \mathcal{H}_y^{-1} g(x_2, y^*(x_2)) \right\|_{y^*(x_2)} \left\| \mathcal{G}_y f(x_1, y^*(x_1)) \right\|_{y^*(x_1)}$$
$$+ \|\mathcal{G}_{xy}^2 g(x_2, y^*(x_2))\|_{x_2} \|\mathcal{H}_y^{-1} g(x_2, y^*(x_2))\|_{y^*(x_2)} \left\| \mathcal{P}_{y^*(x_1)}^{y^*(x_2)} \mathcal{G}_y f(x_1, y^*(x_1)) - \mathcal{G}_y f(x_2, y^*(x_2)) \right\|_{y^*(x_2)}.$$
$$\tag{7}$$

By Assumptions 3.2 and 3.3, for the first term of (7), we have

$$\|\mathcal{P}_{x_1}^{x_2} \mathcal{G}_x f(x_1, y^*(x_1)) - \mathcal{G}_x f(x_2, y^*(x_2))\|_{x_2}$$
$$\leq \|\mathcal{G}_x f(x_1, y^*(x_1)) - \mathcal{G}_x f(x_1, y^*(x_2))\|_{x_1} + \|\mathcal{P}_{x_1}^{x_2} \mathcal{G}_x f(x_1, y^*(x_2)) - \mathcal{G}_x f(x_2, y^*(x_2))\|_{x_2}$$
$$\leq L_f d(y^*(x_1), y^*(x_2)) + L_f d(x_1, x_2) = \left( \frac{L_f L_g}{\mu} + L_f \right) d(x_1, x_2).$$

We then consider the fourth term of (7), we have

$$\left\| \mathcal{P}_{y^*(x_1)}^{y^*(x_2)} \mathcal{G}_y f(x_1, y^*(x_1)) - \mathcal{G}_y f(x_2, y^*(x_2)) \right\|_{y^*(x_2)}$$
$$\leq \left\| \mathcal{P}_{y^*(x_1)}^{y^*(x_2)} \mathcal{G}_y f(x_1, y^*(x_1)) - \mathcal{G}_y f(x_1, y^*(x_2)) \right\|_{y^*(x_2)} + \|\mathcal{G}_y f(x_1, y^*(x_2)) - \mathcal{G}_y f(x_2, y^*(x_2))\|_{y^*(x_2)}$$
$$\leq \left( \frac{L_f L_g}{\mu} + L_f \right) d(x_1, x_2).$$

For the second term of (7), we have

$$\|\mathcal{P}_{x_1}^{x_2} \mathcal{G}_{xy}^2 g(x_1, y^*(x_1)) - \mathcal{G}_{xy}^2 g(x_2, y^*(x_2)) \mathcal{P}_{y^*(x_1)}^{y^*(x_2)}\|_{x_2}$$
$$\leq \|\mathcal{P}_{x_1}^{x_2} \mathcal{G}_{xy}^2 g(x_1, y^*(x_1)) \mathcal{P}_{y^*(x_2)}^{y^*(x_1)} - \mathcal{P}_{x_1}^{x_2} \mathcal{G}_{xy}^2 g(x_1, y^*(x_2))\|_{x_2} + \|\mathcal{P}_{x_1}^{x_2} \mathcal{G}_{xy}^2 g(x_1, y^*(x_2)) - \mathcal{G}_{xy}^2 g(x_2, y^*(x_2))\|_{x_2}$$
$$\leq \left( \frac{\rho L_g}{\mu} + \rho \right) d(x_1, x_2).$$

For the third term of (7), we have

$$\left\| \mathcal{P}_{y^*(x_1)}^{y^*(x_2)} \mathcal{H}_y^{-1} g(x_1, y^*(x_1)) \mathcal{P}_{y^*(x_2)}^{y^*(x_1)} - \mathcal{H}_y^{-1} g(x_2, y^*(x_2)) \right\|_{y^*(x_2)}$$
$$\leq \left\| \mathcal{H}_y^{-1} g(x_1, y^*(x_1)) - \mathcal{H}_y^{-1} g(x_2, y^*(x_1)) \right\|_{y^*(x_1)} + \left\| \mathcal{P}_{y^*(x_1)}^{y^*(x_2)} \mathcal{H}_y^{-1} g(x_2, y^*(x_1)) \mathcal{P}_{y^*(x_2)}^{y^*(x_1)} - \mathcal{H}_y^{-1} g(x_2, y^*(x_2)) \right\|_{y^*(x_2)}$$
$$\leq \left( \frac{\rho}{\mu^2} + \frac{\rho L_g}{\mu^3} \right) d(x_1, x_2),$$

where the last inequality follows from the operator norm satisfying

$$\|A^{-1} - B^{-1}\| \le \|B^{-1}(B - A)A^{-1}\| \le \|A^{-1}\|\|B^{-1}\|\|A - B\|. \tag{8}$$

Combining the above inequalities, we have

$$\|\mathcal{P}_{x_1}^{x_2}\mathcal{G}F(x_1) - \mathcal{G}F(x_2)\|_{x_2}$$

$$\le \left(\frac{L_f L_g}{\mu} + L_f\right)d(x_1, x_2) + \frac{l_f}{\mu}\left(\frac{\rho L_g}{\mu} + \rho\right)d(x_1, x_2)$$

$$+ L_g l_f\left(\frac{\rho}{\mu^2} + \frac{\rho L_g}{\mu^3}\right)d(x_1, x_2) + \frac{L_g}{\mu}\left(\frac{L_f L_g}{\mu} + L_f\right)d(x_1, x_2)$$

$$= \left(\frac{L_f L_g}{\mu} + L_f + \frac{l_f}{\mu}\left(\frac{\rho L_g}{\mu} + \rho\right) + L_g l_f\left(\frac{\rho}{\mu^2} + \frac{\rho L_g}{\mu^3}\right) + \frac{L_g}{\mu}\left(\frac{L_f L_g}{\mu} + L_f\right)\right)d(x_1, x_2)$$

$$= \left(L_f + \frac{\rho l_f}{\mu}\right)\left(1 + \frac{L_g}{\mu}\right)^2 d(x_1, x_2).$$

The proof is complete. $\qquad\square$

Denote $v^*(x) := \arg\min_{v \in T_{y^*(x)}\mathcal{M}_y} R(x, y^*(x), v)$ and $\hat{v}^*(x, y) := \arg\min_{v \in T_y\mathcal{M}_y} R(x, y, v)$.

**Lemma F.3.** *Suppose that Assumptions 3.1, 3.2, and 3.3 hold. Then, the following statements hold,*

*(1) The objective $R(x, y, v)$ is $\mu$-strongly convex and $L_g$-smooth w.r.t. $v$;*

*(2) $v^*(x)$ satisfies $\|v^*(x)\|_{y^*(x)} \le \frac{l_f}{\mu}$, and $\hat{v}^*(x, y)$ also satisfies $\|\hat{v}^*(x, y)\|_y \le \frac{l_f}{\mu}$;*

*(3) For $x, x_1, x_2 \in \mathcal{M}_x$ and $y_1, y_2 \in \mathcal{M}_y$, it holds that*

$$\left\|v^*(x_1) - \mathcal{P}_{y^*(x_2)}^{y^*(x_1)}v^*(x_2)\right\|_{y^*(x_1)} \le L_v d(x_1, x_2), \text{ and } \|\hat{v}^*(x, y_1) - \mathcal{P}_{y_2}^{y_1}\hat{v}^*(x, y_2)\|_{y_1} \le \hat{L}_v d(y_1, y_2),$$

*where $L_v := (\frac{L_f}{\mu} + \frac{l_f \rho}{\mu^2})(1 + L_y)$ and $\hat{L}_v := \frac{L_f}{\mu} + \frac{l_f \rho}{\mu^2}$.*

*Proof.* (1): Since $\nabla_v^2 R(x, y, v) = \mathcal{H}_y g(x, y)$, the strong convexity of $R(x, y, v)$ follows from Assumption 3.2. Then, by Assumption 3.3, for $v_1, v_2 \in T_y\mathcal{M}_y$, we have

$$\|\nabla_v R(x, y, v_1) - \nabla_v R(x, y, v_2)\|_y \le \|\mathcal{H}_y g(x, y)\|_y \|v_1 - v_2\|_y \le L_g\|v_1 - v_2\|_y.$$

(2): For $\hat{v}^*(x, y)$, we have

$$\nabla_v R\left(x, y, \hat{v}^*(x, y)\right) = \mathcal{H}_y g(x, y)[\hat{v}^*(x, y)] - \mathcal{G}_y f(x, y) = 0,$$

which demonstrates that

$$\|\hat{v}^*(x, y)\|_y = \left\|\mathcal{H}_y^{-1}g(x, y)\mathcal{G}_y f(x, y)\right\|_y \le \left\|\mathcal{H}_y^{-1}g(x, y)\right\|_y \cdot \|\mathcal{G}_y f(x, y))\|_y \le \frac{l_f}{\mu}.$$

Since $v^*(x)$ satisfies $v^*(x) = \hat{v}^*(x, y^*(x))$, the desired result follows.

(3): By the definition of $v^*(x)$, we have

$$\left\|v^*(x_1) - \mathcal{P}_{y^*(x_2)}^{y^*(x_1)}v^*(x_2)\right\|_{y^*(x_1)}$$

$$\le \left\|\mathcal{H}_y^{-1}g(x_1, y^*(x_1))[\mathcal{G}_y f(x_1, y^*(x_1))] - \mathcal{P}_{y^*(x_2)}^{y^*(x_1)}\mathcal{H}_y^{-1}g(x_2, y^*(x_2))[\mathcal{G}_y f(x_2, y^*(x_2))]\right\|_{y^*(x_1)}.$$

The remaining proof is similar to that of Lemma F.2 (2), thus, we omit it.

By the definition of $\hat{v}^*(x, y)$, we have

$$\|\hat{v}^*(x, y_1) - \mathcal{P}_{y_2}^{y_1}\hat{v}^*(x, y_2)\|_{y_1} = \left\|\mathcal{H}_y^{-1}g(x, y_1)\mathcal{G}_y f(x, y_1) - \mathcal{P}_{y_2}^{y_1}\mathcal{H}_y^{-1}g(x, y_2)\mathcal{G}_y f(x, y_2)\right\|_{y_1}$$

$$\le \left\|\mathcal{H}_y^{-1}g(x, y_1)\left(\mathcal{G}_y f(x, y_1) - \mathcal{P}_{y_2}^{y_1}\mathcal{G}_y f(x, y_2)\right)\right\|_{y_1} + \left\|\left(\mathcal{H}_y^{-1}g(x, y_1)\mathcal{P}_{y_2}^{y_1} - \mathcal{P}_{y_2}^{y_1}\mathcal{H}_y^{-1}g(x, y_2)\right)\mathcal{G}_y f(x, y_2)\right\|_{y_1}$$

$$\le \frac{L_f}{\mu}d(y_1, y_2) + l_f\frac{\rho}{\mu^2}d(y_1, y_2),$$

where the second inequality follows from (8). $\qquad\square$

**Lemma F.4.** *Suppose that Assumptions 3.1, 3.2, and 3.3 hold. Then, for any $t \geq 0$, we have*

$$d(y_t^{K_t}, y^*(x_t))^2 \leq \frac{\epsilon_y}{\mu^2}, \text{ and } \left\| v_t^{N_t} - \hat{v}^*(x_t, y_t^{K_t}) \right\|_{y_t^{K_t}}^2 \leq \frac{\epsilon_v}{\mu^2}.$$

*Proof.* According to the termination conditions of Algorithm 1 for the lower-level problem and the linear system, we have

$$\|\mathcal{G}_y g(x_t, y_t^{K_t})\|_{y_t^{K_t}}^2 \leq \epsilon_y, \text{ and } \|\nabla_v R(x_t, y_t^{K_t}, v_t^{N_t})\|_{y_t^{K_t}}^2 \leq \epsilon_v.$$

By the strong convexity of $g$ and $R$ (cf. Lemma F.3), we have

$$d^2(y_t^{K_t}, y^*(x_t)) \leq \frac{1}{\mu^2} \left\| \mathcal{G}_y g(x_t, y_t^{K_t}) - \mathcal{P}_{y^*(x_t)}^{y_t^{K_t}} \mathcal{G}_y g(x_t, y^*(x_t)) \right\|_{y_t^{K_t}}^2 \leq \frac{\epsilon_y}{\mu^2}$$

and

$$\left\| v_t^{N_t} - \hat{v}^*(x_t, y_t^{K_t}) \right\|_{y_t^{K_t}}^2 \leq \frac{1}{\mu^2} \left\| \nabla_v R(x_t, y_t^{K_t}, v_t^{N_t}) - \nabla_v R\left(x_t, y_t^{K_t}, \hat{v}^*(x_t, y_t^{K_t})\right) \right\|_{y_t^{K_t}}^2 \leq \frac{\epsilon_v}{\mu^2},$$

where we use the definitions $\|\mathcal{G}_y g(x_t, y^*(x_t))\|^2 = 0$ and $\|\nabla_v R\left(x_t, y_t^{K_t}, \hat{v}^*(x_t, y_t^{K_t})\right)\|^2 = 0$. $\square$

**Lemma F.5.** *Suppose that Assumptions 3.1, 3.2, and 3.3 hold. Then, for any $t \geq 0$, we have*

$$\|\widehat{\mathcal{G}}F(x_t, y_t^{K_t}, v_t^{N_t})\|^2 \leq C_f^2,$$

*where $C_f := \left( \frac{2L_g^2 \epsilon_v}{\mu^2} + \frac{4L_g^2 l_f^2}{\mu^2} + 4l_f^2 \right)^{\frac{1}{2}}$.*

*Proof.* By the definition of $\widehat{\mathcal{G}}F$, we have

$$\|\widehat{\mathcal{G}}F(x_t, y_t^{K_t}, v_t^{N_t})\|_{x_t}^2$$

$$\leq 2 \left\| \widehat{\mathcal{G}}F(x_t, y_t^{K_t}, v_t^{N_t}) - \widehat{\mathcal{G}}F\left(x_t, y_t^{K_t}, \hat{v}^*(x_t, y_t^{K_t})\right) \right\|_{x_t}^2 + 2 \left\| \widehat{\mathcal{G}}F\left(x_t, y_t^{K_t}, \hat{v}^*(x_t, y_t^{K_t})\right) \right\|_{x_t}^2$$

$$= 2 \left\| \mathcal{G}_{xy}^2 g(x_t, y_t^{K_t}) \left(v_t^{N_t} - \hat{v}^*(x_t, y_t^{K_t})\right) \right\|_{x_t}^2 + 2\|\mathcal{G}_x f(x_t, y_t^{K_t}) - \mathcal{G}_{xy}^2 g(x_t, y_t^{K_t})\hat{v}^*(x_t, y_t^{K_t})\|_{x_t}^2$$

$$\leq 2 \left\| \mathcal{G}_{xy}^2 g(x_t, y_t^{K_t}) \right\|_{x_t}^2 \|v_t^{N_t} - \hat{v}^*(x_t, y_t^{K_t})\|_{y_t^{K_t}}^2 + 4\|\mathcal{G}_x f(x_t, y_t^{K_t})\|_{x_t}^2 + 4\|\mathcal{G}_{xy}^2 g(x_t, y_t^{K_t})\hat{v}^*(x_t, y_t^{K_t})\|_{x_t}^2$$

$$\leq \frac{2L_g^2 \epsilon_v}{\mu^2} + \frac{4L_g^2 l_f^2}{\mu^2} + 4l_f^2,$$

where the last inequality follows from Assumptions 3.1, 3.2, 3.3, Lemmas F.3 and F.4. $\square$

## G   Proofs of Section 3.3

### G.1   Proof of Lemma 3.1

*Proof.* By the definitions of $v_t^*(x_t)$ and $\hat{v}_t^*(x_t, y_t^{K_t})$, we have

$$\|\widehat{\mathcal{G}}F(x_t, y_t^{K_t}, v_t^{N_t}) - \mathcal{G}F(x_t)\|_{x_t}$$

$$\leq \|\mathcal{G}_x f(x_t, y_t^{K_t}) - \mathcal{G}_x f(x_t, y^*(x_t))\|_{x_t} + \|\mathcal{G}_{xy}^2 g(x_t, y_t^{K_t})v_t^{N_t} - \mathcal{G}_{xy}^2 g(x_t, y^*(x_t))v_t^*(x_t)\|_{x_t}$$

$$\leq L_f d(y_t^{K_t}, y^*(x_t)) + \|\mathcal{G}_{xy}^2 g(x_t, y_t^{K_t})\|_{x_t}\|v_t^{N_t} - \mathcal{P}_{y^*(x_t)}^{y_t^{K_t}} v_t^*(x_t)\|_{y_t^{K_t}}$$

$$\quad + \|\mathcal{G}_{xy}^2 g(x_t, y_t^{K_t}) - \mathcal{G}_{xy}^2 g(x_t, y^*(x_t))\mathcal{P}_{y_t^{K_t}}^{y^*(x_t)}\|_{x_t}\|v_t^*(x_t)\|_{y^*(x_t)}$$

$$\leq L_f d(y_t^{K_t}, y^*(x_t)) + L_g\|v_t^{N_t} - \mathcal{P}_{y^*(x_t)}^{y_t^{K_t}} v_t^*(x_t)\|_{y_t^{K_t}} + \rho\|v_t^*(x_t)\|_{y^*(x_t)} d(y_t^{K_t}, y^*(x_t))$$

$$\leq \left( L_f + \frac{\rho l_f}{\mu} \right) d(y_t^{K_t}, y^*(x_t)) + L_g\|v_t^{N_t} - \mathcal{P}_{y^*(x_t)}^{y_t^{K_t}} v_t^*(x_t)\|_{y_t^{K_t}}, \tag{9}$$

where the third and fourth inequalities follow from Assumption 3.3 and Lemma F.3 (2), respectively.

For $\|v_t^{N_t} - \mathcal{P}_{y^*(x_t)}^{y_t^{K_t}} v_t^*(x_t)\|_{y_t^{K_t}}$, we have

$$\|v_t^{N_t} - \mathcal{P}_{y_t^{K_t}}^{y^*(x_t)} v_t^*(x_t)\|_{y_t^{K_t}} \leq \|v_t^{N_t} - \hat{v}_t^*(x_t, y_t^{K_t})\|_{y_t^{K_t}} + \|\hat{v}_t^*(x_t, y_t^{K_t}) - \mathcal{P}_{y_t^{K_t}}^{y^*(x_t)} \hat{v}_t^*(x_t, y^*(x_t))\|_{y_t^{K_t}}$$

$$\leq \|v_t^{N_t} - \hat{v}_t^*(x_t, y_t^{K_t})\|_{y_t^{K_t}} + \left(\frac{L_f}{\mu} + \frac{l_f \rho}{\mu^2}\right) d(y_t^{K_t}, y^*(x_t)),$$

where the last inequality follows from Lemma F.3 (3) and the fact that $v_t^*(x_t) = \hat{v}_t^*(x_t, y^*(x_t))$.

Combining this result with (9), we obtain

$$\|\widehat{\mathcal{G}}F(x_t, y_t^{K_t}, v_t^{N_t}) - \mathcal{G}F(x_t)\|_{x_t} \leq \left(L_f + \frac{\rho l_f}{\mu}\right) d(y_t^{K_t}, y^*(x_t)) + L_g \|v_t^{N_t} - \mathcal{P}_{y^*(x_t)}^{y_t^{K_t}} v_t^*(x_t)\|_{y_t^{K_t}}$$

$$\leq \left(L_f + \frac{\rho l_f}{\mu}\right) d(y_t^{K_t}, y^*(x_t)) + L_g \left(\|v_t^{N_t} - \hat{v}_t^*(x_t, y_t^{K_t})\|_{y_t^{K_t}} + \left(\frac{L_f}{\mu} + \frac{l_f \rho}{\mu^2}\right) d(y_t^{K_t}, y^*(x_t))\right)$$

$$= \left(L_f + \frac{l_f \rho}{\mu} + L_g\left(\frac{L_f}{\mu} + \frac{l_f \rho}{\mu^2}\right)\right) d(y_t^{K_t}, y^*(x_t)) + L_g \|v_t^{N_t} - \hat{v}_t^*(x_t, y_t^{K_t})\|_{y_t^{K_t}}.$$

The proof is complete. $\square$

### G.2 Proof of Proposition 3.2

Inspired by Proposition 1 in [93], we first give a result that concerns the step sizes $a_t$, $b_k$, and $c_n$.

**Proposition G.1.** *Suppose that Assumptions 3.1, 3.2, and 3.3 hold. Denote $\{T, K, N\}$ as the iterations of $\{x, y, v\}$. Given any constants $C_a \geq a_0$, $C_b \geq b_0$, $C_c \geq c_0$, then, we have*

*AdaRHD-GD:*

*(1) either $a_t \leq C_a$ for any $t \leq T$, or $\exists t_1 \leq T$ such that $a_{t_1} \leq C_a$, $a_{t_1+1} > C_a$;*

*(2) either $b_k \leq C_b$ for any $k \leq K$, or $\exists k_1 \leq K$ such that $b_{k_1} \leq C_b$, $b_{k_1+1} > C_b$;*

*(3) either $c_n \leq C_c$ for any $n \leq N$, or $\exists n_1 \leq N$ such that $c_{n_1} \leq C_c$, $c_{n_1+1} > C_c$.*

*We then consider the case where we use the conjugate gradient method to solve the linear system. Therefore, we do not need to consider the step size $c_n$.*

*AdaRHD-CG:*

*(1) either $a_t \leq C_a$ for any $t \leq T$, or $\exists t_1 \leq T$ such that $a_{t_1} \leq C_a$, $a_{t_1+1} > C_a$;*

*(2) either $b_k \leq C_b$ for any $k \leq K$, or $\exists k_1 \leq K$ such that $b_{k_1} \leq C_b$, $b_{k_1+1} > C_b$;*

#### G.2.1 Improvement on objective function for one step update

We first define a threshold for $a_t$:

$$C_a := \max\{2L_F, a_0\}. \tag{10}$$

**Lemma G.1.** *Suppose that Assumptions 3.1, 3.2, and 3.3 hold. Then, we have*

$$F(x_{t+1}) \leq F(x_t) - \frac{1}{2a_{t+1}}\|\mathcal{G}F(x_t)\|_{x_t}^2 - \frac{1}{2a_{t+1}}\left(1 - \frac{L_F}{a_{t+1}}\right)\|\widehat{\mathcal{G}}F(x_t, y_t^{K_t}, v_t^{N_t})\|_{x_t}^2 + \frac{\epsilon_{y,v}}{2a_{t+1}}. \tag{11}$$

*Furthermore, if $t_1$ in Proposition G.1 exists, then for any $t \geq t_1$, we have*

$$F(x_{t+1}) \leq F(x_t) - \frac{1}{2a_{t+1}}\|\mathcal{G}F(x_t)\|_{x_t}^2 - \frac{1}{4a_{t+1}}\|\widehat{\mathcal{G}}F(x_t, y_t^{K_t}, v_t^{N_t})\|_{x_t}^2 + \frac{\epsilon_{y,v}}{2a_{t+1}}, \tag{12}$$

*where $\epsilon_{y,v} := \frac{\bar{L}^2}{\mu^2}(\epsilon_y + \epsilon_v)$ and $\bar{L} := \max\{\sqrt{2}L_1, \sqrt{2}L_g\}$ with $L_1$ is defined in Lemma 3.1.*

*Proof.* Combining Lemma F.2 (2) and Proposition 2.1, we have

$$
\begin{aligned}
F(x_{t+1}) \leq & F(x_t) + \left\langle \mathcal{G}F(x_t), \mathrm{Exp}_{x_t}^{-1}(x_{t+1}) \right\rangle_{x_t} + \frac{L_F}{2} d(x_{t+1}, x_t)^2 \\
= & F(x_t) - \frac{1}{a_{t+1}} \left\langle \mathcal{G}F(x_t), \widehat{\mathcal{G}}F\left(x_t, y_t^{K_t}, v_t^{N_t}\right) \right\rangle_{x_t} + \frac{L_F}{2a_{t+1}^2} \left\| \widehat{\mathcal{G}}F\left(x_t, y_t^{K_t}, v_t^{N_t}\right) \right\|_{x_t}^2 \\
= & F(x_t) - \frac{1}{2a_{t+1}} \|\mathcal{G}F(x_t)\|_{x_t}^2 - \frac{1}{2a_{t+1}} \left\| \widehat{\mathcal{G}}F\left(x_t, y_t^{K_t}, v_t^{N_t}\right) \right\|_{x_t}^2 \\
& + \frac{1}{2a_{t+1}} \left\| \mathcal{G}F(x_t) - \widehat{\mathcal{G}}F\left(x_t, y_t^{K_t}, v_t^{N_t}\right) \right\|_{x_t}^2 + \frac{L_F}{2a_{t+1}^2} \left\| \widehat{\mathcal{G}}F\left(x_t, y_t^{K_t}, v_t^{N_t}\right) \right\|_{x_t}^2 \\
= & F(x_t) - \frac{1}{2a_{t+1}} \|\mathcal{G}F(x_t)\|_{x_t}^2 - \frac{1}{2a_{t+1}} \left(1 - \frac{L_F}{a_{t+1}}\right) \|\widehat{\mathcal{G}}F(x_t, y_t^{K_t}, v_t^{N_t})\|_{x_t}^2 \\
& + \frac{1}{2a_{t+1}} \left\| \mathcal{G}F(x_t) - \widehat{\mathcal{G}}F\left(x_t, y_t^{K_t}, v_t^{N_t}\right) \right\|_{x_t}^2 . \tag{13}
\end{aligned}
$$

By Lemma 3.1, it holds that

$$
\begin{aligned}
\left\| \mathcal{G}F(x_t) - \widehat{\mathcal{G}}F\left(x_t, y_t^{K_t}, v_t^{N_t}\right) \right\|_{x_t}^2 \leq & 2L_1^2 d^2(y_t^{K_t}, y^*(x_t)) + 2L_g^2 \left( \|v_t^{N_t} - \mathcal{P}_{y^*(x_t)}^{y_t} \hat{v}^*(x_t, y^*(x_t))\|_{y_t}^2 \right) \\
\leq & \bar{L}^2 \left( d^2(y_t^{K_t}, y^*(x_t)) + \|v_t^{N_t} - \mathcal{P}_{y^*(x_t)}^{y_t} \hat{v}^*(x_t, y^*(x_t))\|_{y_t}^2 \right).
\end{aligned}
$$

Therefore, by Lemma F.4, we have

$$
\left\| \mathcal{G}F(x_t) - \widehat{\mathcal{G}}F\left(x_t, y_t^{K_t}, v_t^{N_t}\right) \right\|^2 \leq \frac{\bar{L}^2}{\mu^2} (\epsilon_y + \epsilon_v) = \epsilon_{y,v}.
$$

This Combine (13) deduces the desired result of (11).

**If $t_1$ in Proposition G.1 exists**, then for $t \geq t_1$, we have $a_{t+1} > C_a \geq 2L_F$. The desired result of (12) follows from (11). The proof is complete. $\qquad\square$

Then, similar to [93, Lemma 8], we have the following upper bound for the step size $a_t$.

**Lemma G.2.** *Suppose that Assumptions 3.1, 3.2, 3.3, and 3.4 hold. If $t_1$ in Proposition G.1 does not exist, we have $a_t \leq C_a$ for all $t \leq T$.*

*If the $t_1$ in Proposition G.1 exists, we have*

$$
\begin{cases}
a_t \leq C_a, & t \leq t_1; \\
a_t \leq C_a + 2F_0 + \dfrac{2t\epsilon_{y,v}}{a_0}, & t > t_1,
\end{cases}
$$

*where*

$$
F_0 := 2\left(F(x_0) - F^*\right) + \frac{L_F C_a^2}{a_0^2}. \tag{14}
$$

*Proof.* **If $t_1$ in Proposition G.1 does not exist**, then for any $t \leq T$, it holds that $a_t \leq C_a$.

**If $t_1$ in Proposition G.1 exists**, then for any $t < t_1$, it holds that $a_{t+1} \leq C_a$. By Lemma G.1, for any $t \geq t_1$, it holds that

$$
F(x_{t+1}) \leq F(x_t) - \frac{1}{2a_{t+1}} \|\mathcal{G}F(x_t)\|_{x_t}^2 - \frac{1}{4a_{t+1}} \|\widehat{\mathcal{G}}F(x_t, y_t^{K_t}, v_t^{N_t})\|_{x_t}^2 + \frac{\epsilon_{y,v}}{2a_{t+1}}.
$$

Removing the nonnegative term $-\frac{1}{2a_{t+1}} \|\mathcal{G}F(x_t)\|_{x_t}^2$, we have

$$
\frac{\|\widehat{\mathcal{G}}F(x_t, y_t^{K_t}, v_t^{N_t})\|_{x_t}^2}{a_{t+1}} \leq 4\left(F(x_t) - F(x_{t+1})\right) + \frac{2\epsilon_{y,v}}{a_{t+1}}. \tag{15}
$$

Summing (15) from $t_1$ to $t$, we have

$$
\sum_{i=t_1}^{t} \frac{\|\widehat{\mathcal{G}}F(x_i, y_i^{K_i}, v_i^{N_i})\|_{x_i}^2}{a_{i+1}} \leq 4\sum_{i=t_1}^{t} \left(F(x_i) - F(x_{i+1})\right) + \sum_{i=t_1}^{i} \frac{2\epsilon_{y,v}}{a_{i+1}} 4\left(F(x_{t_1}) - F(x_{t+1})\right) + \sum_{i=t_1}^{t} \frac{2\epsilon_{y,v}}{a_{t+1}}. \tag{16}
$$

For $F(x_{t_1})$, by (11), we have

$$F(x_{t_1}) \leq F(x_0) + \sum_{i=0}^{t_1-1} \frac{L_F}{2a_{i+1}^2}\|\widehat{\mathcal{G}}F(x_i, y_i^{K_i}, v_i^{N_i})\|_{x_i}^2 + \sum_{i=0}^{t_1-1} \frac{\epsilon_{y,v}}{2a_{i+1}}.$$

This combines with (16), by $F(x_{t+1}) \geq F^*$, we have

$$\sum_{i=t_1}^{t} \frac{\|\widehat{\mathcal{G}}F(x_i, y_i^{K_i}, v_i^{N_i})\|_{x_i}^2}{a_{i+1}} \leq 4\left(F(x_0) - F^*\right) + \sum_{i=0}^{t_1-1} \frac{2L_F}{a_{i+1}^2}\|\widehat{\mathcal{G}}F(x_i, y_i^{K_i}, v_i^{N_i})\|_{x_i}^2 + \sum_{i=0}^{t} \frac{2\epsilon_{y,v}}{a_{i+1}}$$

$$\leq 4\left(F(x_0) - F^*\right) + \frac{2L_F \sum_{i=0}^{t_1-1} \|\widehat{\mathcal{G}}F(x_i, y_i^{K_i}, v_i^{N_i})\|_{x_i}^2}{a_0^2} + \sum_{i=0}^{t} \frac{2\epsilon_{y,v}}{a_{i+1}}$$

$$\leq 4\left(F(x_0) - F^*\right) + \frac{2L_F a_{t_1}^2}{a_0^2} + \frac{2(t+1)\epsilon_{y,v}}{a_0} \leq 4\left(F(x_0) - F^*\right) + \frac{2L_F C_a^2}{a_0^2} + \frac{2(t+1)\epsilon_{y,v}}{a_0}.$$

Since $a_{t+1}^2 = a_t^2 + \|\widehat{\mathcal{G}}F(x_t, y_t^{K_t}, v_t^{N_t})\|_{x_t}^2$, we have

$$a_{t+1} = a_t + \frac{\|\widehat{\mathcal{G}}F(x_t, y_t^{K_t}, v_t^{N_t})\|_{x_t}^2}{a_{t+1} + a_t} \leq a_t + \frac{\|\widehat{\mathcal{G}}F(x_t, y_t^{K_t}, v_t^{N_t})\|_{x_t}^2}{a_{t+1}} \leq a_{t_1} + \sum_{i=t_1}^{t} \frac{\|\widehat{\mathcal{G}}F(x_i, y_i^{K_i}, v_i^{N_i})\|_{x_i}^2}{a_{i+1}}$$

$$\leq C_a + 4\left(F(x_0) - F^*\right) + \frac{2L_F C_a^2}{a_0^2} + \frac{2(t+1)\epsilon_{y,v}}{a_0}. \tag{17}$$

The proof is complete. $\qquad\square$

Similar to (10), we define the thresholds for the step sizes $b_k$ and $c_n$ as follows:

$$C_b := \max\left\{2\zeta L_g \frac{L_g}{\mu}, b_0\right\}, \quad C_c := \max\{L_g, c_0\}. \tag{18}$$

Before proving Proposition 3.2, we need two technical lemmas.

**Lemma G.3.** *Suppose that Assumptions 3.1, 3.2, and 3.3 hold. Then, we have*

$$d^2(y_t^{k+1}, y^*(x_t)) \leq \left(1 + \frac{1}{b_{k+1}^2}\zeta L_g^2 - \frac{\mu}{b_{k+1}}\right) d^2(y_t^k, y^*(x_t)).$$

*Proof.* By Lemma B.1, we have

$$d^2(y_t^{k+1}, y^*(x_t)) \tag{19}$$

$$\leq d^2(y_t^k, y^*(x_t)) + \frac{1}{b_{k+1}^2}\zeta\|\mathcal{G}_y g(x_t, y_t^k)\|_{y_t^k}^2 + \frac{2}{b_{k+1}}\langle \mathcal{G}_y g(x_t, y_t^k), \mathrm{Exp}_{y_t^k}^{-1} y^*(x_t)\rangle_{y_t^k}$$

$$\leq d^2(y_t^k, y^*(x_t)) + \frac{1}{b_{k+1}^2}\zeta\|\mathcal{G}_y g(x_t, y_t^k)\|_{y_t^k}^2 + \frac{2}{b_{k+1}}\left(g(x_t, y^*(x_t)) - g(x_t, y_t^k) - \frac{\mu}{2}d^2(y_t^k, y^*(x_t))\right)$$

$$\leq \left(1 + \frac{1}{b_{k+1}^2}\zeta L_g^2 - \frac{\mu}{b_{k+1}}\right) d^2(y_t^k, y^*(x_t)),$$

where the second inequality follows from the geodesic strong convexity of $g$, and the third inequality follows from

$$\|\mathcal{G}_y g(x_t, y_t^k)\|_{y_t^k}^2 = \|\mathcal{G}_y g(x_t, y_t^k) - \mathcal{P}_{y^*(x_t)}^{y_t^k}\mathcal{G}_y g(x_t, y^*(x_t))\|_{y_t^k}^2 \leq L_g^2 d^2(y_t^k, y^*(x_t)).$$

The proof is complete. $\qquad\square$

**Lemma G.4.** *Suppose that Assumptions 3.1, 3.2, and 3.3 hold. Then, we have*

$$\langle \mathcal{G}_y g(x_t, y_t^k), \mathrm{Exp}_{y_t^k}^{-1}(y^*(x_t))\rangle_{y_t^k} \leq -\frac{1}{2L_g}\|\mathcal{G}_y g(x_t, y_t^k)\|_{y_t^k}^2.$$

*Proof.* By Assumption 3.3, given any $y \in \mathcal{M}_y$, it holds that

$$g(x_t, y^*(x_t)) \leq g(x_t, y) \leq g(x_t, y_t^k) + \langle \mathcal{G}_y g(x_t, y_t^k), \operatorname{Exp}_{y_t^k}^{-1}(y) \rangle_{y_t^k} + \frac{L_g}{2} d^2(y_t^k, y). \quad (20)$$

Taking $y = \operatorname{Exp}_{y_t^k}(-\frac{1}{L_g} \mathcal{G}_y g(x_t, y_t^k))$ in (20) derives

$$g(x_t, y^*(x_t)) \leq g(x_t, y_t^k) - \frac{1}{L_g} \|\mathcal{G}_y g(x_t, y_t^k)\|_{y_t^k}^2 + \frac{L_g}{2} \left\| \frac{1}{L_g} \mathcal{G}_y g(x_t, y_t^k) \right\|_{y_t^k}^2 = g(x_t, y_t^k) - \frac{1}{2L_g} \|\mathcal{G}_y g(x_t, y_t^k)\|_{y_t^k}^2,$$

which demonstrates that

$$g(x_t, y^*(x_t)) - g(x_t, y_t^k) \leq -\frac{1}{2L_g} \|\mathcal{G}_y g(x_t, y_t^k)\|_{y_t^k}^2. \quad (21)$$

By the geodesic strong convexity of $g$ w.r.t. $y$, we have

$$g(x_t, y^*(x_t)) - g(x_t, y_t^k) \geq \langle \mathcal{G}_y g(x_t, y_t^k), \operatorname{Exp}_{y_t^k}^{-1}(y^*(x_t)) \rangle_{y_t^k}.$$

This combines (21) implies that

$$\langle \mathcal{G}_y g(x_t, y_t^k), \operatorname{Exp}_{y_t^k}^{-1}(y^*(x_t)) \rangle_{y_t^k} \leq g(x_t, y^*(x_t)) - g(x_t, y_t^k) \leq -\frac{1}{2L_g} \|\mathcal{G}_y g(x_t, y_t^k)\|_{y_t^k}^2.$$

The proof is complete. $\qquad \square$

**Proof of Proposition 3.2**

*Proof.* For AdaRHD-GD, denote

$$\bar{K} := \frac{\log(C_b^2/b_0^2)}{\log(1 + \epsilon_y/C_b^2)} + \frac{1}{(\mu/\bar{b} - \zeta L_g^2/\bar{b}^2)} \log\left(\frac{\tilde{b}}{\epsilon_y}\right), \quad (22)$$

and

$$\bar{N}_{gd} := \frac{\log(C_c^2/c_0^2)}{\log(1 + \epsilon_v/C_c^2)} + \frac{\bar{c}}{\mu} \log\left(\frac{\tilde{c}}{\epsilon_v}\right), \quad (23)$$

where $\tilde{b} := \max\{\frac{L_g(\bar{b}-C_b)}{2}, 1\}$, $\tilde{c} := \max\{L_g(\bar{c} - C_c), 1\}$, $\bar{b} := C_b + 2L_g \left(\frac{2\epsilon_y}{\mu^2} + \frac{2L_g^2 C_f^2}{\mu^2 a_0^2} + 2\zeta \log\left(\frac{C_b}{b_0}\right) + \zeta\right)$ with $C_f = \left(\frac{2L_g^2 \epsilon_v}{\mu^2} + \frac{4L_g^2 l_f^2}{\mu^2} + 4l_f^2\right)^{\frac{1}{2}}$ is defined in Lemma F.5, and $\bar{c} := C_c + L_g \left(\frac{2\epsilon_y}{\mu^2} + \frac{8l_f^2}{\mu^2} + 2 \log\left(\frac{C_c}{c_0}\right) + 1\right)$.

For AdaRHD-CG, denote

$$\bar{N}_{cg} := \frac{1}{2} \log\left(\frac{4L_g^3 l_f^2}{\mu^3 \epsilon_v}\right) / \log\left(\frac{\sqrt{L_g/\mu} + 1}{\sqrt{L_g/\mu} - 1}\right).$$

We first show that $K_t \leq \bar{K}$ for all $0 \leq t \leq T$.

**If $k_1$ in Proposition G.1 does not exist**, it holds that $b_{K_t} \leq C_b$. By [92, Lemma 2], we must have $K_t \leq \frac{\log(C_b^2/b_0^2)}{\log(1+\epsilon_y/C_b^2)}$. If $K_t > \frac{\log(C_b^2/b_0^2)}{\log(1+\epsilon_y/C_b^2)}$, since $\|\mathcal{G}_y g(x_t, y_t^p)\|_{y_t^k}^2 \geq \epsilon_y$ and $b_k \leq C_b$ hold for all $k < K^t$, we have

$$b_{K_t}^2 = b_{K_t-1}^2 + \|\mathcal{G}_y g(x_t, y_t^{K_t-1})\|_{y_t^{K_t-1}}^2 = b_{K_t-1}^2 \left(1 + \frac{\|\mathcal{G}_y g(x_t, y_t^{K_t-1})\|_{y_t^{K_t-1}}^2}{b_{K_t-1}^2}\right)$$

$$\geq b_0^2 \prod_{k=0}^{K_t-1} \left(1 + \frac{\|\mathcal{G}_y g(x_t, y_t^k)\|_{y_t^k}^2}{b_k^2}\right) \geq b_0^2 \left(1 + \frac{\epsilon_y}{C_b^2}\right)^{K_t} > b_0^2 \left(\left(1 + \frac{\epsilon_y}{C_b^2}\right)^{1/\log(1+\epsilon_y/C_b^2)}\right)^{\log(C_b^2/b_0^2)} = C_b^2,$$

$$(24)$$

which contradicts $b_{K_t} \leq C_b$.

**If $k_1$ in Proposition G.1 exists**, then, we have $b_{k_1} \leq C_b$ and $b_{k_1+1} > C_b$. We first prove $k_1 \leq \frac{\log(C_b^2/b_0^2)}{\log(1+\epsilon_y/C_b^2)}$. If $k_1 > \frac{\log(C_b^2/b_0^2)}{\log(1+\epsilon_y/C_b^2)}$, similar to (24), we have

$$b_{k_1}^2 \geq b_0^2 \left(1 + \frac{\epsilon_y}{C_b^2}\right)^{k_1} > C_b^2,$$

which contradicts $b_{k_1} \leq C_b$.

By Lemma B.1 and the update mode of $y_t^{k_1}$, we have

$$d^2(y_t^{k_1}, y^*(x_t))$$

$$\leq d^2(y_t^{k_1-1}, y^*(x_t)) + \zeta \left\| \frac{\mathcal{G}_y g(x_t, y_t^{k_1-1})}{b_{k_1}} \right\|_{y_t^{k_1-1}}^2 + 2\frac{1}{b_{k_1}} \left\langle \mathcal{G}_y g(x_t, y_t^{k_1-1}), \operatorname{Exp}_{y_t^{k_1-1}}^{-1} y^*(x_t) \right\rangle_{y_t^{k_1-1}}$$

$$\overset{(a)}{\leq} d^2(y_t^{k_1-1}, y^*(x_t)) + \zeta \left\| \frac{\mathcal{G}_y g(x_t, y_t^{k_1-1})}{b_{k_1}} \right\|_{y_t^{k_1-1}}^2 + 2\frac{1}{b_{k_1}} \left( g(x_t, y^*(x_t)) - g(x_t, y_t^{k_1-1}) - \frac{\mu}{2} d^2(y_t^{k_1-1}, y^*(x_t)) \right)$$

$$\leq d^2(y_t^{k_1-1}, y^*(x_t)) + \zeta \frac{\|\mathcal{G}_y g(x_t, y_t^{k_1-1})\|_{y_t^{k_1-1}}^2}{b_{k_1}^2} \leq d^2(y_t^0, y^*(x_t)) + \zeta \sum_{k=0}^{k_1-1} \frac{\|\mathcal{G}_y g(x_t, y_t^k)\|_{y_t^k}^2}{b_{k+1}^2}$$

$$\overset{(b)}{\leq} d^2(y_{t-1}^{K_{t-1}}, y^*(x_t)) + \zeta \sum_{k=0}^{k_1-1} \frac{\|\mathcal{G}_y g(x_t, y_t^k)\|_{y_t^k}^2/b_0^2}{\sum_{i=0}^k \|\mathcal{G}_y g(x_t, y_t^i)\|_{y_t^i}^2/b_0^2}$$

$$\overset{(c)}{\leq} 2d^2(y_{t-1}^{K_{t-1}}, y^*(x_{t-1})) + 2d^2(y^*(x_{t-1}), y^*(x_t)) + \zeta \log\left(\sum_{k=0}^{k_1-1} \frac{\|\mathcal{G}_y g(x_t, y_t^k)\|_{y_t^k}^2}{b_0^2}\right) + \zeta$$

$$\overset{(d)}{\leq} \frac{2\epsilon_y}{\mu^2} + \frac{2L_g^2 \|\widehat{\mathcal{G}} F(x_{t-1}, y_{t-1}^{K_{t-1}}, v_{t-1}^{N_{t-1}})\|_{y_{t-1}^{K_{t-1}}}^2}{\mu^2 a_t^2} + \zeta \log\left(\sum_{k=0}^{k_1-1} \frac{\|\mathcal{G}_y g(x_t, y_t^k)\|_{y_t^k}^2}{b_0^2}\right) + \zeta$$

$$\overset{(e)}{\leq} \frac{2\epsilon_y}{\mu^2} + \frac{2L_g^2 C_f^2}{\mu^2 a_0^2} + 2\zeta \log\left(\frac{C_b}{b_0}\right) + \zeta, \tag{25}$$

where (a) follows from (2), (b) follows from the warm start of $y_t^0$, (c) follows from [10, Definition 10.1] and Lemma F.1, (d) uses Lemmas F.2 (1) and F.4, (e) follows from Lemma F.5 and $b_{k_1} \leq C_b$.

For any $K > k_1$, by Lemma G.3, we have

$$d^2(y_t^K, y^*(x_t)) \leq \left(1 - \left(\frac{\mu}{b_K} - \zeta \frac{L_g^2}{b_K^2}\right)\right) d^2(y_t^{K-1}, y^*(x_t)) \leq e^{-\left(\frac{\mu}{b_K} - \zeta \frac{L_g^2}{b_K^2}\right)(K-k_1)} d^2(y_t^{k_1}, y^*(x_t))$$

$$\leq e^{-\left(\frac{\mu}{b_K} - \zeta \frac{L_g^2}{b_K^2}\right)(K-k_1)} \left(\frac{2\epsilon_y}{\mu^2} + \frac{2L_g^2 C_f^2}{\mu^2 a_0^2} + 2\zeta \log\left(\frac{C_b}{b_0}\right) + \zeta\right), \tag{26}$$

where the second inequality follows from $b_K \geq C_b \geq 2\zeta L_g \frac{L_g}{\mu}$ and $1 - x \leq e^{-x}$ for $0 < x < 1$, specifically, when $b_K \geq 2\zeta L_g \frac{L_g}{\mu}$, it holds that $0 < \frac{\mu}{b_K} - \zeta \frac{L_g^2}{b_K^2} < 1$, and the third inequality follows from (25).

Similar to (17), we have

$$b_K = b_{K-1} + \frac{\|\mathcal{G}_y g(x_t, y_t^{K-1})\|_{y_t^{K-1}}^2}{b_K + b_{K-1}} \leq b_{k_1} + \sum_{k=k_1}^{K-1} \frac{\|\mathcal{G}_y g(x_t, y_t^k)\|_{y_t^k}^2}{b_{k+1}}. \tag{27}$$

Moreover, by the update mode of $y_t^K$ and Lemma B.1, we have

$$d^2(y_t^K, y^*(x_t)) \leq d^2(y_t^{K-1}, y^*(x_t)) + \zeta \frac{\|\mathcal{G}_y g(x_t, y_t^{K-1})\|_{y_t^{K-1}}^2}{b_K^2} + \frac{2\left\langle \mathcal{G}_y g(x_t, y_t^{K-1}), \operatorname{Exp}_{y_t^{K-1}}^{-1} y^*(x_t) \right\rangle_{y_t^{K-1}}}{b_K}$$

$$\leq d^2(y_t^{K-1}, y^*(x_t)) + \zeta \frac{\|\mathcal{G}_y g(x_t, y_t^{K-1})\|_{y_t^{K-1}}^2}{b_K^2} - \frac{\|\mathcal{G}_y g(x_t, y_t^{K-1})\|_{y_t^{K-1}}^2}{b_K L_g}$$

$$\leq d^2(y_t^{K-1}, y^*(x_t)) + \zeta \frac{\|\mathcal{G}_y g(x_t, y_t^{K-1})\|_{y_t^{K-1}}^2}{2 b_K \zeta L_g \frac{L_g}{\mu}} - \frac{\|\mathcal{G}_y g(x_t, y_t^{K-1})\|_{y_t^{K-1}}^2}{b_K L_g}$$

$$\leq d^2(y_t^{K-1}, y^*(x_t)) - \frac{(1-\mu/2L_g)}{L_g} \frac{\|\mathcal{G}_y g(x_t, y_t^{K-1})\|^2}{b_K} \leq d^2(y_t^{k_1}, y^*(x_t)) - \frac{(1-\mu/2L_g)}{L_g} \sum_{k=k_1}^{K-1} \frac{\|\mathcal{G}_y g(x_t, y_t^k)\|^2}{b_{k+1}},$$

$$\tag{28}$$

where the second and third inequalities follow from Lemma G.4 and $b_K \geq C_b \geq 2\zeta L_g \frac{L_g}{\mu}$, respectively.

By (25), we have

$$\left(1 - \frac{\mu}{2L_g}\right) \sum_{k=k_1}^{K-1} \frac{\|\mathcal{G}_y g(x_t, y_t^k)\|_{y_t^k}^2}{b_{k+1}} \leq L_g \left( d^2(y_t^{k_1}, y^*(x_t)) - d^2(y_t^K, y^*(x_t)) \right) \leq L_g d^2(y_t^{k_1}, y^*(x_t))$$

$$\leq L_g \left( \frac{2\epsilon_y}{\mu^2} + \frac{2L_g^2 C_f^2}{\mu^2 a_0^2} + 2\zeta \log\left(\frac{C_b}{b_0}\right) + \zeta \right).$$

Then, by (27), $\frac{1}{1-\mu/2L_g} \leq 2$, and $b_{k_1} \leq C_b$, we have

$$b_K \leq C_b + 2L_g \left( \frac{2\epsilon_y}{\mu^2} + \frac{2L_g^2 C_f^2}{\mu^2 a_0^2} + 2\zeta \log\left(\frac{C_b}{b_0}\right) + \zeta \right).$$

Denote $\bar{b} := C_b + 2L_g \left( \frac{2\epsilon_y}{\mu^2} + \frac{2L_g^2 C_f^2}{\mu^2 a_0^2} + 2\zeta \log\left(\frac{C_b}{b_0}\right) + \zeta \right)$. We need to show that $\frac{\mu}{\bar{b}} - \zeta \frac{L_g^2}{\bar{b}^2} > 0$.

Under (26), considering the monotonicity of $\frac{\mu}{b} - \zeta \frac{L_g^2}{b^2}$ w.r.t. $b$ when $b \geq 2\zeta L_g \frac{L_g}{\mu}$, it holds that

$0 < \frac{\mu}{\bar{b}} - \zeta \frac{L_g^2}{\bar{b}^2} \leq \frac{\mu}{b_K} - \zeta \frac{L_g^2}{b_K^2}$ as $\bar{b} \geq b_K \geq 2\zeta L_g \frac{L_g}{\mu}$. Then, we have

$$d^2(y_t^K, y^*(x_t)) \leq e^{-\left(\frac{\mu}{\bar{b}} - \zeta \frac{L_g^2}{\bar{b}^2}\right)(K-k_1)} \left( \frac{2\epsilon_y}{\mu^2} + \frac{2L_g^2 C_f^2}{\mu^2 a_0^2} + 2\zeta \log\left(\frac{C_b}{b_0}\right) + \zeta \right) = e^{-\left(\frac{\mu}{\bar{b}} - \zeta \frac{L_g^2}{\bar{b}^2}\right)(K-k_1)} \frac{\bar{b} - C_b}{2L_g}.$$

$$\tag{29}$$

Since $k_1 \leq \frac{\log(C_b^2/b_0^2)}{\log(1+\epsilon_y/C_b^2)}$, by the definition of $\bar{K}$ in (22), it holds that $\bar{K} > k_1$ as $\bar{b} \geq 1 > \epsilon_y$. If $\frac{L_g(\bar{b}-C_b)}{2} \leq 1$, it holds that $\tilde{b} = 1$ in (22). Then, by (29), we have

$$\|\mathcal{G}_y g(x_t, y_t^{\bar{K}})\|_{y_t^{\bar{K}}}^2 \leq L_g^2 d^2(y_t^{\bar{K}}, y^*(x_t)) \leq e^{-\left(\frac{\mu}{\bar{b}} - \zeta \frac{L_g^2}{\bar{b}^2}\right)(\bar{K}-k_1)} \frac{L_g(\bar{b}-C_b)}{2}$$

$$\overset{(a)}{\leq} e^{-\left(\frac{\mu}{\bar{b}} - \zeta \frac{L_g^2}{\bar{b}^2}\right)\frac{1}{(\mu/\bar{b} - \zeta L_g^2/\bar{b}^2)} \log\left(\frac{1}{\epsilon_y}\right)} \frac{L_g(\bar{b}-C_b)}{2} \overset{(b)}{\leq} e^{-\left(\frac{\mu}{\bar{b}} - \zeta \frac{L_g^2}{\bar{b}^2}\right)\frac{1}{(\mu/\bar{b} - \zeta L_g^2/\bar{b}^2)} \log\left(\frac{1}{\epsilon_y}\right)} = \epsilon_y,$$

$$\tag{30}$$

where (a) follows from the definition of $\bar{K}$, and (b) follows from $\frac{L_g(\bar{b}-C_b)}{2} \leq 1$.

If $\frac{L_g(\bar{b}-C_b)}{2} > 1$, it holds that $\tilde{b} = \frac{L_g(\bar{b}-C_b)}{2}$ in (22), and we have

$$\|\mathcal{G}_y g(x_t, y_t^{\bar{K}})\|_{y_t^{\bar{K}}}^2 \leq L_g^2 d^2(y_t^{\bar{K}}, y^*(x_t)) \leq e^{-\left(\frac{\mu}{\bar{b}} - \zeta \frac{L_g^2}{\bar{b}^2}\right)(\bar{K}-k_1)} \frac{L_g(\bar{b}-C_b)}{2}$$

$$\leq e^{-\left(\frac{\mu}{\bar{b}} - \zeta \frac{L_g^2}{\bar{b}^2}\right)\frac{1}{(\mu/\bar{b} - \zeta L_g^2/\bar{b}^2)} \log\left(\frac{L_g(\bar{b}-C_b)/2}{\epsilon_y}\right)} \frac{L_g(\bar{b}-C_b)}{2} = \epsilon_y, \tag{31}$$

Above all, we conclude that after at most $\bar{K}$ iterations, the condition $\|\mathcal{G}_y g(x_t, y_t^{\bar{K}})\|_{y_t^{\bar{K}}}^2 \leq \epsilon_y$ is satisfied, i.e., $K_t \leq \bar{K}$ holds for all $t \geq 0$. We complete the proof.

**AdaRHD-GD:** We show that $N_t \leq \bar{N}_{gd}$ for all $0 \leq t \leq T$.

**If $n_1$ in Proposition G.1 does not exist**, we have $c_{N_t} \leq C_c$. Similar to $K_t$ of the lower-level problem, we have $N_t \leq \frac{\log(C_c^2/c_0^2)}{\log(1+\epsilon_v/C_c^2)}$. If $N_t > \frac{\log(C_c^2/c_0^2)}{\log(1+\epsilon_v/C_c^2)}$, by $\|\nabla_v R(x_t, y_t^{K_t}, v_t^n)\|_{y_t^{K_t}}^2 \geq \epsilon_v$ and $c_n \leq C_c$ hold for all $n < N_t$, we have

$$c_{N_t}^2 = c_{N_t-1}^2 + \|\nabla_v R(x_t, y_t^{K_t}, v_t^{N_t-1})\|_{y_t^{K_t}}^2 = c_{N_t-1}^2 \left( 1 + \frac{\|\nabla_v R(x_t, y_t^{K_t}, v_t^{N_t-1})\|_{y_t^{K_t}}^2}{c_{N_t-1}^2} \right)$$

$$\geq c_0^2 \prod_{n=0}^{N_t-1} \left( 1 + \frac{\|\nabla_v R(x_t, y_t^{K_t}, v_t^n)\|_{y_t^{K_t}}^2}{c_n^2} \right) \geq c_0^2 \left( 1 + \frac{\epsilon_v}{C_c^2} \right)^{N_t} > C_c^2,$$

which contradicts $c_{N_t} \leq C_c$.

**If $n_1$ in Proposition G.1 exists** and $N_t \geq n_1$. Then, it holds that $c_{n_1} \leq C_c$ and $c_{n_1+1} > C_c$. Similarly, we have $n_1 \leq \frac{\log(C_c^2/c_0^2)}{\log(1+\epsilon_v/C_c^2)}$.

By the update rule of $v_t^{n_1}$ and the definition of $\hat{v}^*(x_t, y_t^{K_t}) = \arg\min_{v \in T_{y_t^{K_t}} \mathcal{M}_y} R(x_t, y_t^{K_t}, v)$, we have

$$\left\| v_t^{n_1} - \hat{v}^*(x_t, y_t^{K_t}) \right\|_{y_t^{K_t}}^2 = \left\| v_t^{n_1-1} - \frac{\nabla_v R(x_t, y_t^{K_t}, v_t^{n_1-1})}{c_{n_1}} - \hat{v}^*(x_t, y_t^{K_t}) \right\|_{y_t^{K_t}}^2$$

$$= \left\| v_t^{n_1-1} - \hat{v}^*(x_t, y_t^{K_t}) \right\|_{y_t^{K_t}}^2 + \left\| \frac{\nabla_v R(x_t, y_t^{K_t}, v_t^{n_1-1})}{c_{n_1}} \right\|_{y_t^{K_t}}^2$$

$$- \frac{2}{c_{n_1}} \left\langle v_t^{n_1-1} - \hat{v}^*(x_t, y_t^{K_t}), \nabla_v R(x_t, y_t^{K_t}, v_t^{n_1-1}) \right\rangle_{y_t^{K_t}}$$

$$\overset{(a)}{\leq} \left\| v_t^{n_1-1} - \hat{v}^*(x_t, y_t^{K_t}) \right\|_{y_t^{K_t}}^2 + \left\| \frac{\nabla_v R(x_t, y_t^{K_t}, v_t^{n_1-1})}{c_{n_1}} \right\|_{y_t^{K_t}}^2$$

$$- \frac{2}{c_{n_1} L_g} \left\| \nabla_v R(x_t, y_t^{K_t}, v_t^{n_1-1}) - \nabla_v R\left( x_t, y_t^{K_t}, \hat{v}^*(x_t, y_t^{K_t}) \right) \right\|_{y_t^{K_t}}^2$$

$$\leq \left\| v_t^{n_1-1} - \hat{v}^*(x_t, y_t^{K_t}) \right\|_{y_t^{K_t}}^2 + \left\| \frac{\nabla_v R(x_t, y_t^{K_t}, v_t^{n_1-1})}{c_{n_1}} \right\|_{y_t^{K_t}}^2$$

$$\leq \left\| v_t^0 - \hat{v}^*(x_t, y_t^{K_t}) \right\|_{y_t^{K_t}}^2 + \sum_{n=0}^{n_1-1} \left\| \frac{\nabla_v R(x_t, y_t^{K_t}, v_t^n)}{c_{n_1}} \right\|_{y_t^{K_t}}^2$$

$$\overset{(b)}{\leq} \left\| \mathcal{P}_{y_{t-1}^{K_{t-1}}}^{y_t^{K_t}} v_{t-1}^{N_{t-1}} - \hat{v}^*(x_t, y_t^{K_t}) \right\|_{y_t^{K_t}}^2 + \sum_{n=0}^{n_1-1} \frac{\|\nabla_v R(x_t, y_t^{K_t}, v_t^n)\|_{y_t^{K_t}}^2 / c_0^2}{\sum_{i=0}^n \|\nabla_v R(x_t, y_t^{K_t}, v_t^i)\|_{y_t^{K_t}}^2 / c_0^2}$$

$$\overset{(c)}{\leq} 2 \left\| v_{t-1}^{N_{t-1}} - \hat{v}^*(x_{t-1}, y_{t-1}^{K_{t-1}}) \right\|_{y_{t-1}^{K_{t-1}}}^2 + 2 \left\| \hat{v}^*(x_{t-1}, y_{t-1}^{K_{t-1}}) - \mathcal{P}_{y_t^{K_t}}^{y_{t-1}^{K_{t-1}}} \hat{v}^*(x_t, y_t^{K_t}) \right\|_{y_{t-1}^{K_{t-1}}}^2$$

$$+ \log \left( \sum_{n=0}^{n_1-1} \|\nabla_v R(x_t, y_t^{K_t}, v_t^n)\|_{y_t^{K_t}}^2 / c_0^2 \right) + 1$$

$$\leq 2 \left\| v_{t-1}^{K_{t-1}} - \hat{v}^*(x_{t-1}, y_{t-1}^{K_{t-1}}) \right\|_{y_{t-1}^{K_{t-1}}}^2 + 4 \left\| \hat{v}^*(x_{t-1}, y_{t-1}^{K_{t-1}}) \right\|_{y_{t-1}^{K_{t-1}}}^2 + 4 \left\| \hat{v}^*(x_t, y_t^{K_t}) \right\|_{y_t^{K_t}}^2$$

$$+ \log \left( \sum_{n=0}^{n_1-1} \|\nabla_v R(x_t, y_t^{K_t}, v_t^n)\|_{y_t^{K_t}}^2 / c_0^2 \right) + 1$$

$$\overset{(d)}{\leq} \frac{2\epsilon_v}{\mu^2} + \frac{8l_f^2}{\mu^2} + 2\log\left(\frac{C_c}{c_0}\right) + 1, \tag{32}$$

where (a) follows from Baillon-Haddad Theorem [4, Theorem 5.8 (iv)], i.e., for $y \in \mathcal{M}_y$ and $v \in T_y \mathcal{M}_y$, it holds that

$$\langle v - \hat{v}^*(x_t, y), \nabla_v R(x_t, y, v) - \nabla_v R(x_t, y, \hat{v}^*(x_t, y)) \rangle_y \geq \frac{1}{L_g} \| \nabla_v R(x_t, y, v) - \nabla_v R(x_t, y, \hat{v}^*(x_t, y)) \|_y^2, \tag{33}$$

(b) follows from the warm start of $v_t^0$, (c) uses Lemma F.1, and (d) follows from Lemmas F.3 (2) and F.4, and $c_{n_1} \leq C_c$.

Then, for all $N > n_1$, we have

$$\left\| v_t^N - \hat{v}^*(x_t, y_t^{K_t}) \right\|_{y_t^{K_t}}^2$$

$$= \left\| v_t^{N-1} - \hat{v}^*(x_t, y_t^{K_t}) \right\|_{y_t^{K_t}}^2 + \frac{\| \nabla_v R(x_t, y_t^{K_t}, v_t^{N-1}) \|_{y_t^{K_t}}^2}{c_n^2} - \frac{2 \left\langle v_t^{N-1} - \hat{v}^*(x_t, y_t^{K_t}), \nabla_v R(x_t, y_t^{K_t}, v_t^{N-1}) \right\rangle_{y_t^{K_t}}}{c_n}$$

$$\overset{(a)}{\leq} \left\| v_t^{N-1} - \hat{v}^*(x_t, y_t^{K_t}) \right\|_{y_t^{K_t}}^2 - \frac{1}{c_n}\left(2 - \frac{L_g}{c_n}\right) \left\langle v_t^{N-1} - \hat{v}^*(x_t, y_t^{K_t}), \nabla_v R(x_t, y_t^{K_t}, v_t^{N-1}) \right\rangle_{y_t^{K_t}}$$

$$\overset{(b)}{\leq} \left\| v_t^{N-1} - \hat{v}^*(x_t, y_t^{K_t}) \right\|_{y_t^{K_t}}^2 - \frac{1}{c_n}\left\langle v_t^{N-1} - \hat{v}^*(x_t, y_t^{K_t}), \nabla_v R(x_t, y_t^{K_t}, v_t^{N-1}) \right\rangle_{y_t^{K_t}}$$

$$\overset{(c)}{\leq} \left(1 - \frac{\mu}{c_n}\right) \left\| v_t^{N-1} - \hat{v}^*(x_t, y_t^{K_t}) \right\|_{y_t^{K_t}}^2 \overset{(d)}{\leq} e^{-\mu(N-n_1)/c_n} \left\| v_t^{n_1} - \hat{v}^*(x_t, y_t^{K_t}) \right\|_{y_t^{K_t}}^2$$

$$\overset{(e)}{\leq} e^{-\mu(N-n_1)/c_n} \left(\frac{2\epsilon_v}{\mu^2} + \frac{8l_f^2}{\mu^2} + 2\log\left(\frac{C_c}{c_0}\right) + 1\right),$$

where (a) follows from (33), (b) follows from $c_n > C_c \geq L_g$, (c) follows from $\nabla_v R(x_t, y_t^{K_t}, \hat{v}^*(x_t, y_t^{K_t})) = 0$ and the $\mu$-strong convexity of $R$, (d) follows from $c_n \geq C_c \geq L_g \geq \mu$ and $1 - x \leq e^{-x}$ for $0 < x < 1$, and (e) follows from (32).

Similar to (27), we have

$$c_N = c_{N-1} + \frac{\| \nabla_v R(x_t, y_t^{K_t}, v_t^{N-1}) \|_{y_t^{K_t}}^2}{c_N + c_{N-1}} \leq c_{n_1} + \sum_{n=n_1}^{N-1} \frac{\| \nabla_v R(x_t, y_t^{K_t}, v_t^n) \|_{y_t^{K_t}}^2}{c_{n+1}}. \tag{34}$$

For the second term of (34), we note that

$$\left\| v_t^N - \hat{v}^*(x_t, y_t^{K_t}) \right\|_{y_t^{K_t}}^2$$

$$= \left\| v_t^{N-1} - \hat{v}^*(x_t, y_t^{K_t}) \right\|_{y_t^{K_t}}^2 + \frac{\| \nabla_v R(x_t, y_t^{K_t}, v_t^{N-1}) \|_{y_t^{K_t}}^2}{c_N^2} - \frac{2 \left\langle v_t^N - \hat{v}^*(x_t, y_t^{K_t}), \nabla_v R(x_t, y_t^{K_t}, v_t^{N-1}) \right\rangle_{y_t^{K_t}}}{c_N}$$

$$\leq \left\| v_t^{N-1} - \hat{v}^*(x_t, y_t^{K_t}) \right\|_{y_t^{K_t}}^2 + \frac{\| \nabla_v R(x_t, y_t^{K_t}, v_t^{N-1}) \|_{y_t^{K_t}}^2}{c_N^2}$$

$$\quad - \frac{2 \left\| \nabla_v R(x_t, y_t^{K_t}, v_t^{N-1}) - \nabla_v R\left(x_t, y_t^{K_t}, \hat{v}^*(x_t, y_t^{K_t})\right) \right\|_{y_t^{K_t}}^2}{c_N L_g}$$

$$\leq \left\| v_t^{N-1} - \hat{v}^*(x_t, y_t^{K_t}) \right\|_{y_t^{K_t}}^2 - \frac{\left\| \nabla_v R(x_t, y_t^{K_t}, v_t^{N-1}) \right\|_{y_t^{K_t}}^2}{c_N L_g}$$

$$\leq \left\| v_t^{n_1} - \hat{v}^*(x_t, y_t^{K_t}) \right\|_{y_t^{K_t}}^2 - \sum_{n=n_1}^{N-1} \frac{\left\| \nabla_v R(x_t, y_t^{K_t}, v_t^n) \right\|_{y_t^{K_t}}^2}{c_{n+1} L_g},$$

where the first inequality follows from (33), and the second inequality follows from $c_N \geq C_c \geq L_g$. Then, by (32), it holds that

$$\sum_{n=n_1}^{N-1} \frac{\left\| \nabla_v R(x_t, y_t^{K_t}, v_t^n) \right\|_{y_t^{K_t}}^2}{c_{n+1}} \leq L_g \left( \left\| v_t^{n_1} - \hat{v}^*(x_t, y_t^{K_t}) \right\|_{y_t^{K_t}}^2 - \left\| v_t^N - \hat{v}^*(x_t, y_t^{K_t}) \right\|_{y_t^{K_t}}^2 \right)$$

$$\leq L_g \left\| v_t^{n_1} - \hat{v}^*(x_t, y_t^{K_t}) \right\|_{y_t^{K_t}}^2 \leq L_g \left( \frac{2\epsilon_y}{\mu^2} + \frac{8l_f^2}{\mu^2} + 2\log\left(\frac{C_c}{c_0}\right) + 1 \right).$$

Combining (34) derives

$$c_n \leq C_c + L_g \left( \frac{2\epsilon_y}{\mu^2} + \frac{8l_f^2}{\mu^2} + 2\log\left(\frac{C_c}{c_0}\right) + 1 \right).$$

Denote $\bar{c} := C_c + L_g \left( \frac{2\epsilon_y}{\mu^2} + \frac{8l_f^2}{\mu^2} + 2\log\left(\frac{C_c}{c_0}\right) + 1 \right)$. Then, considering the monotonicity of $\frac{\mu}{c}$ w.r.t. $c$, we have

$$\left\| v_t^N - \hat{v}^*(x_t, y_t^{K_t}) \right\|_{y_t^{K_t}}^2 \leq e^{-\mu(N_t - n_1)/\bar{c}} \left( \frac{2\epsilon_y}{\mu^2} + \frac{8l_f^2}{\mu^2} + 2\log\left(\frac{C_c}{c_0}\right) + 1 \right) = e^{-\mu(N_t - n_1)/\bar{c}} \frac{\bar{c} - C_c}{L_g}.$$

Since $n_1 \leq \frac{\log(C_c^2/c_0^2)}{\log(1 + \epsilon_v/C_c^2)}$, by the definition of $\bar{N}_{gd}$ in (23), it holds that $\bar{N}_{gd} > n_1$. Then, similar to (30) and (31), we have

$$\|\nabla_v R(x_t, y_t^{K_t}, v_t^{\bar{N}_{gd}})\|_{y_t^{K_t}}^2 \leq L_g^2 \left\| v_t^{\bar{N}_{gd}} - \hat{v}^*(x_t, y_t^{K_t}) \right\|_{y_t^{K_t}}^2 \leq e^{-\mu(\bar{N}_{gd} - n_1)/\bar{c}} L_g(\bar{c} - C_c) \leq \epsilon_v,$$

where the last inequality follows from the definition of $\bar{N}_{gd}$. This indicates that, after at most $\bar{N}_{gd}$ iterations, the condition $\|\nabla_v R(x_t, y_t^{K_t}, v_t^{\bar{N}_{gd}})\|_{y_t^{K_t}}^2 \leq \epsilon_v$ is satisfied, i.e, $N_t \leq \bar{N}_{gd}$ holds for all $t \geq 0$.

**AdaRHD-CG:** We show that $N_t \leq \bar{N}_{cg}$ for all $0 \leq t \leq T$.

Denote $\kappa_g = \frac{L_g}{\mu}$. From [10, Equation (6.19)], given $x_t \in \mathcal{M}_x$ and $y_t^{N_t} \in \mathcal{M}_y$, we have

$$\|v_t^{N_t} - \hat{v}_t^*(x_t, y_t^{N_t})\|_{y_t^{N_t}}^2 \leq 4\kappa_g \left( \frac{\sqrt{\kappa_g} - 1}{\sqrt{\kappa_g} + 1} \right)^{2N_t} \|\hat{v}_t^0 - \hat{v}_t^*(x_t, y_t^{N_t})\|_{y_t^{N_t}}^2 \leq 4\kappa_g \left( \frac{\sqrt{\kappa_g} - 1}{\sqrt{\kappa_g} + 1} \right)^{2N_t} \frac{l_f^2}{\mu^2},$$

where the last inequality we use the setting of that $\hat{v}_t^0 = 0$ and Lemma F.3 (2). Then we have

$$\|\nabla_v R(x_t, y_t^{K_t}, v_t^{N_t})\|_{y_t^{K_t}}^2 \leq L_g^2 \left\| v_t^{N_t} - \hat{v}^*(x_t, y_t^{K_t}) \right\|_{y_t^{K_t}}^2 \leq 4\kappa_g L_g^2 \frac{l_f^2}{\mu^2} \left( \frac{\sqrt{\kappa_g} - 1}{\sqrt{\kappa_g} + 1} \right)^{2N_t},$$

which implies that, after at most $\bar{N}_{cg}$ iterations, it holds that $\|\nabla_v R(x_t, y_t^{K_t}, v_t^{N_t})\|_{y_t^{K_t}}^2 \leq \epsilon_v$, where

$$\bar{N}_{cg} = \frac{1}{2}\log\left( \frac{4\kappa_g L_g^2 l_f^2}{\mu^2 \epsilon_v} \right) / \log\left( \frac{\sqrt{\kappa_g} + 1}{\sqrt{\kappa_g} - 1} \right) = \frac{1}{2}\log\left( \frac{4L_g^3 l_f^2}{\mu^3 \epsilon_v} \right) / \log\left( \frac{\sqrt{L_g/\mu} + 1}{\sqrt{L_g/\mu} - 1} \right).$$

We complete the proof. □

### G.3 Proof of Theorem 3.1

*Proof.* **If $t_1$ in Proposition G.1 does not exist**, we have $a_T \leq C_a$. Then, by (11) in Lemma G.1, for $t < T$, we have

$$\frac{\|\mathcal{G}F(x_t)\|_{x_t}^2}{a_{t+1}} \leq 2\left( F(x_t) - F(x_{t+1}) \right) + \frac{L_F}{a_{t+1}^2} \|\widehat{\mathcal{G}}F(x_t, y_t^{K_t}, v_t^{N_t})\|_{x_t}^2 + \frac{\epsilon_{y,v}}{a_{t+1}},$$

where $\epsilon_{y,v}$ is defined in Lemma G.1. Summing it from $t = 0$ to $T - 1$, we have

$$\frac{1}{T} \sum_{t=0}^{T-1} \frac{\|\mathcal{G}F(x_t)\|_{x_t}^2}{a_{t+1}} \leq \frac{2}{T} \left(F(x_0) - F(x_T)\right) + \frac{L_F}{a_0^2} \frac{1}{T} \sum_{t=0}^{T-1} \left\|\widehat{\mathcal{G}}F(x_t, y_t^{K_t}, v_t^{N_t})\right\|_{x_t}^2 + \frac{1}{T} \sum_{t=0}^{T-1} \frac{\epsilon_{y,v}}{a_{t+1}}$$

$$\leq \frac{1}{T} \left(2\left(F(x_0) - F^*\right) + \frac{L_F C_a^2}{a_0^2}\right) + \frac{\epsilon_{y,v}}{a_0} = \frac{F_0}{T} + \frac{\epsilon_{y,v}}{a_0}, \tag{35}$$

where the second inequality follows from $\sum_{t=0}^{T-1} \|\widehat{\mathcal{G}}F(x_t, y_t^{K_t}, v_t^{N_t})\|_{x_t}^2 \leq a_T^2 \leq C_a^2$, and $F_0$ is defined in (14).

**If $t_1$ in Proposition G.1 exists**, for any $t < t_1$, by (11) in Lemma G.1, we have

$$\frac{\|\mathcal{G}F(x_t)\|_{x_t}^2}{a_{t+1}} \leq 2\left(F(x_t) - F(x_{t+1})\right) + \frac{L_F}{a_{t+1}^2} \|\widehat{\mathcal{G}}F(x_t, y_t^{K_t}, v_t^{N_t})\|_{x_t}^2 + \frac{\epsilon_{y,v}}{a_{t+1}}. \tag{36}$$

For any $t \geq t_1$, by (12) in Lemma G.1, we have

$$\frac{\|\mathcal{G}F(x_t)\|_{x_t}^2}{a_{t+1}} \leq 2\left(F(x_t) - F(x_{t+1})\right) + \frac{\epsilon_{y,v}}{a_{t+1}}. \tag{37}$$

Summing (36) and (37) from 0 to $T - 1$, we have

$$\frac{1}{T} \sum_{t=0}^{T-1} \frac{\|\mathcal{G}F(x_t)\|_{x_t}^2}{a_{t+1}} = \frac{1}{T} \left(\sum_{t=0}^{t_1-1} \frac{\|\mathcal{G}F(x_t)\|_{x_t}^2}{a_{t+1}} + \sum_{t=t_1}^{T-1} \frac{\|\mathcal{G}F(x_t)\|_{x_t}^2}{a_{t+1}}\right)$$

$$\leq \frac{1}{T} \left(\sum_{t=0}^{T-1} 2\left(F(x_t) - F(x_{t+1})\right) + \frac{L_F}{a_0^2} \sum_{t=0}^{t_1-1} \left\|\widehat{\mathcal{G}}F(x_t, y_t^{K_t}, v_t^{N_t})\right\|_{x_t}^2 + \sum_{t=0}^{T-1} \frac{\epsilon_{y,v}}{a_{t+1}}\right)$$

$$\leq \frac{1}{T} \left(2\left(F(x_0) - F^*\right) + \frac{L_F C_a^2}{a_0^2}\right) + \frac{\epsilon_{y,v}}{a_0} = \frac{F_0}{T} + \frac{\epsilon_{y,v}}{a_0},$$

where the last inequality follows from Assumption 3.4 and $a_{t_1} \leq C_a$, and $F_0$ is defined in Lemma G.2. This result is equivalent to (35).

Then, since $a_{t+1} \leq a_T$, by Lemma G.2, we have

$$\frac{1}{T} \sum_{t=0}^{T-1} \|\mathcal{G}F(x_t)\|_{x_t}^2 \leq \left(\frac{F_0}{T} + \frac{\epsilon_{y,v}}{a_0}\right) a_T \leq \left(\frac{F_0}{T} + \frac{\epsilon_{y,v}}{a_0}\right) \left(C_a + 2F_0 + \frac{2T\epsilon_{y,v}}{a_0}\right). \tag{38}$$

Since $\epsilon_y = 1/T$ and $\epsilon_v = 1/T$, by the definition of $\epsilon_{y,v} := \frac{\bar{L}^2}{\mu^2}(\epsilon_y + \epsilon_v)$ in Lemma G.1, we have $\epsilon_{y,v} = \frac{2\bar{L}^2}{T\mu^2}$. Then, by the definition of $F_0$, (38) is equivalent to

$$\frac{1}{T} \sum_{t=0}^{T-1} \|\mathcal{G}F(x_t)\|_{x_t}^2 \leq \frac{\left(F_0 + 2\bar{L}^2/a_0\mu^2\right)\left(C_a + 2\left(F_0 + 2\bar{L}^2/a_0\mu^2\right)\right)}{T} = \mathcal{O}\left(\frac{1}{T}\right).$$

Let $C := (F_0 + \frac{2\bar{L}^2}{a_0\mu^2})(C_a + 2(F_0 + \frac{2\bar{L}^2}{a_0\mu^2}))$. The proof is complete. $\qquad \square$

### G.4 Proof of Corollary 3.1

*Proof.* By Theorem 3.1, we have
$$T = \mathcal{O}(1/\epsilon).$$
For the resolution of the lower-level problem, we have

$$K_t = \mathcal{O}\left(\frac{1}{\log(1 + \epsilon)}\right) = \mathcal{O}\left(\frac{1}{\epsilon}\right).$$

Similarly, for the resolution of the linear system, we have

AdaRHD-GD:

$$N_t = \mathcal{O}\left(\frac{1}{\epsilon}\right).$$

AdaRHD-CG:

$$N_t = \mathcal{O}\left(\log \frac{1}{\epsilon}\right).$$

Then, it is evident that the gradient complexities of $f$ and $g$ are $G_f = \mathcal{O}(1/\epsilon)$ and $G_g = \mathcal{O}(1/\epsilon^2)$, respectively. The complexities of computing the second-order cross derivative and Hessian-vector product of $g$ are $JV_g = \mathcal{O}(1/\epsilon)$, $HV_g = \mathcal{O}(1/\epsilon^2)$ for AdaRHD-GD, and $HV_g = \tilde{\mathcal{O}}(1/\epsilon)$ for AdaRHD-CG, respectively. $\qquad \square$

# H   Proofs for Section 3.4

This section provides the proofs of the results from Section 3.4. For consistency, we retain the notations introduced in Section 3.3, such as $C_a$, $C_b$, $C_c$, etc.

Firstly, similar to Proposition G.1, we first give a lemma that concerns the step sizes $a_1$, $b_t$, and $c_t$.

**Proposition H.1.** *Suppose that Assumptions 3.1, 3.2, 3.3, 3.5, and 3.6 hold. Denote $\{T, K, N\}$ as the iterations of $\{x, y, v\}$. Given any constants $C_a \geq a_0$, $C_b \geq b_0$, $C_c \geq c_0$, then, we have*

*AdaRHD-GD:*

*(1) either $a_t \leq C_a$ for any $t \leq T$, or $\exists t_1 \leq T$ such that $a_{t_1} \leq C_a$, $a_{t_1+1} > C_a$;*

*(2) either $b_t \leq C_b$ for any $t \leq K$, or $\exists k_1 \leq K$ such that $b_{k_1} \leq C_b$, $b_{k_1+1} > C_b$;*

*(3) either $c_t \leq C_c$ for any $t \leq N$, or $\exists n_1 \leq N$ such that $c_{n_1} \leq C_c$, $c_{n_1+1} > C_c$.*

*We then consider the case where the conjugate gradient solves the linear system. Therefore, we do not need to consider the step size $c_t$.*

*AdaRHD-CG:*

*(1) either $a_t \leq C_a$ for any $t \leq T$, or $\exists t_1 \leq T$ such that $a_{t_1} \leq C_a$, $a_{t_1+1} > C_a$;*

*(2) either $b_t \leq C_b$ for any $t \leq K$, or $\exists k_1 \leq K$ such that $b_{k_1} \leq C_b$, $b_{k_1+1} > C_b$;*

## H.1   Proof of Proposition D.1

### H.1.1   Improvement on objective function for one step update

Similar to Lemma G.1, we have the following lemma.

**Lemma H.1.** *Suppose that Assumptions 3.1, 3.2, 3.3, 3.5, and 3.6 hold. Then, we have*

$$F(x_{t+1}) \leq F(x_t) - \frac{1}{2a_{t+1}}\|\mathcal{G}F(x_t)\|_{x_t}^2 - \frac{1}{2a_{t+1}}\left(1 - \frac{\bar{L}_F}{a_{t+1}}\right)\|\widehat{\mathcal{G}}F(x_t, y_t^{K_t}, v_t^{N_t})\|_{x_t}^2 + \frac{\epsilon_{y,v}}{2a_{t+1}}.$$

*Furthermore, if $t_1$ in Proposition H.1 exists, then for $t \geq t_1$, we have*

$$F(x_{t+1}) \leq F(x_t) - \frac{1}{2a_{t+1}}\|\mathcal{G}F(x_t)\|_{x_t}^2 - \frac{1}{4a_{t+1}}\|\widehat{\mathcal{G}}F(x_t, y_t^{K_t}, v_t^{N_t})\|_{x_t}^2 + \frac{\epsilon_{y,v}}{2a_{t+1}}, \qquad (39)$$

*where $\bar{L}_F = (L_F c_u + 2c_R(l_f + l_f L_g/\mu)$, $\epsilon_{y,v} = \frac{\bar{L}^2}{\mu^2}(\epsilon_y + \epsilon_v)$, $\bar{L} := \max\{\sqrt{2}(L_g(1 + \frac{L_f}{\mu} + \frac{l_f \rho}{\mu^2}) + \frac{l_f \rho}{\mu}), \sqrt{2}L_g\}$, and $c_u, c_R$ are defined in Assumption 3.6.*

*Proof.* By Lemma F.2, we have

$$F(x_{t+1}) \leq F(x_t) + \left\langle \mathcal{G}F(x_t), \operatorname{Exp}_{x_t}^{-1}(x_{t+1})\right\rangle_{x_t} + \frac{L_F}{2}d(x_{t+1}, x_t)^2$$

$$=F(x_t) + \left\langle \mathcal{G}F(x_t), \operatorname{Exp}_{x_t}^{-1}(x_{t+1}) - \operatorname{Retr}_{x_t}^{-1}(x_{t+1})\right\rangle_{x_t} + \left\langle \mathcal{G}F(x_t), \operatorname{Retr}_{x_t}^{-1}(x_{t+1})\right\rangle_{x_t} + \frac{L_F}{2}d(x_{t+1}, x_t)^2$$

$$\leq F(x_t) + c_R\left(l_f + l_f\frac{L_g}{\mu}\right)\|\operatorname{Retr}_{x_t}^{-1}(x_{t+1})\|_{x_t}^2 - \frac{1}{a_{t+1}}\left\langle \mathcal{G}F(x_t), \widehat{\mathcal{G}}F\left(x_t, y_t^{K_t}, v_t^{N_t}\right)\right\rangle_{x_t}$$

$$+ \frac{L_F c_u}{2a_{t+1}^2} \|\widehat{\mathcal{G}}F\left(x_t, y_t^{K_t}, v_t^{N_t}\right)\|_{x_t}^2$$

$$\leq F(x_t) + \frac{c_R \left(l_f + l_f L_g/\mu\right)}{a_{t+1}^2} \|\widehat{\mathcal{G}}F\left(x_t, y_t^{K_t}, v_t^{N_t}\right)\|_{x_t}^2 + \frac{L_F c_u}{2a_{t+1}^2} \|\widehat{\mathcal{G}}F\left(x_t, y_t^{K_t}, v_t^{N_t}\right)\|_{x_t}^2$$

$$- \frac{1}{a_{t+1}} \|\mathcal{G}F(x_t)\|_{x_t}^2 - \frac{1}{a_{t+1}} \|\widehat{\mathcal{G}}F\left(x_t, y_t^{K_t}, v_t^{N_t}\right)\|_{x_t}^2 + \frac{1}{a_{t+1}} \|\mathcal{G}F(x_t) - \widehat{\mathcal{G}}F\left(x_t, y_t^{K_t}, v_t^{N_t}\right)\|_{x_t}^2$$

$$= F(x_t) - \frac{1}{2a_{t+1}} \|\mathcal{G}F(x_t)\|_{x_t}^2 - \frac{1}{2a_{t+1}}\left(1 - \frac{L_F c_u + 2c_R\left(l_f + l_f L_g/\mu\right)}{a_{t+1}}\right) \|\widehat{\mathcal{G}}F(x_t, y_t^{K_t}, v_t^{N_t})\|_{x_t}^2$$

$$+ \frac{1}{2a_{t+1}} \left\|\mathcal{G}F(x_t) - \widehat{\mathcal{G}}F\left(x_t, y_t^{K_t}, v_t^{N_t}\right)\right\|_{x_t}^2$$

$$\leq F(x_t) - \frac{1}{2a_{t+1}} \|\mathcal{G}F(x_t)\|_{x_t}^2 - \frac{1}{2a_{t+1}}\left(1 - \frac{L_F c_u + 2c_R\left(l_f + l_f L_g/\mu\right)}{a_{t+1}}\right) \|\widehat{\mathcal{G}}F(x_t, y_t^{K_t}, v_t^{N_t})\|_{x_t}^2 + \frac{\epsilon_{y,v}}{2a_{t+1}},$$

$$(40)$$

where the second inequality uses Lemma F.2 (2) and Assumption 3.6.

**If $t_1$ in Proposition H.1 exists,** then for $t \geq t_1$, we have $a_{t+1} > C_a \geq 2(L_F c_u + 2c_R(l_f + l_f L_g/\mu))$. The desired result of (39) follows from (40). The proof is complete. $\qquad\square$

Similar to (10), we define a threshold for parameters $a_t$ when giving an upper bound of the step size $a_t$.
$$C_a := \max\{2\bar{L}_F, a_0\}.$$
where $\bar{L}_F = (L_F c_u + 2c_R(l_f + l_f L_g/\mu))$ is defined in Lemmas H.1.

Similar to Lemma G.2, we have the following upper bound for the step size $a_t$.

**Lemma H.2.** *Suppose that Assumptions 3.1, 3.2, 3.3, 3.4, 3.5, and 3.6 hold. If $t_1$ in Proposition H.1 does not exist, we have $a_t \leq C_a$ for any $t \leq T$.*

*If there exists $t_1 \leq T$ described in Proposition H.1, we have*

$$\begin{cases} a_t \leq C_a, & t \leq t_1; \\ a_t \leq C_a + 2F_0 + \dfrac{2t\epsilon_{y,v}}{a_0}, & t \geq t_1, \end{cases}$$

*where we define*

$$F_0 := 2\left(F(x_0) - F^*\right) + \frac{\bar{L}_F C_a^2}{a_0^2}. \tag{41}$$

The proof of Lemma H.2 closely parallels that of Lemma G.2, requiring only the substitution of $L_F$ with $\bar{L}_F$. Here we omit it.

Before proving Proposition D.1, similar to Lemma G.3, we present the following technical lemma when substituting the exponential mapping with the retraction mapping.

**Lemma H.3.** *Suppose that Assumptions 3.1, 3.2, 3.3, 3.5, and 3.6 hold. Then, we have*

$$d^2(y_t^{k+1}, y^*(x_t)) \leq \left(1 + \frac{1}{b_{k+1}^2}\bar{\zeta}L_g^2 - \frac{\mu}{b_{k+1}}\right) d^2(y_t^k, y^*(x_t)),$$

*where $\bar{\zeta} := \zeta c_u + 2\bar{D}c_R$.*

*Proof.* From Lemma B.1, by the definition that $\mathrm{Retr}_{y_t^k}^{-1} y_t^{k+1} = -\frac{1}{b_{k+1}}\mathcal{G}g(x_t, y_t^k)$ we have

$$d^2(y_t^{k+1}, y^*(x_t)) \leq d^2(y_t^k, y^*(x_t)) + \zeta d^2(y_t^k, y_t^{k+1}) - 2\left\langle \mathrm{Exp}_{y_t^k}^{-1} y_t^{k+1}, \mathrm{Exp}_{y_t^k}^{-1} y^*(x_t) \right\rangle_{y_t^k}$$

$$\leq d^2(y_t^k, y^*(x_t)) + \frac{1}{b_{k+1}^2}\zeta c_u \|\mathcal{G}_y g(x_t, y_t^k)\|_{y_t^k}^2 - 2\left\langle \mathrm{Exp}_{y_t^k}^{-1} y_t^{k+1} - \mathrm{Retr}_{y_t^k}^{-1} y_t^{k+1}, \mathrm{Exp}_{y_t^k}^{-1} y^*(x_t) \right\rangle_{y_t^k}$$

$$+ 2\frac{1}{b_{k+1}}\left\langle \mathcal{G}_y g(x_t, y_t^k), \mathrm{Exp}_{y_t^k}^{-1} y^*(x_t) \right\rangle_{y_t^k}$$

$$\leq d^2(y_t^k, y^*(x_t)) + \frac{1}{b_{k+1}^2}\zeta c_u \|\mathcal{G}_y g(x_t, y_t^k)\|_{y_t^k}^2 + 2\frac{1}{b_{k+1}}\left\langle \mathcal{G}_y g(x_t, y_t^k), \mathrm{Exp}_{y_t^k}^{-1} y^*(x_t)\right\rangle_{y_t^k}$$

$$+ 2\bar{D}\|\mathrm{Exp}_{y_t^k}^{-1} y_t^{k+1} - \mathrm{Retr}_{y_t^k}^{-1} y_t^{k+1}\|_{y_t^k}$$

$$\leq d^2(y_t^k, y^*(x_t)) + \frac{1}{b_{k+1}^2}(\zeta c_u + 2\bar{D}c_R)\|\mathcal{G}_y g(x_t, y_t^k)\|_{y_t^k}^2 + 2\frac{1}{b_{k+1}}\left\langle \mathcal{G}_y g(x_t, y_t^k), \mathrm{Exp}_{y_t^k}^{-1} y^*(x_t)\right\rangle_{y_t^k}$$

$$\leq \left(1 + \frac{1}{b_{k+1}^2}(\zeta c_u + 2\bar{D}c_R)L_g^2 - \frac{\mu}{b_{k+1}}\right) d^2(y_t^k, y^*(x_t)), \tag{42}$$

where the second inequality follows from Assumption 3.6 and

$$\langle \mathrm{Retr}_{y_t^k}^{-1} y_t^{k+1}, \mathrm{Exp}_{y_t^k}^{-1} y^*(x_t)\rangle_{y_t^k} = -\frac{1}{b_{k+1}}\langle \mathcal{G}_y g(x_t, y_t^k), \mathrm{Exp}_{y_t^k}^{-1} y^*(x_t)\rangle_{y_t^k},$$

the third inequality follows from Assumption 3.5 by employing [10, Proposition 10.22], i.e.,

$$\|\mathrm{Exp}_{y_t^k}^{-1} y^*(x_t)\| = d(y_t^k, y^*(x_t)) \leq \bar{D},$$

and the fourth inequality follows from Assumption 3.6. $\qquad\square$

Similar to (18), we define the following thresholds for the step sizes $b_k$ and $c_n$.

$$C_b := \max\left\{2\bar{\zeta} L_g \frac{L_g}{\mu}, b_0\right\}, \quad C_c := \max\{L_g, c_0\},$$

where $\bar{\zeta} = \zeta c_u + 2\bar{D}c_R$ are defined in Lemma H.3.

**Proof of Proposition D.1**

*Proof.* For AdaRHD-GD, denote

$$\bar{K} := \frac{\log(C_b^2/b_0^2)}{\log(1 + \epsilon_y/C_b^2)} + \frac{1}{(\mu/\bar{b} - \bar{\zeta}L_g^2/\bar{b}^2)}\log\left(\frac{\tilde{b}}{\epsilon_y}\right), \tag{43}$$

and

$$\bar{N}_{gd} := \frac{\log(C_c^2/c_0^2)}{\log(1 + \epsilon_v/C_c^2)} + \frac{\bar{c}}{\mu}\log\left(\frac{\tilde{c}}{\epsilon_v}\right),$$

where $\tilde{b} := \max\{\frac{L_g(\bar{b} - C_b)}{2}, 1\}$, $\tilde{c} := \max\{L_g(\bar{c} - C_c), 1\}$, $\bar{b} := C_b + 2L_g\left(\frac{2\epsilon_y}{\mu^2} + \frac{2L_g^2 C_f^2}{\mu^2 a_0^2} + 2\bar{\zeta}\log\left(\frac{C_b}{b_0}\right) + \bar{\zeta}\right)$ with $C_f = \left(\frac{2L_g^2\epsilon_v}{\mu^2} + \frac{4L_g^2 l_f^2}{\mu^2} + 4l_f^2\right)^{\frac{1}{2}}$ is defined in Lemma F.5, and $\bar{c} := C_c + L_g\left(\frac{2\epsilon_y}{\mu^2} + \frac{8l_f^2}{\mu^2} + 2\log\left(\frac{C_c}{c_0}\right) + 1\right)$.

For AdaRHD-CG, denote

$$\bar{N}_{cg} := \frac{1}{2}\log\left(\frac{4L_g^3 l_f^2}{\mu^3 \epsilon_v}\right)\Big/\log\left(\frac{\sqrt{L_g/\mu} + 1}{\sqrt{L_g/\mu} - 1}\right).$$

First, we show that $K_t \leq \bar{K}$ for all $0 \leq t \leq T$.

**If $k_1$ in Proposition H.1 does not exist**, we have $b_{K_t} \leq C_b$. By [92, Lemma 2], it holds that $K_t \leq \frac{\log(C_b^2/b_0^2)}{\log(1+\epsilon_y/C_b^2)}$. If $K_t > \frac{\log(C_b^2/b_0^2)}{\log(1+\epsilon_y/C_b^2)}$, by $\|\mathcal{G}_y g(x_t, y_t^p)\|_{x_t}^2 \geq \epsilon_y$ and $b_k \leq C_b$ holds for all $k < K^t$, we have

$$b_{K_t}^2 = b_{K_t-1}^2 + \|\mathcal{G}_y g(x_t, y_t^{K_t-1})\|_{y_t^{K_t-1}}^2 = b_{K_t-1}^2\left(1 + \frac{\|\mathcal{G}_y g(x_t, y_t^{K_t-1})\|_{y_t^{K_t-1}}^2}{b_{K_t-1}^2}\right)$$

$$\geq b_0^2 \prod_{k=0}^{K_t-1}\left(1 + \frac{\|\mathcal{G}_y g(x_t, y_t^k)\|_{y_t^k}^2}{b_k^2}\right) \geq b_0^2\left(1 + \frac{\epsilon_y}{C_b^2}\right)^{K_t} > b_0^2\left((1 + \frac{\epsilon_y}{C_b^2})^{1/\log(1+\epsilon_y/C_b^2)}\right)^{\log(C_b^2/b_0^2)} = C_b^2. \tag{44}$$

This contradicts $b_{K_t} \leq C_b$.

**If $k_1$ in Proposition H.1 exists**, we have $b_{k_1} \leq C_b$ and $b_{k_1+1} > C_b$. We first prove $k_1 \leq \frac{\log(C_b^2/b_0^2)}{\log(1+\epsilon_y/C_b^2)}$. If $k_1 > \frac{\log(C_b^2/b_0^2)}{\log(1+\epsilon_y/C_b^2)}$, similar to (44), we have

$$b_{k_1}^2 \geq b_0^2 \left(1 + \frac{\epsilon_y}{C_b^2}\right)^{k_1} > C_b^2,$$

which contradicts the setting that $b_{k_1} \leq C_b$.

By Lemma H.3, we have

$$d^2(y_t^{k_1}, y^*(x_t)) \leq d^2(y_t^{k_1-1}, y^*(x_t)) + \zeta d^2(y_t^{k_1-1}, y_t^{k_1}) - 2\left\langle \operatorname{Exp}_{y_t^{k_1-1}}^{-1} y_t^{k_1}, \operatorname{Exp}_{y_t^{k_1-1}}^{-1} y^*(x_t)\right\rangle_{y_t^{k_1-1}}$$

$$\overset{(a)}{\leq} d^2(y_t^{k_1-1}, y^*(x_t)) + \zeta c_u \left\|\frac{\mathcal{G}_y g(x_t, y_t^{k_1-1})}{b_{k_1}}\right\|_{y_t^{k_1-1}}^2 - 2\left\langle \operatorname{Exp}_{y_t^{k_1-1}}^{-1} y_t^{k_1} - \operatorname{Retr}_{y_t^{k_1-1}}^{-1} y_t^{k_1}, \operatorname{Exp}_{y_t^{k_1-1}}^{-1} y^*(x_t)\right\rangle_{y_t^{k_1-1}}$$

$$+ 2\frac{1}{b_{k_1}}\left\langle \mathcal{G}_y g(x_t, y_t^{k_1-1}), \operatorname{Exp}_{y_t^{k_1-1}}^{-1} y^*(x_t)\right\rangle_{y_t^{k_1-1}}$$

$$\overset{(b)}{\leq} d^2(y_t^{k_1-1}, y^*(x_t)) + \zeta c_u \left\|\frac{\mathcal{G}_y g(x_t, y_t^{k_1-1})}{b_{k_1}}\right\|_{y_t^{k_1-1}}^2 + 2\frac{1}{b_{k_1}}\left\langle \mathcal{G}_y g(x_t, y_t^{k_1-1}), \operatorname{Exp}_{y_t^{k_1-1}}^{-1} y^*(x_t)\right\rangle_{y_t^{k_1-1}}$$

$$+ 2\bar{D}\|\operatorname{Exp}_{y_t^{k_1-1}}^{-1} y_t^{k_1} - \operatorname{Retr}_{y_t^{k_1-1}}^{-1} y_t^{k_1}\|_{y_t^{k_1-1}}$$

$$\overset{(c)}{\leq} d^2(y_t^{k_1-1}, y^*(x_t)) + (\zeta c_u + 2\bar{D}c_R)\left\|\frac{\mathcal{G}_y g(x_t, y_t^{k_1-1})}{b_{k_1}}\right\|_{y_t^{k_1-1}}^2 + 2\frac{1}{b_{k_1}}\left\langle \mathcal{G}_y g(x_t, y_t^{k_1-1}), \operatorname{Exp}_{y_t^{k_1-1}}^{-1} y^*(x_t)\right\rangle_{y_t^{k_1-1}}$$

$$\overset{(d)}{\leq} d^2(y_t^{k_1-1}, y^*(x_t)) + \bar{\zeta}\frac{\|\mathcal{G}_y g(x_t, y_t^{k_1-1})\|_{y_t^{k_1-1}}^2}{b_{k_1}^2} \leq d^2(y_t^0, y^*(x_t)) + \bar{\zeta}\sum_{p=0}^{k_1-1}\frac{\|\mathcal{G}_y g(x_t, y_t^p)\|_{y_t^p}^2}{b_{k+1}^2}$$

$$\overset{(e)}{\leq} d^2(y_{t-1}^{K_t-1}, y^*(x_t)) + \bar{\zeta}\sum_{p=0}^{k_1-1}\frac{\|\mathcal{G}_y g(x_t, y_t^p)\|_{y_t^p}^2/b_0^2}{\sum_{k=0}^p \|\mathcal{G}_y g(x_t, y_t^k)\|_{y_t^k}^2/b_0^2}$$

$$\overset{(f)}{\leq} 2d^2(y_{t-1}^{K_t-1}, y^*(x_{t-1})) + 2d^2(y^*(x_{t-1}), y^*(x_t)) + \bar{\zeta}\log\left(\sum_{p=0}^{k_1-1}\frac{\|\mathcal{G}_y g(x_t, y_t^p)\|_{y_t^p}^2}{b_0^2}\right) + \bar{\zeta}$$

$$\overset{(g)}{\leq} \frac{2\epsilon_y}{\mu^2} + \frac{2L_g^2\|\widehat{\mathcal{G}}F(x_{t-1}, y_{t-1}^{K_{t-1}}, v_{t-1}^{N_{t-1}})\|_{y_{t-1}^{K_{t-1}}}^2}{\mu^2 a_t^2} + \bar{\zeta}\log\left(\sum_{p=0}^{k_1-1}\frac{\|\mathcal{G}_y g(x_t, y_t^p)\|_{y_t^p}^2}{b_0^2}\right) + \bar{\zeta}$$

$$\overset{(h)}{\leq} \frac{2\epsilon_y}{\mu^2} + \frac{2L_g^2 C_f^2}{\mu^2 a_0^2} + 2\bar{\zeta}\log\left(\frac{C_b}{b_0}\right) + \bar{\zeta}, \tag{45}$$

where (a) follows from the second inequality of (42), (b) follows from Assumption 3.5, (c) follows from Assumption 3.6, (d) follows from the convexity of $g$, (e) follows from the warm start of $y_t^0$, (f) follows from Lemma F.1, (g) follows from Lemmas F.2 (1) and F.4, and (h) follows from Lemma F.5 and $b_{k_1} \leq C_b$.

For all $K > k_1$, by Lemma H.3, we have

$$d^2(y_t^K, y^*(x_t)) \leq \left(1 - \left(\frac{\mu}{b_k} - \bar{\zeta}\frac{L_g^2}{b_k^2}\right)\right)d^2(y_t^{K-1}, y^*(x_t)) \leq e^{-\left(\frac{\mu}{b_k} - \bar{\zeta}\frac{L_g^2}{b_k^2}\right)(K-k_1)}d^2(y_t^{k_1}, y^*(x_t))$$

$$\leq e^{-\left(\frac{\mu}{b_k} - \bar{\zeta}\frac{L_g^2}{b_k^2}\right)(K-k_1)}\left(\frac{2\epsilon_y}{\mu^2} + \frac{2L_g^2 C_f^2}{\mu^2 a_0^2} + 2\bar{\zeta}\log\left(\frac{C_b}{b_0}\right) + \bar{\zeta}\right), \tag{46}$$

where the first inequality follows from $b_k \geq C_b \geq 2\bar{\zeta}L_g\frac{L_g}{\mu}$ and $1 - m \leq e^{-m}$ for $0 < m < 1$, specifically, we have $C_b \geq 2\bar{\zeta}L_g\frac{L_g}{\mu}$ and it holds that $0 < \frac{\mu}{b_k} - \bar{\zeta}\frac{L_g^2}{b_k^2} < 1$, and the second inequality follows from (45).

Similar to (27), we have

$$b_K = b_{K-1} + \frac{\|\mathcal{G}_y g(x_t, y_t^{K-1})\|_{y_t^{K-1}}^2}{b_K + b_{K-1}} \leq b_{k_1} + \sum_{k=k_1}^{K-1} \frac{\|\mathcal{G}_y g(x_t, y_t^k)\|_{y_t^k}^2}{b_{k+1}}. \tag{47}$$

Similar to (28), we have

$$d^2(y_t^K, y^*(x_t)) \leq d^2(y_t^{K-1}, y^*(x_t)) + \bar{\zeta}\frac{\|\mathcal{G}_y g(x_t, y_t^{K-1})\|_{y_t^{K-1}}^2}{b_K^2} + \frac{2\left\langle \mathcal{G}_y g(x_t, y_t^{K-1}), \text{Exp}_{y_t^{K-1}}^{-1} y^*(x_t) \right\rangle_{y_t^{K-1}}}{b_K}$$

$$\leq d^2(y_t^{K-1}, y^*(x_t)) + \bar{\zeta}\frac{\|\mathcal{G}_y g(x_t, y_t^{K-1})\|_{y_t^{K-1}}^2}{2b_K\bar{\zeta}L_g\frac{L_g}{\mu}} - \frac{\|\mathcal{G}_y g(x_t, y_t^{K-1})\|_{y_t^{K-1}}}{b_K L_g}$$

$$\leq d^2(y_t^{K-1}, y^*(x_t)) - \frac{(1 - \mu/2L_g)}{L_g}\frac{\|\mathcal{G}_y g(x_t, y_t^{K-1})\|^2}{b_K} \leq d^2(y_t^{k_1-1}, y^*(x_t)) - \frac{(1 - \mu/2L_g)}{L_g}\sum_{k=k_1}^{K-1}\frac{\|\mathcal{G}_y g(x_t, y_t^k)\|^2}{b_{k+1}},$$

where the second inequality follows from Lemma G.4 and $b_K \geq C_b \geq 2\bar{\zeta}L_g\frac{L_g}{\mu}$.

By (45), we have

$$\left(1 - \frac{\mu}{2L_g}\right)\sum_{k=k_1}^{K-1}\frac{\|\mathcal{G}_y g(x_t, y_t^k)\|_{y_t^k}^2}{b_{k+1}} \leq L_g\left(d^2(y_t^{k_1}, y^*(x_t)) - d^2(y_t^K, y^*(x_t))\right) \leq L_g d^2(y_t^{k_1}, y^*(x_t))$$

$$\leq L_g\left(\frac{2\epsilon_y}{\mu^2} + \frac{2L_g^2 C_f^2}{\mu^2 a_0^2} + 2\bar{\zeta}\log\left(\frac{C_b}{b_0}\right) + \bar{\zeta}\right).$$

Then, by (47) and $\frac{1}{1-\mu/2L_g} \leq 2$, we have

$$b_K \leq C_b + 2L_g\left(\frac{2\epsilon_y}{\mu^2} + \frac{2L_g^2 C_f^2}{\mu^2 a_0^2} + 2\bar{\zeta}\log\left(\frac{C_b}{b_0}\right) + \bar{\zeta}\right).$$

Denote $\bar{b} := C_b + 2L_g\left(\frac{2\epsilon_y}{\mu^2} + \frac{2L_g^2 C_f^2}{\mu^2 a_0^2} + 2\bar{\zeta}\log\left(\frac{C_b}{b_0}\right) + \bar{\zeta}\right)$. Then, under (46), considering the monotonicity of $\frac{\mu}{b} - \bar{\zeta}\frac{L_g^2}{b^2}$ w.r.t. $b$ when $b \geq 2\bar{\zeta}L_g\frac{L_g}{\mu}$, it holds that $\frac{\mu}{\bar{b}} - \bar{\zeta}\frac{L_g^2}{\bar{b}^2} \leq \frac{\mu}{b_K} - \bar{\zeta}\frac{L_g^2}{b_K^2}$ as $\bar{b} \geq b_K \geq 2\bar{\zeta}L_g\frac{L_g}{\mu}$, we have

$$d^2(y_t^K, y^*(x_t)) \leq e^{-\left(\frac{\mu}{\bar{b}} - \bar{\zeta}\frac{L_g^2}{\bar{b}^2}\right)(K-k_1)}\left(\frac{2\epsilon_y}{\mu^2} + \frac{2L_g^2 C_f^2}{\mu^2 a_0^2} + 2\bar{\zeta}\log\left(\frac{C_b}{b_0}\right) + \bar{\zeta}\right) = e^{-\left(\frac{\mu}{\bar{b}} - \bar{\zeta}\frac{L_g^2}{\bar{b}^2}\right)(K-k_1)}\frac{\bar{b} - C_b}{2L_g}. \tag{48}$$

Since $k_1 \leq \frac{\log(C_b^2/b_0^2)}{\log(1+\epsilon_y/C_b^2)}$, by the definition of $\bar{K}$ in (43), it is obvious that $\bar{K} \geq k_1$. Therefore, similar to (30) and (31), by (48), we have

$$\|\mathcal{G}_y g(x_t, y_t^{\bar{K}})\|_{y_t^{\bar{K}}}^2 \leq L_g^2 d^2(y_t^{\bar{K}}, y^*(x_t)) \leq e^{-\left(\frac{\mu}{\bar{b}} - \bar{\zeta}\frac{L_g^2}{\bar{b}^2}\right)(\bar{K}-k_1)}\frac{L_g(\bar{b} - C_b)}{2} \leq \epsilon_y,$$

which indicates that after at most $\bar{K}$ iterations, the condition $\|\mathcal{G}_y g(x_t, y_t^{\bar{K}})\|_{y_t^{\bar{K}}}^2 \leq \epsilon_y$ is satisfied, i.e., $K_t \leq \bar{K}$ holds for all $t \geq 0$.

Regarding the total number of iterations $N_t$ required to solve the linear system, since solving it does not involve retraction mapping, the total iterations match those of Algorithm 1. The proof is complete. $\square$

## H.2 Proof of Theorem 3.2

*Proof.* The proof of Theorem 3.2 follows a similar approach to Theorem 3.1, requiring only the substitution of $L_F$ with $\bar{L}_F$ and the definition of the constant $C_{\text{retr}}$ as

$$C_{\text{retr}} := \left( F_0 + \frac{2\bar{L}_F^2}{a_0 \mu^2} \right) \left( C_a + 2 \left( F_0 + \frac{2\bar{L}_F^2}{a_0 \mu^2} \right) \right),$$

where $F_0$ is defined in (41) of Lemma H.2. Due to this structural similarity, the remaining proof is omitted. $\square$

# I Experimental details

This section expands on the experiments discussed in Section 4 and presents supplemental empirical evaluations. Subsection I.1 details the experimental configurations, while Subsection I.2 includes an additional implementation to further demonstrate the robustness of our proposed algorithm through additional empirical validation.

## I.1 Experimental Settings

All implementations are executed using the Geoopt framework [48], matching the reference implementation [33] to guarantee equitable comparison conditions. Furthermore, all the experiments are implemented based on Geoopt [48] and are implemented using Python 3.8 on a Linux server with 256GB RAM and 96-core AMD EPYC 7402 2.8GHz CPU.

### I.1.1 Simple problem

For $n = 100$, the maximum number of outer iterations in RHGD [33] is set to 200, whereas for $n = 1000$, this number is increased to 400. In Algorithm 3, the number of outer iterations is set to $T = 1000$ for AdaRHD-GD and $T = 10000$ for AdaRHD-CG. Following [33], we set $\lambda = 0.01$ and fix the step sizes in RHGD as $\eta_x = \eta_y = 0.5$. To ensure consistency, the initial step sizes in Algorithm 3 are set to $a_0 = b_0 = c_0 = 2$.

For solving the linear system, we adopt the conjugate gradient (CG) method as the default approach in RHGD. For our algorithm, the CG procedure is terminated when the residual norm falls below a tolerance of $10^{-10}$ or the number of CG iterations reaches 50. Note that the tolerance $10^{-10}$ for CG procedure can be seen as $\mathcal{O}(\frac{1}{T^\alpha})$ with $\alpha > 1$ is a constant, this will not influence the total iterations and the complexity results since the convergence rate of the CG procedure is linear. Moreover, for the GD procedure, we employ the stopping criteria specified in Algorithm 3 or terminate when the number of iterations reaches $\min\{50\lfloor t/5 \rfloor, 500\}$, where $\lfloor \cdot \rfloor$ denotes the floor function, and $t$ represents the iteration round $t$. This warm-start-inspired strategy aims to rapidly obtain a reasonable solution during early iterations, and progressively increase internal iterations to satisfy the gradient norm tolerance $\epsilon_v$ in later stages.

For solving the lower-level problem, our algorithm follows the stopping criteria outlined in Algorithm 1 or terminates when the number of iterations reaches $\min\{50\lfloor t/5 \rfloor, 500\}$. For RHGD, the maximum number of iterations is set to either 50 or 20.

Moreover, the number of outer iterations in our algorithm is set to $T = 1000$ for AdaRHD-CG and $T = 10000$ for AdaRHD-GD. Additionally, our algorithm terminates if the approximate hypergradient norm falls below $10^{-4}$. For RHGD, the maximum number of outer iterations is set to 200.

### I.1.2 Robustness analysis

The experimental configurations in Subsection 4.2 are maintained consistently across all comparative analyses in Subsection 4.1, except that the number of outer iterations is set to be $T = 300$ for AdaRHD-CG and $T = 500$ for AdaRHD-GD.

## I.2  Additional experiments: hyper-representation problem

### I.2.1  Shallow hyper-representation for regression

In this subsection, we adopt a shallow regression framework from [33], incorporating a Stiefel manifold constraint [40, 34, 36] to preserve positive-definiteness in learned representations. The SPD representation is transformed into Euclidean space using the matrix logarithm, thereby establishing a bijection between SPD and symmetric matrices, followed by vectorizing the upper-triangular entries via the $\text{vec}(\cdot)$ operation. We maintain the problem framework established in Subsection 4.2, replacing the cross-entropy loss with the least-squares loss. The revised problem is as follows:

$$
\begin{aligned}
\min_{\mathbf{A}\in\text{St}(d,r)} \quad & \sum_{i\in\mathcal{D}_{\text{val}}} \frac{(\text{vec}(\log(\mathbf{A}^\top \mathbf{D}_i \mathbf{A}))\beta^*(\mathbf{A})-y_i)^2}{2|\mathcal{D}_{\text{val}}|}, \\
\text{s.t.} \quad & \beta^*(\mathbf{A}) = \arg\min_{\beta\in\mathbb{R}^{r(r+1)/2}} \sum_{i\in\mathcal{D}_{\text{tr}}} \frac{(\text{vec}(\log(\mathbf{A}^\top \mathbf{D}_i \mathbf{A}))\beta-y_i)^2}{2|\mathcal{D}_{\text{tr}}|} + \frac{\lambda}{2}\|\beta\|^2,
\end{aligned}
\tag{49}
$$

where $\lambda > 0$ is the regular parameter. Following [33], we set $d = 50$, $r = 10$, and $\lambda = 0.1$.

In Problem (49), the upper-level objective is optimized on the validation set, whereas the lower-level problem is solved on the training set. Following [33], each element $y_i$ is defined by $y_i = \text{vec}(\log(\mathbf{A}^\top \mathbf{D}_i \mathbf{A}))\beta + \epsilon_i$, where $\mathbf{A}$ and $\mathbf{D}_i$ are randomly generated matrices, $\beta$ is a randomly generated vector, and $\epsilon_i$ represents a Gaussian noise term. We generate $n$ samples of $\mathbf{D}_i$, evenly divided between the validation and training sets. To align with the step size initialization in [33], we initialize parameters $a_0$, $b_0$, and $c_0$ in our algorithm to 20. Moreover, we configure the number of outer iterations as $T = 1000$ for AdaRHD-CG and $T = 10000$ for AdaRHD-GD. All other settings align with Section 4.1.

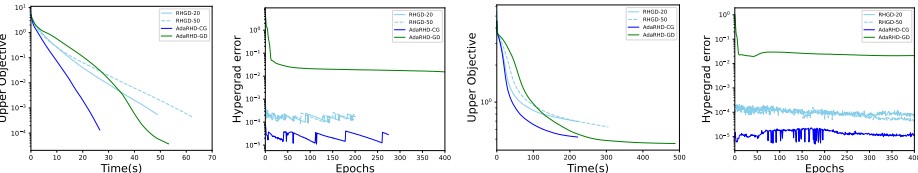

Figure 4: Shallow hyper-representation for regression (Left two column: $n = 200$, Right two column: $n = 1000$).

In Figure 4, we illustrate the validation set loss (Upper Objective) versus time for $n = 200$ and $n = 1000$, respectively. We observe that AdaRHD-CG converges more effectively and rapidly to a smaller objective loss compared to AdaRHD-GD and RHGD. Furthermore, note that although AdaRHD-GD initially converges more slowly than RHGD (potentially due to imprecise hypergradient estimation affecting the upper-level loss reduction), Figure 4 demonstrates that AdaRHD-GD achieves a lower objective loss than RHGD at specific time intervals. These results collectively highlight the efficiency and robustness of our algorithm, as demonstrated in Section 4.1.

### I.2.2  Deep hyper-representation for classification

In this subsection, we further evaluate the efficiency and robustness of our proposed algorithm by extending the deep hyper-representation classification framework introduced in Subsection 4.2 to dataset sampling ratios of $12.5\%$ and $25\%$. For each configuration, five independent trials are executed using distinct random seeds. In each trial, a randomly sampled validation set is reserved for the upper-level problem, with an equally sized training partition allocated to the lower-level task. All remaining experimental parameters align with those defined in Subsection 4.2.

Figure 5 depicts the validation accuracy against outer epochs (iterations) with $12.5\%$ and $25\%$ dataset sampling ratios (due to computational constraints, a subset of the full dataset is utilized to ensure manageable training durations). Table 3 summarizes the computational time required to achieve specified validation accuracy thresholds across algorithms. For each random seed, an independent data subset is sampled. Our algorithm exhibits rapid validation accuracy improvements during initial outer iterations, followed by more gradual advancement in later stages. This behavior likely originates from the inner-loop strategy: early iterations necessitate more intensive inner-loop computations to attain high precision, whereas later stages leverage accumulated progress, thereby requiring fewer

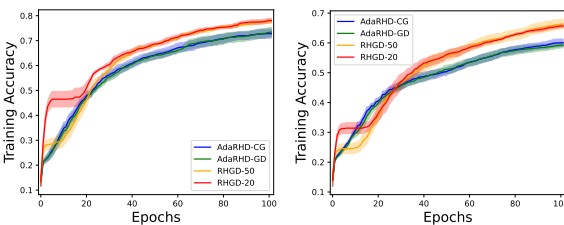

Figure 5: Deep hyper-representation for classification (Left: sampling ratio 12.5%, Right: sampling ratio 25%).

inner-loop iterations. In contrast, RHGD-20 inner iterations demonstrate rapid initial accuracy gains but experience pronounced degradation subsequently, particularly at the 25% sampling ratio, where its accuracy drops below that of our method. This instability is inherent in RHGD when fewer inner iterations are employed.

Furthermore, while RHGD-20 exhibits faster time-to-50%-accuracy, its elevated variance (evident in Table 3) confirms inherent instability. Notably, our method achieves target accuracy faster than RHGD despite requiring additional outer iterations. This advantage, maintained under identical initial step sizes, demonstrates both the efficacy and stability of our adaptive approach in addressing Riemannian bilevel optimization problems.

Table 3: Time required for each algorithm to first achieve a specified validation accuracy. "Time(s) to $X\%$" is defined as the time elapsed until the validation accuracy first reaches $X\%$. Values are presented as mean $\pm$ standard deviation across five independent trials using fixed random seeds for all algorithms. Statistically significant results are marked in bold. "Data ratio" refers to the proportion of the full dataset.

|  | AdaRHD-CG | AdaRHD-GD | RHGD-50 | RHGD-20 |
|---|---|---|---|---|
| data ratio = 12.5% | | | | |
| Time(s) to 50% | $878.60\pm$ **90.44** | $1258.83 \pm 138.54$ | $1188.66 \pm 148.46$ | **530.03** $\pm 306.29$ |
| Time(s) to 70% | **1666.17** $\pm$ **96.74** | $1865.13 \pm 426.80$ | $3070.22 \pm 169.30$ | $2085.61 \pm 236.33$ |
| data ratio = 25% | | | | |
| Time(s) to 50% | **2511.13** $\pm$ **233.16** | $3408.80 \pm 508.51$ | $3944.26 \pm 589.55$ | $2598.87 \pm 645.08$ |

### I.2.3 Robust optimization on manifolds

In this subsection, following [54, Section 7.1], we consider the robust optimization on manifolds, which have the following forms:

$$\min_{p\in\Delta_n} \quad \left\|p - \tfrac{1}{n}\right\|^2 - \sum_{i=1}^n p_i\ell(y;\xi_i) \\ \text{s.t.} \quad y \in \arg\min_{y\in\mathbb{S}_{++}^d} \sum_{i=1}^n p_i\ell(y;\xi_i), \tag{50}$$

where $\Delta_n := \{p \in \mathbb{R}^n : \sum_{i=1}^n p_i = 1, p_i > 0\}$ and $\mathbb{S}_{++}^d := \{y \in \mathbb{R}^{d\times d} : y \succ \mathbf{0}\}$ represent the multinomial manifold (or probability simplex) and positive definite matrix space, respectively. For robust Karcher mean (KM) problem, $\ell(y;\xi_i)$ in Problem (50) represents the geodesic distance of two positive definite matrices [7]:

$$\ell(y;\xi_i) = \|\log(y^{-1/2}\xi_i y^{-1/2})\|_F^2,$$

where $\xi_i$'s are the symmetric positive definite data matrices. For robust maximum likelihood estimation (MLE) problem, $\ell(y;\xi_i)$ in Problem (50) represents the log likelihood of the Gaussian distribution [75]:

$$\ell(y;\xi_i) = \frac{1}{2}\log\det(y) + \frac{\xi_i^\top y^{-1}\xi_i}{2},$$

where $\xi_i$'s are sample vectors from the Gaussian distribution.

In this experiment, we set the step sizes as $\eta_x = \eta_y = 1/a_0 = 1/b_0 = 1/c_0 = 0.1$, with all other settings identical to those in Section 4.2. For the robust Karcher mean problem, we set $d = 20$ and $n \in \{5, 10, 20, 50\}$, randomly generating the symmetric positive definite data matrices $\xi_i$'s. For the

robust maximum likelihood estimation problem, we set $d = 50$ and $n \in \{100, 300, 500, 1000\}$, and randomly generate the sample vectors from a Gaussian distribution with mean $\mathbf{0}$ and covariance matrix being a random positive semidefinite symmetric matrix. We use five different random seeds to conduct experiments for each group. The results are shown in Figure 6, where the x-axis represents the average time, and the y-axis represents the average value of the ergodic performance $\min_{i \in [0,t]} \|\widehat{\mathcal{G}}F(x_i, y_i^{K_i}, v_i^{N_i})\|_{x_i}^2$. Note that for the MLE experiment with $n = 100$, we do not report the results of RHGD-20 because it fails to converge in this case. As shown in Figure 6, compared to RHGD, AdaRHD achieves superior results more efficiently and robustly, which further demonstrates the superiority and robustness of our algorithm.

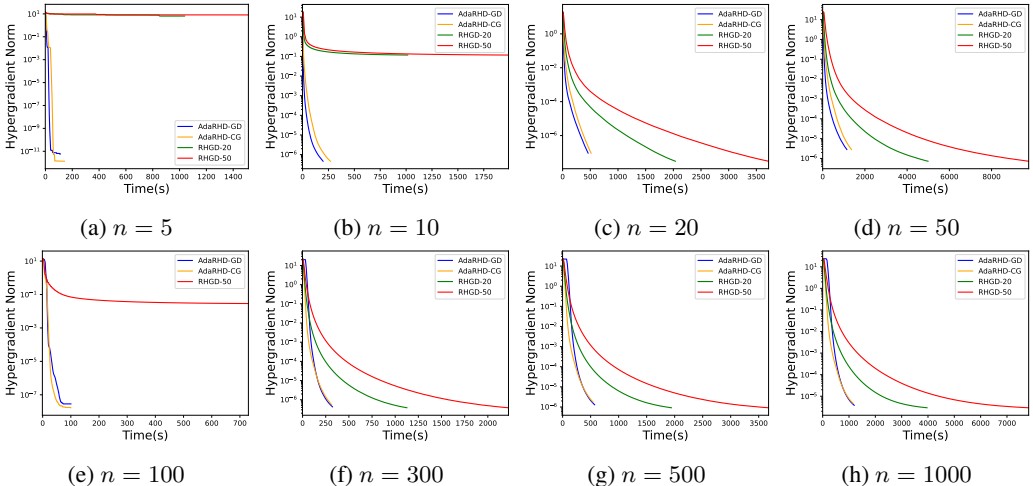

Figure 6: Robust optimization on manifolds (Top row: KM model with $d = 20$ and different $n$, Bottom row: MLE model with $d = 50$ and different $n$).

## J    Broader impacts

This paper proposes an algorithm for solving Riemannian bilevel optimization problems. No negative societal impacts are anticipated from this research that warrant disclosure.

