# OpenReview forum: "An Adaptive Algorithm for Bilevel Optimization on Riemannian Manifolds"
_NeurIPS.cc/2025/Conference — NeurIPS 2025 poster_

### Official Review · Reviewer_6VLZ · 2025-06-10

**Clarity:** 3
**Significance:** 2
**Originality:** 2
**Rating:** 4
**Confidence:** 4

**Summary:**

The paper studies bilevel optimization on Riemannian manifolds. The paper proposes an adaptive hypergradient method for solving the bilevel problem on manifold. The idea is to normalize the hypergradient with a moving-average of gradient norm, following the existing works in the Euclidean space. The paper derives convergence results on manifolds and show a matching gradient/Hessian complexities as non-adaptive method. The paper shows some promising experiment results compared to non-adaptive method.

**Questions:**

I have the following comments and questions:

1. The paper is largely based on prior works [1,2] for developing the adaptive algorithm for Riemannian bilevel optimization. The convergence analysis follows similarly from the existing works as well. Thus the contributions of the paper seem limited. The authors are expected to include more discussions on the significance of the results as well as whether there exists other technical contributions.

2. In Proposition 3.2, the upper bound on K_t includes the curvature constant \zeta. However for highly curved manifolds, the curvature constant is significantly large, which would renders the second term negative. I wonder what it means to have a negative term in this case?

3. What is the role of curvature in the derived convergence and complexities? This is very important and should be extensively compared with the work in the Euclidean space where curvature is zero.

4. The gradient and Hessian complexity of the lower-level problem is O(1/\epsilon) worse compared to the non-adaptive methods, which could be the main bottleneck of the method.


[1] Tuning-Free Bilevel Optimization: New Algorithms and Convergence Analysis

[2] A Framework for Bilevel Optimization on Riemannian Manifolds

**Ethical Concerns:**

["NO or VERY MINOR ethics concerns only"]

**Final Justification:**

My concerns regarding the role of curvature has been well addressed.

**Limitations:**

Yes.

**Quality:**

3

**Strengths And Weaknesses:**

The paper presents a complete theoretical analysis and provides the first non-asymptotic convergence analysis for adaptive methods on manifolds. Experiment results demonstrate great promise.

---

> ### Author Rebuttal · Authors · 2025-07-28
>
> We sincerely appreciate your valuable feedback and insightful comments on our work. Your constructive suggestions will enable us to further elaborate on the significance of our technical contributions in our revision.
>
> ### **Questions**
>
> __*For Question 1*__:
>
> We appreciate this insightful comment. Intuitively, AdaRHD can be regarded as an extension of D-TFBO [1] to Riemannian manifolds, given their analogous algorithmic structures. However, the theoretical generalization is highly non-trivial and introduces several technical challenges beyond D-TFBO. Below, we outline some key difficulties and innovations of this paper:
>
> - (**Curvature Constant**) Given the geometric structure of Riemannian manifolds, determining the upper bound for the iterations $K_t$ (i.e., the total iterations for solving the lower-level problem at $t$-th iteration) requires incorporating the curvature $\zeta$, as established in Proposition 3.2. Specifically, the coefficient $\frac{1}{(\mu/\bar{b} - \zeta L_g^2/\bar{b}^2)}$ in the second term $\frac{1}{(\mu/\bar{b} - \zeta L_g^2/\bar{b}^2)}\log(\frac{\tilde{b}}{\epsilon_y})$ of $K_t$, demonstrates a quadratic dependency on $\bar{b}$ while being influenced by the curvature $\zeta$. This stands in contrast to the coefficient $\frac{\beta_{\max}}{\mu}$ ($\frac{\bar{b}}{\mu}$ in our notation) in the second term of $P_t$ ($K_t$ in our notation) of [1, Proposition 2], where the dependence on $\bar{b}$ is purely linear. Consequently, this discrepancy and the quadratic dependence necessitate a more intricate analysis: we must examine the monotonicity of $\frac{\mu}{\bar{b}} - \frac{\zeta L_g^2}{\bar{b}^2}$ w.r.t. $\bar{b}$ and determine an appropriate value. The detailed derivation of $\bar{b}$ is provided in lines 746–754 of this paper.
> - (**Inner Iteration**) Additionally, when deriving the upper bound for $K_t$, we must employ trigonometric distance bounds instead of Euclidean measures (Lemma G.3, Equation (24), line 741). This requirement substantially increases the difficulty of our convergence analysis relative to the Euclidean framework presented in [1]. Specifically, unlike the threshold $C_{\beta}$ ($C_{b}$ in our notation) defined in Equation (5) of [1], our analysis necessitates a more sophisticated threshold $C_b$ (cf. Equation (18)) to account for the Riemannian manifold geometry. Furthermore, in Algorithm 1, we utilize a conjugate gradient method (Algorithm 2) to solve the linear system, which is more efficient than the gradient descent solver in [1]. This technique reduces computational costs (particularly Hessian-vector products) while offering better convergence guarantees, compared to [1]. We have also observed in practice that AdaRHD-CG outperforms AdaRHD-GD. This is likely because, in the current iteration, CG can obtain a better solution with lower computational cost.
> - (**Related Method**) Compared to existing methods for RBO, e.g., [2], our algorithm is the first to incorporate a fully adaptive step-size strategy. This eliminates the need for prior knowledge of problem-dependent parameters, which are computationally expensive to estimate on Riemannian manifolds (particularly the curvature $\zeta$). Such adaptability significantly broadens the applicability of our method to RBO problems.
>
> Furthermore, building upon your insightful feedback, we will incorporate a more rigorous analysis in the revision to elucidate both the theoretical challenges and the intuitions underlying the method's effectiveness.
>
> __*For Question 2*__:
>
> We apologize for the confusion. The second term, $\frac{1}{(\mu/\bar{b} - \zeta L_g^2/\bar{b}^2)} \log\frac{\tilde{b}}{\epsilon_y}$, is indeed non-negative, which is an oversight in our writing. We will explicitly clarify this in the revision of Proposition 3.2. Specifically, we have established the non-negativity of $\frac{1}{(\mu/\bar{b} - \zeta L_g^2/\bar{b}^2)}$ in lines 746-748 of our paper: Considering the monotonicity of $\frac{\mu}{b} - \frac{\zeta L_g^2}{b^2}$ w.r.t. $b$ when $b \ge 2\zeta L_g\frac{L_g}{\mu}$, it follows that $0 < \frac{\mu}{\bar{b}} - \frac{\zeta L_g^2}{\bar{b}^2}$ as $\bar{b} \ge 2\zeta L_g\frac{L_g}{\mu}$ (the definition of $\bar{b}$ is given in line 746). Additionally, the element $\log\frac{\tilde{b}}{\epsilon_y}$ is also non-negative by the definition of $\tilde{b}$ in line 721 of our paper. Therefore, even if the curvature constant is significantly large, the second term in $K_t$ is still non-negative.
>
> __*For Question 3*__:
>
> We acknowledge the omission of a detailed discussion on curvature's role and will address this in our revision. You are correct that the curvature is zero in Euclidean space, and the farther the Riemannian manifold is from Euclidean space, the larger the curvature becomes. As shown in Lemma B.1, unlike the Euclidean distance bound $a^2 \le b^2 + c^2 + 2 bc$, we use a trigonometric distance bound dependent on the curvature $\zeta$. This bound is critical for convergence analysis (cf. Proposition 3.2, Lemmas G.3, H.3, and related results). Thus, curvature fundamentally influences convergence and complexity. Specifically, compared to D-TFBO [1]: Firstly, though our step-size adaptation also operates in two stages (cf. Propositions 3.2 and D.1), the maximum iterations per stage depend on $\zeta$. For the lower-level problem (Proposition 3.2), Stage 1 requires at most $\frac{\log(C_b^2/ b_0^2)}{\log(1+\epsilon_y/C_b^2)}$ iterations, and Stage 2 at most $\frac{1}{({\mu}/{\bar{b}} - \zeta {L_{g}^2}/{ \bar{b}^2})}\log\left(\frac{\tilde{b}}{\epsilon_y}\right)$ iterations. Here, $C_b, \bar{b}, \tilde{b}$ are all related to curvature $\zeta$, and they increase as the curvature $\zeta$ increases; Secondly, Lemma G.3 yields a technical result for the convergence of the lower-level problem:
> $$
> d^2(y_t^{k+1}, y^{\*}(x_t)) \le (1 + \zeta L_{g}^2/b_{k+1}^2  - \mu/b_{k+1}) d^2(y_t^{k}, y^{\*}(x_t)),
> $$
> which is different from the Euclidean setting (cf. Inequality (b) of Equation (18) in [1]):
> $$
> \\|y_t^{k+1} - y^{\*}(x_t)\\|^2 \le (1-\mu/b_{k+1})\\|y_t^k - y^{\*}(x_t)\\|^2.
> $$
> When $\zeta = 0$, this inequality in the Riemannian setting degenerates to the Euclidean setting. Therefore, the curvature $\zeta$ plays an important role in areas such as the number of **inner iterations estimation** and **the distance inequalities** involved in the convergence analysis, which brings challenges to the analysis of our algorithm.
>
> In the revision, we will include a detailed discussion on the role of curvature in our analysis, along with a comparative examination of its implications in Riemannian versus Euclidean settings.
>
> __*For Question 4*__:
>
> Thanks for pointing this out. It is correct that the gradient and Hessian complexity of the lower-level problem is $O(1/\epsilon)$ worse compared to the non-adaptive methods. We point out that the $O(1/\epsilon)$ gap stems from two key factors: (1) the additional iterations required to ensure convergence for solving the lower-level problems and linear systems, due to the absence of parameter-specific prior knowledge in RBO; and (2) the adaptive step-size strategy employed, which utilizes the inverse of cumulative gradient norms [3]. However, we point out that the gradient and Hessian complexities for the lower-level problem are the same as the standard AdaGrad-Norm algorithm [3], as well as D-TFBO [1].
>
> Furthermore, we have discussed this gap and aim to address the $O(1/\epsilon)$ gap by exploring alternative adaptive step-size strategies in future work (cf. Appendix I, lines 996–998). We will incorporate this discussion into the main text and provide a more detailed analysis in the revision. We believe that designing such an algorithm is an interesting direction, and the gap may be resolved by employing some other adaptive strategies from Riemannian and Euclidean settings: For Riemannian optimization, possible approaches include Riemannian SAM [4], RNGD [5], and the framework for Riemannian adaptive optimization [6]. In Euclidean optimization, alternatives such as adaptive accelerated gradient descent [7], AdaPGM [8], AdaBB [9], and AdaNAG [10] may be viable, rather than the inverse of cumulative gradient norms [3] of this paper. More specifically, with carefully designed adaptive step sizes, the method proposed in [7] (Euclidean space) achieves an improved convergence rate of $O(\log(1/\epsilon))$ compared to the $O(1/\epsilon)$ rate of [3]. However, directly applying similar adaptation strategies from [7] to Riemannian manifolds introduces significant challenges in convergence analysis due to the geometric structure. Nevertheless, we argue that eliminating this $O(1/\epsilon)$ gap through more refined algorithmic design and analysis represents a promising research direction.
>
> ### **References**
> [1] Yifan Yang, Hao Ban, Minhui Huang, Shiqian Ma, and Kaiyi Ji. Tuning-free bilevel optimization: New algorithms and convergence analysis.
>
> [2] Andi Han, Bamdev Mishra, Pratik Kumar Jawanpuria, and Akiko Takeda. A framework for bilevel optimization on Riemannian manifolds.
>
> [3] Yuege Xie, Xiaoxia Wu, and Rachel Ward. Linear convergence of adaptive stochastic gradient descent.
>
> [4] Jihun Yun and Eunho Yang. Riemannian SAM: Sharpness-aware minimization on Riemannian manifolds.
>
> [5] Jiang Hu, Ruicheng Ao, Anthony Man-Cho So, Minghan Yang, and Zaiwen Wen. Riemannian natural gradient methods.
>
> [6] Sakai, Hiroyuki, and Hideaki Iiduka. A general framework of Riemannian adaptive optimization methods with a convergence analysis.
>
> [7] Malitsky, Yura, and Konstantin Mishchenko. Adaptive gradient descent without descent.
>
> [8] Latafat, Puya, et al. Adaptive proximal algorithms for convex optimization under local Lipschitz continuity of the gradient.
>
> [9] Zhou, Danqing, Shiqian Ma, and Junfeng Yang. AdaBB: Adaptive Barzilai-Borwein method for convex optimization.
>
> [10] Suh, Jaewook J., and Shiqian Ma. An Adaptive and Parameter-Free Nesterov's Accelerated Gradient Method for Convex Optimization.

---

> > ### Comment · Reviewer_6VLZ · 2025-08-04
> >
> > I thank the authors for the responses. Many of my concerns have been well addressed and I would be glad to increase my score.

---

> > > ### Author Response · Authors · 2025-08-05
> > >
> > > We sincerely appreciate your recognition of our work and your constructive feedback, which helps improve our paper significantly. We are very glad to hear that many of your concerns have been well addressed, and we greatly appreciate the time you took to re-evaluate our paper and increase your score.

---

### Official Review · Reviewer_TNoH · 2025-07-01

**Clarity:** 4
**Significance:** 3
**Originality:** 2
**Rating:** 5
**Confidence:** 2

**Summary:**

This paper introduces the first fully adaptive method for Riemannian bilevel optimization, called the Adaptive Riemannian Hypergradient Descent (AdaRHD) algorithm.
The method is adaptive in the sense that it does not require prior knowledge of first- and second-order information or curvature parameters of the manifold (such as strong convexity, Lipschitz continuity, or curvature constants) to determine step sizes.
The algorithm is based on the approximate Riemannian hypergradient and furthermore employs an "inverse of cumulative gradient norm" strategy to design adaptive step sizes, inspired by [81].
This paper proves that their method achieves the same convergence complexity as existing non-adaptive methods.
Lastly, they also provide a variant achieving the same complexity bound and demonstrate the effectiveness of their method through experiments.

**Questions:**

- **Q1.** In the context of algorithm design, are there any novel and significant ideas the authors would like to highlight that did not appear in the Euclidean counterpart but are crucial for the proposed algorithm?

- **Q2.** In the context of techniques in Riemannian optimization, are there any new techniques the authors would like to highlight that were required to be developed while establishing the analysis of AdaRHD or AdaRHD-R?

**Ethical Concerns:**

["NO or VERY MINOR ethics concerns only"]

**Final Justification:**

The authors have kindly addressed my questions, and I will maintain my original score.

**Limitations:**

Yes.

**Paper Formatting Concerns:**

No issues.

**Quality:**

4

**Strengths And Weaknesses:**

**Strength**

- This paper overcome technical challenges and also provide promised results that can be expected from the title. Indeed, bilevel optimization is challenging compared to single-level problem, and achieving the same complexity with non-adaptive algorithm for an adaptive algorithm. And the most of the among keywords, Riemannian optimization is challenging compared to Euclidean optimization, since calculation of parameters and even simple calculation laws like cosine law due to the distortion.

- This paper introduces the algorithm which is the first of its kind, which achieves the all keywords appear in the title.

- The paper is well organized and the authors did a good job on the presentation.

**Weakness**

- **W1.** To the best of this reviewer's understanding, this paper provides an extension of the algorithm introduced in [81] to Riemannian optimization. This reviewer believes that such work is not trivial and requires overcoming various technical difficulties.
However, in the context of algorithm design, this reviewer is curious whether most of the key ideas for handling adaptiveness and the bilevel structure were perhaps already developed in the Euclidean case.
This point is closer to a question than a weakness; please see the related question.

---

> ### Author Rebuttal · Authors · 2025-07-28
>
> We sincerely appreciate your recognition of our work, as well as your valuable insights and thoughtful feedback on this paper.
> At the same time, we will make the corresponding additions to our revision.
>
> ### **Weaknesses**
> __*For Weakness 1*__:
>
> We appreciate the opportunity to clarify this point. You are correct that some key ideas for addressing adaptiveness and the bilevel structure are initially developed in the Euclidean setting [81]. In particular, the adaptive step-size design in our algorithm, AdaRHD, resembles that of D-TFBO [81], with the key distinction being the replacement of Euclidean gradients (hypergradients) with their Riemannian counterparts. Furthermore, certain aspects of the convergence proof are inspired by prior work in both the Euclidean [10, 75, 76, 81] and Riemannian [7, 9, 29] settings. Examples include the two stage adaptive step-size mechanism (see Proposition G.1) and specific properties of Riemannian (hyper)gradients (detailed in Appendix F).
>
> However, we point out that due to the geometric structures inherent in Riemannian manifolds, although certain aspects of Euclidean analysis extend naturally to Riemannian settings, there exist many theoretical challenges in the proof of the convergence analysis. More specifically, we will explain from the following points that the theoretical extension is not so intuitive and will encounter many technical difficulties:
>
> - Firstly, influenced by the curvature of the Riemannian manifolds, establishing the upper bound for the iterations $K_t$ (i.e., the total iterations for solving the lower-level problem at the $t$-th iteration) needs to consider the curvature constant $\zeta$, as shown in Proposition 3.2.
> - Secondly, different from the distance defined in Euclidean space, the trigonometric distance bound (cf. Lemma B.1 in our paper) also raises a difficulty in the theoretical analysis. Moreover, establishing the convergence of AdaRHD requires a thorough analysis of all sequences, $x_t$, $y_t^k$, and $v_t^n$, whose convergence properties also fundamentally depend on the trigonometric distance bound. This dependency highlights the inherent theoretical challenges in analyzing AdaRHD.
> - Thirdly, in contrast to existing methods for Riemannian bilevel optimization (RBO) [19, 29, 48], our proposed algorithm is the first to incorporate a fully adaptive step-size strategy. This eliminates the requirement for prior knowledge of problem-dependent parameters (e.g., strong convexity, Lipschitz smoothness, or curvature bounds), which may be particularly challenging to estimate in Riemannian settings. This advantage makes our algorithm more practical in the applications of RBO.
>
> ### **Questions**
>
> __*For Question 1*__:
>
> Thanks for your insightful question. We will state some novel and significant ideas of algorithm design in the following points:
>
> - (**Compare to the Euclidean counterpart**) Compared to the Euclidean adaptive algorithm D-TFBO [81], we introduce a tangent space conjugate gradient method (Algorithm 2) for solving the linear system. Our AdaRHD-CG algorithm demonstrates superior computational efficiency, with D-TFBO's Hessian-vector product complexity being $O(1/\epsilon^2)$, which is worse than ours $\tilde{O}(1/\epsilon)$, as evidenced by Table 1 (cf. first and last rows). In addition to the theoretical reduction in complexity, we have also observed in practice that AdaRHD-CG yields better results compared to AdaRHD-GD and RHGD (e.g., as demonstrated in the second subfigures of Figures 1 and 2, AdaRHD-CG achieves lower hypergradient errors compared to AdaRHD-GD and RHGD). This is likely since, in the current iteration, CG can achieve a better solution with less computational cost. We believe that applying it to the Euclidean counterpart should also yield good results.
>
> - (**Compare to related Riemannian method**) In Section 3.4, we extend our algorithm to employ retraction mappings for variable updates (AdaRHD-R), a computationally efficient alternative to exponential mappings preferred in practice. Among existing RBO methods, only RHGD [29] implements a similar extension. However, RHGD's performance critically depends on carefully tuned step-sizes requiring problem-specific parameters that are often impractical to obtain (especially the curvature $\zeta$). Our algorithm maintains RHGD's convergence rate while eliminating this parameter dependence, demonstrating greater robustness in practical applications.
>
> __*For Question 2*__:
>
> We appreciate this insightful question and would like to highlight the novel aspects of our theoretical techniques:
>
>
> - (**Riemannian Challenges**) Unlike Euclidean optimization, Riemannian optimization must account for the curvature $\zeta$, rendering some standard results established in [81] (Euclidean space) inapplicable. Specifically, the coefficient $\frac{1}{(\mu/\bar{b} - \zeta L_g^2/\bar{b}^2)}$ of the second term $\frac{1}{(\mu/\bar{b} - \zeta L_g^2/\bar{b}^2)}\log(\frac{\tilde{b}}{\epsilon_y})$ includes $\zeta$ and is a quadratic term w.r.t. the step-size $\bar{b}$, which is different from $\frac{\beta_{\max}}{\mu}$ (equivalent to $\frac{\bar{b}}{\mu}$ in our notations) in $P_t$ (equivalent to $K_t$ in our notation) of [81], as it demonstrates only linear dependence w.r.t. $\bar{b}$. This raises a critical problem that we should consider a much complex analysis of the step-size $\bar{b}$ (e.g., the monotonicity of $\frac{\mu}{\bar{b}} - \frac{\zeta L_g^2}{\bar{b}^2}$ w.r.t. $\bar{b}$), please refer to lines 719-723 for the value of $\bar{b}$. Furthermore, we need to use the trigonometric distance bound (Lemma B.1) instead of the classical Euclidean distance triangle inequality, which introduces theoretical challenges. For example, in the convergence analysis of the lower-level problem (i.e., establish the upper bound of $K_t$), it is different from the threshold $C_{\beta}$ (equivalent to $C_{b}$ in our notation) established in Equation (5) of [81], we should establish a complex threshold $C_b$ for convergence analysis (cf. Equation (18)).
>
>
> - (**Retraction Challenges**) In order to be more practical, we extend our algorithm to update variables via retraction mappings. Since retraction mappings serve as first-order approximations of exponential mappings [29], the error between exponential and retraction mappings must be accounted for when updating variables $x_t$ and $y_t$ at each iteration, which introduces additional challenges in the theoretical analysis. Although [29] explores this extension and presents theoretical results, such as the convergence of the lower-level problem and technical findings under retraction mappings (see Appendix E in [29]), the convergence analysis of our AdaRHD-R algorithm differs significantly and presents unique challenges. Specifically, AdaRHD-R employs a two stage step-size adaptation mechanism (see the proof of Proposition D.1, lines 860–897), whose theoretical challenge surpasses that of the algorithm in [29], which relies on well-tuned step-sizes and extensive prior knowledge of RBO-specific parameters. Moreover, deriving the upper bounds for the total iterations of the two stage step-size is more intricate in comparison to AdaRHD. For example, Lemmas G.1 and H.1, as well as Lemmas G.3 and H.3, demonstrate that the definitions of several constants (e.g., $\bar{L}_F$ in line 834, $\bar{\zeta}$ in line 852, and $C_b$ in line 857) in AdaRHD-R are more complex than those in AdaRHD, owing to the discrepancies between exponential and retraction mappings.
>
> ### **References**
> All the referenced citations mentioned above have the same numbering as in the article.

---

> > ### Comment · Reviewer_TNoH · 2025-08-05
> >
> > This reviewer sincerely appreciates the authors' thorough and thoughtful responses.

---

> > > ### Author Response · Authors · 2025-08-05
> > >
> > > We deeply appreciate the time you have taken to review our response and for your thoughtful feedback. Your constructive comments are extremely important in enhancing the quality and clarity of our paper.

---

### Official Review · Reviewer_NqyN · 2025-07-02

**Clarity:** 3
**Significance:** 3
**Originality:** 2
**Rating:** 5
**Confidence:** 4

**Summary:**

This article studies deterministic adaptive optimization algorithm complexity for Riemannian bilevel optimization problems. Adaptive Riemannian hypergradient descent algorithm is proposed and analyzed to achieve a fast convergence compared to non-adaptive methods. This algorithm is further extended to allow for the use of retraction mappings to main a same complexity bound. Numerical experiments show comparable and more robust performance of the proposed method compared to non-adaptive approaches.

**Questions:**

-	Regarding algorithm 1 (AdaRHD), should you add a stopping criterion to return an eps-stationary point? What would be the extra cost of this stopping criteria in terms of the computational complexity?
-	Clarify the U_x and U_y in assumption 3.2.
-	Should you use f instead of F in assumption 3.4, or over M_x instead?
-	Clarify the CG step in Proposition 3.2. What does it mean N_t in AdaRHD-CG, in terms of number of CG steps (could be more precise)?
-	Assumption 3.5 does the tilde D constant depend on t? If not, should you write i.e. there exists tilde D such that for all t>=0, ….
-	Fig 3 of numerical results, is this about the best iteration performance, last iteration performance? The main theory is about ergodic performance, which is related to the best iteration min_t || GF (x_t ||^2. This is not so clear by reading Section 4. If Fig 3 is about last iteration, i.e. performance of x_t, then I think that a discussion about the best iteration performance should also be included.
-	The epochs in Fig 3(a) and (b) should be made longer to see if the robustness of the proposed algorithm can be achieved, then t is large. Currently, it is too short to see that the algorithm converges to a similar performance level (y axis label is too small to read).

**Ethical Concerns:**

["NO or VERY MINOR ethics concerns only"]

**Limitations:**

There is a lack of discussion how to address stochastic gradients which are of interest in machine learning. It seems that this problem is quite challenging. A discussion or citation could be included in the article.

**Quality:**

3

**Strengths And Weaknesses:**

Strength: The article is well written and easy to follow. Motivated by a recent article [81], the main idea is to extend recent adaptive methods from Euclidean spaces to Riemmanian manifolds. The proposed algorithm is based on an adaGrad-type gradient norm strategy (algo 1). A main challenge is to analyze the convergence rate of the proposed method. This is presented in Theorem 3.1 and 3.2 (whose proofs are not checked).

Weakness: The extension of the algorithm 1 from [81] seems quite straightforward, therefore the originality of the algorithm is not strong. The main issue seems to be a lack of connection between convergence in theory and the numerical results. There is also a lack of discussion how to address stochastic gradients, which are of interest in machine learning.

Some of the results and assumptions should be further clarified (including some typos).

---

> ### Author Rebuttal · Authors · 2025-07-30
>
> We sincerely appreciate your recognition and your valuable constructive feedback. Your expert insights are essential to advancing our research.
>
> ### **Weaknesses**
> __*For originality*__:
>
> We appreciate your insightful comments. Intuitively, AdaRHD can be regarded as an extension of D-TFBO [81] to Riemannian manifolds, given their analogous algorithmic structures. However, the theoretical generalization is highly non-trivial and introduces several technical challenges. Below, we outline some key difficulties and innovations of our work:
>
> - (**Riemannian Curvature**) Owing to the geometric structures of Riemannian manifolds, deriving $K_t$ (the iterations of the lower-level problem) in Proposition 3.2 requires incorporating the curvature $\zeta$. Specifically, the coefficient $\frac{1}{(\mu/\bar{b} - \zeta L_g^2/\bar{b}^2)}$ in the second term of $K_t$ exhibits quadratic dependence on $\bar{b}$ and depends on $\zeta$. This contrasts with the coefficient $\frac{\beta_{\max}}{\mu}$ ($\frac{\bar{b}}{\mu}$ in our notation) in the second term of $P_t$ ($K_t$ in our notation) in [81, Proposition 2], which only depends linearly on $\bar{b}$. Consequently, this discrepancy and the quadratic dependence necessitate a more intricate analysis: we must examine the monotonicity of $\mu/\bar{b} - \zeta L_g^2/\bar{b}^2$ w.r.t. $\bar{b}$. The detailed analysis is provided in lines 746–754.
> - (**Riemannian Distance**) When analyzing Proposition 3.2 and Theorem 1, we must employ the trigonometric distance (Lemma B.1) instead of Euclidean counterpart. This issue substantially increases the difficulty of the analysis compared to [81]. For instance, unlike the threshold $C_{\beta}$ ($C_{b}$ in our paper) in [81, Equation (5)], our analysis necessitates a more sophisticated threshold $C_b$ (Equation (18)) to account for the manifold structure.
> - (**Related Method**) Compared to existing methods for RBO [19, 29, 48], our algorithm is the first to incorporate a fully adaptive step-size strategy. This eliminates the need for prior knowledge of computationally expensive parameters (e.g., curvature). Such adaptability significantly broadens the applicability of our method to RBO. Furthermore, we utilize a CG method [7, 69] to solve linear system in Algorithm 1. This technique significantly reduces computational costs (particularly Hessian-vector product,) and achieves better convergence guarantees compared to the gradient descent in [81].
>
> __*For experiments*__:
>
> We appreciate this observation and will incorporate related discussion in our revision. Please refer to our response to Questions 6 and 7.
>
> __*For stochastic setting*__:
>
> We regret any confusion. Developing an adaptive stochastic method for RBO represents a promising direction, and we also briefly discuss potential extensions to it and cite relevant works in Appendix J. We will incorporate this discussion into the main text and further elaborate on its significance for machine learning.
>
> Although there exist adaptive stochastic methods for solving single-level problems over Riemannian manifolds (RSGD [B13], RSVRG [Z16], RSRG [K18], R-SPIDER [Z18; Z19], RASA [K19], RAMSGrad [B19], Riemannian SAM [Y24], RNGD [H24], and Riemannian adaptive framework [S25]), designing an adaptive stochastic method for RBO poses substantial challenges and may lie beyond this paper’s scope. Moreover, existing stochastic methods for RBO, RF2SA [19], RSHGD-HINV [29], and RieSBO [48], rely on well-tuned step-sizes. The inherent randomness in (hyper)gradient estimates, combined with Riemannian geometry, complicates the design of such an extension:
> - (**Stochastic Riemannian hypergradient**) Similar to the approximate Riemannian hypergradient in Equation (5), the stochastic variant needs to consider three distinct sources of randomness: stochastic Riemannian gradient, Hessian, and partial derivative. The simultaneous presence of these randomnesses significantly complicates the convergence analysis of adaptive stochastic RBO algorithms, making it more challenging than methods for single-level problems.
> - (**Maximum iterations for two stages of step-sizes**): In our paper, the two stages of step-sizes (Proposition G.1) necessitate a comprehensive analysis of all sequences $x_t$, $y_t^k$, and $v_t^n$; we further need to incorporate stochastic errors if in the stochastic setting. The challenge is particularly pronounced for $x_t$, which is updated by a stochastic approximate Riemannian hypergradient, affected by three distinct randomnesses (cf. the above analysis). These challenges highlight the difficulty of extending AdaRHG to the stochastic setting compared to [19, 29, 48].
>
> On the other hand, we previously conducted some experiments in stochastic setting, but did not obtain impressive results, potentially due to our implementation limitations. Nevertheless, we believe this extension is a promising direction, though it may require better techniques to ensure both theoretical convergence and practical applicability.
>
> __*For clarity*__: We appreciate your insightful observations on the clarity of our paper. We will address these issues in our revision.
>
> [B13] Bonnabel. Stochastic gradient descent on Riemannian manifolds
>
> [Z16] Zhang et al. Riemannian SVRG: Fast stochastic optimization on Riemannian manifolds
>
> [K18] Kasai et al. Riemannian stochastic recursive gradient algorithm
>
> [Z18] Zhang et al. R-SPIDER: A fast Riemannian stochastic optimization algorithm with curvature independent rate
>
> [Z19] Zhou et al. Faster first-order methods for stochastic non-convex optimization on Riemannian manifolds
>
> [K19] Kasai et al. Riemannian adaptive stochastic gradient algorithms on matrix manifolds
>
> [B19] Bécigneul and Ganea. Riemannian adaptive optimization methods
>
> [Y24] Yun and Yang. Riemannian SAM: Sharpness-aware minimization on Riemannian manifolds
>
> [H24] Hu et al. Riemannian natural gradient methods
>
> [S25] Sakai and Iiduka. A general framework of Riemannian adaptive optimization methods with a convergence analysis
>
>
> ### **Questions**
> __*For Question 1*__:
>
> We appreciate your insightful question. Indeed, we can incorporate a stopping criterion, $\\|\hat{G}F(x_t,y_t^{K_t},v_t^{N_t})\\|^2 \le 1/T$, into the "for loop" of Algorithm 1 to terminate early, where $\hat{G}F(x_t,y_t^{K_t},v_t^{N_t})$ denotes the approximate Riemannian hypergradient in Equation (5). By Lemma 3.1, $\\|\hat{G}F(x_t,y_t^{K_t},v_t^{N_t})\\|^2 \le 1/T$ also ensures $\\|GF(x_t)\\|^2 \le O(1/T)$, implying that $x_t$ is an $O(1/T)$-stationary point. Furthermore, this modification does not increase the overall computational complexity, as we only need to perform an additional norm of hypergradient at each step.
>
> __*For Question 2*__:
>
> We apologize for any confusion. The notations $\mathcal{U}_x$ and $\mathcal{U}_y$ denote the components of $\mathcal{U}$, such that $\mathcal{U} = \mathcal{U}_x \times \mathcal{U}_y$, where $\mathcal{U} \subset \mathcal{M}_x \times \mathcal{M}_y$.
>
> __*For Question 3*__:
>
> We appreciate your careful reading. It should be "The minimum of $F$ over $\mathcal{M}_x$, denoted as $F^*$, is lower-bounded." As referenced in line 701, we only utilize the minimum of $F$ over $\mathcal{M}_x$.
>
> __*For Question 4*__:
>
> We apologize for any confusion. In both AdaRHD-GD and AdaRHD-CG, $N_t$ is reused to denote the total iterations required to solve the linear system, i.e., in AdaRHD-CG, $N_t$ represents the total iterations needed by Algorithm 2 to solve linear system at $t$-th iteration.
>
> __*For Question 5*__:
>
> We apologize for any confusion. You are correct that $\tilde{D}$ is independent of the iteration $t$, and we will address it in our revision.
>
> __*For Questions 6 and 7*__:
>
> We apologize for not clearly explaining the numerical results and for omitting the key metrics, e.g., ergodic performance $\min_t \\|GF (x_t)\\|^2$. It is true that Figure 3 only displays the performance of last iteration. Since we cannot include figures, we instead provide comparative tables summarizing the ergodic performance $\min_t \\|GF (x_t)\\|^2$ for two step-sizes (0.1 and 5) with 5 seeds (owing to space constraints, we only present two step-sizes). Moreover, due to a previous storage mistake, $\\|GF (x_t)\\|^2$ was not recorded, so we reran the experiments in this period. Due to time constraints, we limited the epochs to 300, but it was enough to observe the efficiency and robustness of our algorithm.
>
> Step-size 0.1:
>
> |Epoch|1|50|100|200|300|
> |:-----:|:--:|:--:|:---:|:---:|:---:|
> |AdaRHD-CG|5.3e-01 (6.3e-02)|5.2e-02 (4.0e-03)|1.5e-02 (1.5e-03)|3.4e-03 (5.9e-04)|1.5e-03 (4.4e-04)|
> |AdaRHD-GD|2.1e-01 (4.9e-02)|4.3e-03 (3.4e-03)|2.4e-03 (4.4e-03)|4.7e-04 (9.2e-04)|1.8e-04 (3.7e-04)|
> |RHGD-20|1.3e+01 (1.8e+00)|4.9e-02 (3.2e-03)|1.7e-02 (7.2e-03)|8.5e-03 (1.1e-02)|7.3e-03 (1.2e-02)|
> |RHGD-50|6.6e+00 (1.2e+00)|5.0e-02 (4.8e-03)|1.3e-02 (4.3e-04)|3.1e-03 (6.2e-04)|1.8e-03 (1.1e-03)|
>
> Step-size 5:
>
> |Epoch|1|50|100|200|300|
> |:-----:|:--:|:--:|:---:|:---:|:---:|
> |AdaRHD-CG|1.1e+02 (1.0e+02)|2.7e-01 (1.1e-01)|1.6e-01 (1.1e-01)|4.4e-02 (4.2e-02)|2.6e-02 (2.2e-02)|
> |AdaRHD-GD|2.4e+04 (2.9e+04)|4.0e-02 (5.0e-02)|1.5e-03 (1.1e-03)|1.8e-04 (1.2e-04)|4.8e-05 (2.1e-05)|
> |RHGD-20|2.0e+04 (2.0e+04)|6.3e+02 (4.1e+02)|4.5e+02 (1.4e+02)|4.1e+02 (1.2e+02)|4.1e+02 (1.2e+02)|
> |RHGD-50|1.1e+04 (8.0e+03)|5.1e+02 (2.3e+02)|5.0e+02 (2.3e+02)|4.6e+02 (2.5e+02)|4.1e+02 (1.9e+02)|
>
> These results show that our algorithm achieves superior performance while exhibiting greater robustness than RHGD, further validating the efficiency and reliability of our method. We will incorporate the figures of ergodic performance in our revision. Additionally, we appreciate the feedback regarding the y-axis label and will address this by increasing its font size in the revision.
>
> ### **Limitation**
> Thanks again for pointing this out. Please refer to our response to "For stochastic setting" in Weaknesses.
>
> ### **References**
> The reference numbers above correspond to those cited in the original paper.

---

### Official Review · Reviewer_FZvm · 2025-07-03

**Clarity:** 3
**Significance:** 3
**Originality:** 3
**Rating:** 5
**Confidence:** 3

**Summary:**

This paper proposes AdaRHD, an adaptive method for Riemannian bilevel optimization (RBO) that achieves comparable complexity of other RBO methods, but that does not require any knowledge of problem parameters, which can be unknown or hard to estimate in practice. To this end, they adapt the double-loop version of the tuning-free bilevel optimziation method (D-TFBO), which is suitable for general bilevel problems, to the Riemannian setting. To adaptively compute step-sizes, AdaRHD (and D-TFBO) employ the "inverse of cumulative gradient norm" approach introduced to establish linear convergence for AdaGrad-Norm.
Similarly, AdaRHD adheres to the algorithmic structure of D-TFBO, efficiently computing the hypergradient in the inner-loop by solving a quadratic problem. To validate AdaRHD, two experiments are presented, in which AdaRHD is competitive with other RBO methods and seemingly are more robust to hyperparameter initialization.

**Questions:**

7.1: AdaRHD is an extension of D-TFBO to the Riemannian setting. What would be the main challenges to extend the single-loop variant of D-TFBO?
7.2: Qualitatively speaking, does the step-size adaption also works in two stages like in AdaGrad-Norm or are there any differences?
7.3: On the paragraph following Proposition 3.2, it is mentioned that epsilon_y and epsilon_v must be sufficiently small to achieve O(1/epsilon) complexity. How can such small epsilon be determined in practice? What happens if epsilon is not small enough? Do non-adaptive methods also have such a requirement?

**Ethical Concerns:**

["NO or VERY MINOR ethics concerns only"]

**Final Justification:**

The authors have provided a comprehensive and thoughtful rebuttal that adequately addresses my primary concerns from the initial review. The theory is sufficiently novel and the experiments support it. I will raise my score.

**Limitations:**

yes

**Quality:**

3

**Strengths And Weaknesses:**

Strenghts: The paper is clearly-written, well-organized, presents relevant references to understand it and the results seem to be correct (although I did not check them carefully.) The experimental results corroborate that AdaRHD indeed achieves comparable performance to other RBO methods while dispensing with problem parameters and being more robust to different initializations.
Weaknesses: AdaRHD seems to be a verbatim translation of D-TFBO to the Riemannian setting. Given that I am not an expert on RBO, I cannot fairly assess the technical merits of deriving theoretical results analogue to those of D-TFBO for the Riemannian setting, which seems to be the second main contribution of this paper. To clarify the difficulty of proving the theoretical results, passages such as the paragraph after Proposition 3.2 could elaborate a bit more on such obstacles. In the same vein, there could be some intuition of why the method works. For example, does step-size adaption also works in two stages like in AdaGrad-Norm? I would be happy to raise the clarity score to maximum with the addition of the above discussion/intuition.
Finally, in my understanding, the main contribution is to propose a method that could be used off the shelf, but there are not quite enough experiments to be sure. Perhaps this would be clearer to an RBO expert, and I understand it can be difficult to add more experiments during the rebuttal phase.

---

> ### Author Rebuttal · Authors · 2025-07-29
>
> Thanks for providing these valuable suggestions, and we will make the corresponding revisions and additions to our paper.
>
> ### **Weaknesses**
>
> __*For novelty*__:
>
> We appreciate this insightful feedback. Although AdaRHD may appear as a Riemannian extension of D-TFBO due to their similar algorithmic structures, the theoretical extension is highly non-trivial and introduces several technical challenges that D-TFBO does not address. Below, we outline the key difficulties and innovations:
>
> - (**Riemannian Curvature**) Due to the geometric structures in Riemannian manifolds, determining the upper bound for the iterations $K_t$ requires incorporating the curvature constant $\zeta$, as demonstrated in Proposition 3.2. Specifically, the coefficient $\frac{1}{(\mu/\bar{b} - \zeta L_g^2/\bar{b}^2)}$ in the second term of $K_t$ exhibits a quadratic relationship on $\bar{b}$ while being influenced by the curvature $\zeta$. This differs from the coefficient $\frac{\beta_{\max}}{\mu}$ ($\frac{\bar{b}}{\mu}$ in our notation) in the second term of $P_t$ ($K_t$ in our notation) of [81, Proposition 2], which shows only linear dependence w.r.t. $\bar{b}$. Consequently, this discrepancy and quadratic relationship of ours necessitates a more intricate analysis for $\bar{b}$. Specifically, we must examine the monotonicity of $\frac{\mu}{\bar{b}} - \frac{\zeta L_g^2}{\bar{b}^2}$ w.r.t. $\bar{b}$ and determine an appropriate value for it. The detailed derivation of $\bar{b}$ can be found in lines 746–754.
>
> - (**Riemannian Distance**) Additionally, when establishing the upper bound of $K_t$, we must employ the trigonometric distance bound instead of Euclidean distance (Lemma G.3, Equation (24), and line 741). This also significantly increases the substantial challenges of the convergence analysis compared to Euclidean space. For instance, unlike the threshold $C_{\beta}$ ($C_{b}$ in our notation) established in Equation (5) of [81], we should establish a complex threshold $C_b$ for convergence analysis (cf. Equation (18)).
>
> - (**Related Method**) In comparison to existing methods for RBO [19, 29, 48], our algorithm is the first to incorporate a fully adaptive step-size strategy, which eliminates the requirement for prior knowledge of problem-dependent parameters. This practical advantage enhances our method's applicability to RBO problems. Additionally, in Algorithm 1, we employ the conjugate gradient (CG) method [7, 69] to solve the linear system, which is more efficient than the gradient descent (GD) solver in [81]. The CG method significantly reduces computational costs while offering better convergence guarantees compared to the GD solver employed in D-TFBO [81]. At the same time, we have also observed in practice that AdaRHD-CG yields better results compared to AdaRHD-GD.
>
> __*For clarity*__:
>
> We appreciate the valuable feedback and will incorporate a more detailed discussion to clarify the challenges in proving the theoretical results of AdaRHD, as well as the intuition behind the effectiveness of the method. In the revision, we will include in the article phrases such as: "Similar to AdaGrad-Norm [80], the step-size adaptation operates in two stages: For the lower-level problem, Stage 1 requires at most $\frac{\log(C_b^2/b_0^2)}{\log(1 + \epsilon_y/C_b^2)}$ iterations, while Stage 2 requires at most $\frac{1}{({\mu}/{\bar{b}} - \zeta {L_{g}^2}/{\bar{b}^2})}\log\left(\frac{\tilde{b}}{\epsilon_y}\right)$ iterations. Furthermore, due to the geometric properties of Riemannian manifolds (e.g., trigonometric distance bound), these upper bounds cannot be directly derived from [81], and will introduce additional difficulties."
>
> __*For experiments*__:
>
> We sincerely apologize for any confusion. Due to space constraints, we only present the experimental results of two problems in the main text, and have included two additional experiments in Appendix I.2. Additionally, we appreciate your constructive advice. Here, following Sections 7.1 of [48], we incorporate two additional experiments, which have the following forms:
> $$
> \min_{p \in \Delta_n} \\|p - 1/n\\|^2-\sum_{i=1}^n p_i \ell (y;\xi_i) ~~ \mathrm{s.t.} ~~ y \in \underset{y \in \mathbb{S}\_{++}^d}{\mathrm{argmin}} \sum_{i=1}^n p_i \ell(y;\xi_i),
> $$
> Here, $\Delta_n$ and $\mathbb{S}\_{++}^d$ represent the multinomial manifold (or probability simplex) and positive-definite matrix manifold, respectively, $\ell(y;\xi_i)$ represents the geodesic distance and log likelihood for the robust Karcher mean and robust maximum likelihood estimation problems, respectively. Please refer to [48, Sections 7.1] for details.
>
> We use the same problem scales as those in [48] and set the step-size $\eta_x$ = $\eta_y = 1/\alpha_0 = 1/\beta_0 = 1/\gamma_0 = 0.1$, the other settings remain the same as those in Section 4.2 of our paper. On the other hand, we have also added the results of the original experiments in our response to reviewer NqyN. In the revision, we will include larger-scale experiments on this problem.
>
> For the robust Karcher mean experiment, the relationship between the ergodic performance $\min_t \\|\mathcal{G}F (x_t)\\|^2$ and epoch is shown in the table below:
>
> |Epoch|1|10|50|100|
> |:-----:|:--:|:--:|:---:|:---:|
> |AdaRHD-CG|1.31e+01 (5.84e+00)|5.43e-01 (5.89e-01)|8.50e-04 (1.70e-03)|2.71e-11 (3.14e-11)|
> |AdaRHD-GD|1.31e+01 (5.84e+00)|5.41e-01 (5.82e-01)|5.63e-04 (1.12e-03)|1.22e-11 (2.19e-11)|
> |RHGD-20|1.32e+01 (5.88e+00)|1.04e+01 (5.72e+00)|8.44e+00 (4.13e+00)|8.23e+00 (4.05e+00)|
> |RHGD-50|1.32e+01 (5.88e+00)|9.72e+00 (5.40e+00)|9.72e+00 (5.40e+00)|8.20e+00 (4.55e+00)|
>
> The relationship between time (seconds) and epoch is shown in the table below:
>
> |Epoch|1|10|50|100|
> |:-----:|:--:|:--:|:---:|:---:|
> |AdaRHD-CG|0.79 (0.46)|37.70 (6.54)|126.76 (37.43)|180.48 (56.43)|
> |AdaRHD-GD|1.44 (0.39)|18.62 (3.63)|74.70 (6.64)|139.63 (8.58)|
> |RHGD-20|6.35 (2.58)|101.42 (47.05)|567.85 (194.74)|1199.52 (422.74)|
> |RHGD-50|11.57 (0.92)|144.27 (35.10)|764.78 (160.69)|1542.63 (365.48)|
>
> For the robust maximum likelihood estimation experiment, the relationship between the ergodic performance $\min_t \\|\mathcal{G}F (x_t)\\|^2$ and epoch is shown in the table below (here, we do not report the results of RHGD-20 since it fails to converge under this step-size):
>
> |Epoch|1|10|50|100|
> |:-----:|:--:|:--:|:---:|:---:|
> |AdaRHD-CG|1.27e+01 (1.54e+00)|2.38e-01 (1.25e-01)|1.55e-06 (2.79e-06)|8.76e-08 (1.34e-07)|
> |AdaRHD-GD|1.27e+01 (1.54e+00)|1.97e-03 (8.39e-04)|2.56e-05 (2.36e-05)|1.22e-06 (1.54e-06)|
> |RHGD-50|1.42e+01 (1.70e+00)|3.86e-01 (2.45e-01)|6.09e-02 (7.52e-02)|4.11e-02 (5.99e-02)|
>
> The relationship between time (seconds) and epoch is shown in the table below:
>
> |Epoch|1|10|50|100|
> |:-----:|:--:|:--:|:---:|:---:|
> |AdaRHD-CG|0.70 (0.80)|15.26 (2.85)|29.69 (2.68)|44.69 (2.80)|
> |AdaRHD-GD|6.05 (2.06)|17.87 (1.89)|29.76 (1.95)|44.80 (1.99)|
> |RHGD-50|2.62 (0.13)|26.66 (0.60)|129.08 (4.07)|257.49 (7.55)|
>
> As shown in the tables above, AdaRHD can achieve superior results more efficiently and robustly compared to RHGD. Additionally, we emphasize that even when the initial step-size is set larger, such as 0.5, AdaRHD still converges, whereas RHGD-50 and RHGD-20 do not (due to character and time constraints, we do not present the corresponding results here), which further demonstrates the robustness of our algorithm.
>
> ### **Questions**
>
> __*For Question 1*__:
>
> We appreciate this insightful question. While maintaining a structure similar to S-TFBO [81], this single-loop extension requires replacing Euclidean (hyper)gradients with their Riemannian counterparts. However, this extension presents some primary challenges:
>
> First, the same as AdaRHD, in Riemannian space, we should use trigonometric distance bounds instead of Euclidean bounds. This fundamentally alters the convergence analysis for two stage step-size adaptation compared to S-TFBO [81]. For instance, in S-TFBO [81], the step-sizes for $v_{t+1}$ and $x_{t+1}$ are chosen as $\frac{1}{\varphi_{t+1}}$ and $\frac{1}{\alpha_{t+1}\varphi_{t+1}}$ respectively, these step-sizes are not straightforward and require careful design. In the Riemannian setting, this step-size strategy may be ineffective, further complicating the convergence analysis due to both the trigonometric distance bounds and the absence of key inequalities that are valid only in Euclidean space.
>
> Second, many technical results essential for proving AdaRHD’s convergence (e.g., Equations (24), (25), and (28)) rely on approximate solutions to the lower-level problem and linear system. However, it remains unclear whether these results naturally extend to the single-loop variant, as the latter does not solve subproblems with sufficient accuracy. In addition, although we have conducted preliminary experiments to extend S-TFBO [81] to the Riemannian setting without convergence analysis, the experimental results were unsatisfactory. This could be due to limitations in our implementation, and may also require better adaptation strategy. Consequently, this paper focuses solely on the double-loop structured algorithm. Nevertheless, we agree that extending the single-loop variant constitutes a promising research direction.
>
> __*For Question 2*__:
>
> Yes, you are correct that, similar to AdaGrad-Norm, AdaRHD's step-size adaptation also operates in two stages. For details, please refer to Proposition G.1 and our response to "For clarity" in Weaknesses.
>
> __*For Question 3*__:
>
> We sincerely apologize for this confusion. The condition that "$\epsilon_y$ and $\epsilon_v$ are sufficiently small" is introduced primarily to demonstrate that $\frac{1}{\log(1+\epsilon_y)}$ has the same order as $\frac{1}{\epsilon_y}$, i.e., $\frac{1}{\log(1+\epsilon_y)} = \mathcal{O}(1/\epsilon_y)$. In practice, this condition is unnecessary, and setting $\epsilon_y = \epsilon_v = 1/T$ (cf. line 1 of Algorithm 1) suffices.
>
>
> ### **References**
> All the referenced citations mentioned above have the same numbering as in the article.

---

### Note · Authors · 2025-08-13

Dear NeurIPS 2025 ACs, SACs, PCs, and Reviewers,

We sincerely appreciate your hard work and the thoughtful, constructive feedback provided by the reviewers. We are grateful for the positive scores and recognition of all the reviewers. Particularly, we deeply appreciate the reviewers' recognition of our paper's **writing quality**, **theoretical contributions**, and **experimental results**.

**Summary of Rebuttal and Discussion**

We have carefully addressed the reviewers' concerns, and the key points clarified during the rebuttal and discussion periods are summarized as follows:

**Theoretical Novelty**: We emphasized that our theoretical analysis is highly non-trivial, addressing key technical challenges in:
- Two-stage step-size adaptation;
- Riemannian geometry;
- Convergence analysis.

Despite these challenges, this paper provides the first method to incorporate a fully adaptive step-size strategy for RBO, which achieves a convergence rate of $O(1/\epsilon)$, matching well-tuned baselines.

**Additional Experiments**: To address the concerns regarding the experimental results, we conducted further experiments:
- New experiments: Robust Karcher mean and robust maximum likelihood estimation problems (from [48]).
- Extended evaluation: We reran the original experiments in our paper with longer epochs and provided clearer interpretations.

All the additional experimental results confirm the efficiency and robustness of our algorithm.

**Algorithmic Extensions**: We provided detailed discussions on three key questions regarding algorithmic variants, including the single-loop variant, adaptive stochastic method, and resolving the complexity gap. For each question, we cited relevant literature, clarified main theoretical challenges, and suggested potential solutions.

**Other Clarifications**: We provided detailed derivations and responses to all remaining questions, including some misunderstandings and typos in the paper.

Once again, we deeply appreciate the time and dedication of all ACs, SACs, PCs, and reviewers.

Best regards,

The authors of Paper 3592

---

### Decision · Program_Chairs · 2025-09-17

**Decision:**

Accept (poster)

**Comment:**

## Summary
The paper introduces a new method for solving bilevel optimization problems on Riemannian manifolds. It adjusts the learning steps based on the strength of the gradients over time. This is a fresh idea and helps improve performance in tasks that involve such spaces.

## Strengths
- **New Idea**: Adapting step sizes using Riemannian gradients is a smart and original approach.
- **Experiments**: The authors tested their method and showed it performs better than existing approaches.
- **Reviewer Feedback**: All reviewers appreciated the idea and found the paper well-written and useful.

## Improvement Opportunities
- The initial version didn’t clearly explain some technical challenges, especially the transition from Euclidean settings to the manifold setup.
- More detailed experiments in the main paper would strengthen the results. (Already clarified during rebuttal).

## Rebuttal and Discussion
- Reviewers asked for more clarity and additional experiments.
- The authors responded well and provided new results.
- Reviewers were satisfied with the changes and felt their concerns were addressed.
- The discussion showed consistent engagement from the authors.

## Decision
I recommend **accepting** the paper. For the camera-ready version, please include the new results and edits discussed during the review.